# Neural Policy Gradient Methods: Global Optimality and Rates of Convergence

**Lingxiao Wang**[*†]   **Qi Cai**[*‡]   **Zhuoran Yang**[§]   **Zhaoran Wang**[¶]

## Abstract

Policy gradient methods with actor-critic schemes demonstrate tremendous empirical successes, especially when the actors and critics are parameterized by neural networks. However, it remains less clear whether such "neural" policy gradient methods converge to globally optimal policies and whether they even converge at all. We answer both the questions affirmatively under the overparameterized two-layer neural-network parameterization. In detail, assuming independent sampling, we prove that neural natural policy gradient converges to a globally optimal policy at a sublinear rate. Also, we show that neural vanilla policy gradient converges sublinearly to a stationary point. Meanwhile, by relating the suboptimality of the stationary points to the representation power of neural actor and critic classes, we prove the global optimality of all stationary points under mild regularity conditions. Particularly, we show that a key to the global optimality and convergence is the "compatibility" between the actor and critic, which is ensured by sharing neural architectures and random initializations across the actor and critic. To the best of our knowledge, our analysis establishes the first global optimality and convergence guarantees for neural policy gradient methods. [1]

## 1 Introduction

In reinforcement learning (Sutton and Barto, 2018), an agent aims to maximize its expected total reward by taking a sequence of actions according to a policy in a stochastic environment, which is modeled as a Markov decision process (MDP) (Puterman, 2014). To obtain the optimal policy, policy gradient methods (Williams, 1992; Baxter and Bartlett, 2000; Sutton et al., 2000) directly maximize the expected total reward via gradient-based optimization. As policy gradient methods are easily implementable and readily integrable with advanced optimization techniques such as variance reduction (Johnson and Zhang, 2013; Papini et al., 2018) and distributed optimization (Mnih et al., 2016; Espeholt et al., 2018), they enjoy wide popularity among practitioners. In particular, when the policy (actor) and action-value function (critic) are parameterized by neural networks, policy gradient methods achieve significant empirical successes in challenging applications, such as playing Go (Silver et al., 2016; 2017), real-time strategy gaming (Vinyals et al., 2019), robot manipulation (Peters and Schaal, 2006; Duan et al., 2016), and natural language processing (Wang et al., 2018). See Li (2017) for a detailed survey.

In stark contrast to the tremendous empirical successes, policy gradient methods remain much less well understood in terms of theory, especially when they involve neural networks. More specifically, most existing work analyzes the REINFORCE algorithm (Williams, 1992; Sutton et al., 2000), which estimates the policy gradient via Monte Carlo sampling. Based on the recent progress in non-convex optimization, Papini et al. (2018); Shen et al. (2019); Xu et al. (2019a); Karimi et al. (2019);

---

[*]equal contribution

[†]Northwestern University; `lingxiaowang2022@u.northwestern.edu`

[‡]Northwestern University; `qicai2022@u.northwestern.edu`

[§]Princeton University; `zy6@princeton.edu`

[¶]Northwestern University; `zhaoranwang@gmail.com`

[1]See https://arxiv.org/abs/1909.01150 for the full version.

Zhang et al. (2019) establish the rate of convergence of REINFORCE to a first- or second-order stationary point. However, the global optimality of the attained stationary point remains unclear. A more commonly used class of policy gradient methods is equipped with the actor-critic scheme (Konda and Tsitsiklis, 2000), which alternately estimates the action-value function in the policy gradient via a policy evaluation step (critic update), and performs a policy improvement step using the estimated policy gradient (actor update). The global optimality and rate of convergence of such a class are even more challenging to analyze than that of REINFORCE. In particular, the policy evaluation step itself may converge to an undesirable stationary point or even diverge (Tsitsiklis and Van Roy, 1997), especially when it involves both nonlinear action-value function approximator, such as neural network, and temporal-difference update (Sutton, 1988). As a result, the estimated policy gradient may be biased, which possibly leads to divergence. Even if the algorithm converges to a stationary point, due to the nonconvexity of the expected total reward with respect to the policy as well as its parameter, the global optimality of such a stationary point remains unclear. The only exception is the linear-quadratic regulator (LQR) setting (Fazel et al., 2018; Malik et al., 2018; Tu and Recht, 2018; Yang et al., 2019a; Bu et al., 2019), which is, however, more restrictive than the general MDP setting that possibly involves neural networks.

To bridge the gap between practice and theory, we analyze neural policy gradient methods equipped with actor-critic schemes, where the actors and critics are represented by overparameterized two-layer neural networks. In detail, we study two settings, where the policy improvement steps are based on vanilla policy gradient and natural policy gradient, respectively. In both settings, the policy evaluation steps are based on the TD(0) algorithm (Sutton, 1988) with independent sampling. In the first setting, we prove that neural vanilla policy gradient converges to a stationary point of the expected total reward at a $1/\sqrt{T}$-rate in the expected squared norm of the policy gradient, where $T$ is the number of policy improvement steps. Meanwhile, through a geometric characterization that relates the suboptimality of the stationary points to the representation power of the neural networks parameterizing the actor and critic, we establish the global optimality of all stationary points under mild regularity conditions. In the second setting, through the lens of Kullback-Leibler (KL) divergence regularization, we prove that neural natural policy gradient converges to a globally optimal policy at a $1/\sqrt{T}$-rate in the expected total reward. In particular, a key to such global optimality and convergence guarantees is a notion of compatibility between the actor and critic, which connects the accuracy of policy evaluation steps with the efficacy of policy improvement steps. We show that such a notion of compatibility is ensured by using shared neural architectures and random initializations for both the actor and critic, which is often used as a practical heuristic (Mnih et al., 2016). To our best knowledge, our analysis gives the first global optimality and convergence guarantees for neural policy gradient methods, which corroborate their significant empirical successes.

**Related Work.** In contrast to the huge body of empirical literature on policy gradient methods, theoretical results on their convergence remain relatively scarce. In particular, Sutton et al. (2000) and Kakade (2002) analyze vanilla policy gradient (REINFORCE) and natural policy gradient with compatible action-value function approximators, respectively, which are further extended by Konda and Tsitsiklis (2000); Peters and Schaal (2008); Castro and Meir (2010) to incorporate actor-critic schemes. Most of this line of work only establishes the asymptotic convergence based on stochastic approximation techniques (Kushner and Yin, 2003; Borkar, 2009) and requires the actor and critic to be parameterized by linear functions. Another line of work (Papini et al., 2018; Xu et al., 2019a;b; Shen et al., 2019; Karimi et al., 2019; Zhang et al., 2019) builds on the recent progress in nonconvex optimization to establish the nonasymptotic rates of convergence of REINFORCE (Williams, 1992; Baxter and Bartlett, 2000; Sutton et al., 2000) and its variants, but only to first- or second-order stationary points, which, however, lacks global optimality guarantees. Moreover, when actor-critic schemes are involved, due to the error of policy evaluation steps and its impact on policy improvement steps, the nonasymptotic rates of convergence of policy gradient methods, even to first- or second-order stationary points, remain rather open.

Compared with the convergence of policy gradient methods, their global optimality is even less explored in terms of theory. Fazel et al. (2018); Malik et al. (2018); Tu and Recht (2018); Yang et al. (2019a); Bu et al. (2019) prove that policy gradient methods converge to globally optimal policies in the LQR setting, which is more restrictive. In very recent work, Bhandari and Russo (2019) establish the global optimality of vanilla policy gradient (REINFORCE) in the general MDP setting. However, they require the policy class to be convex, which restricts its applicability to the tabular

and LQR settings. In independent work, Agarwal et al. (2019) prove that vanilla policy gradient and natural policy gradient converge to globally optimal policies at $1/\sqrt{T}$-rates in the tabular and linear settings. In the tabular setting, their rate of convergence of vanilla policy gradient depends on the size of the state space. In contrast, we focus on the nonlinear setting with the actor-critic scheme, where the actor and critic are parameterized by neural networks. It is worth mentioning that when such neural networks have linear activation functions, our analysis also covers the linear setting, which is, however, not our focus. In addition, Liu et al. (2019) analyze the proximal policy optimization (PPO) and trust region policy optimization (TRPO) algorithms (Schulman et al., 2015; 2017), where the actors and critics are parameterized by neural networks, and establish their $1/\sqrt{T}$-rates of convergence to globally optimal policies. However, they require solving a subproblem of policy improvement in the functional space using multiple stochastic gradient steps in the parameter space, whereas vanilla policy gradient and natural policy gradient only require a single stochastic (natural) gradient step in the parameter space, which makes the analysis even more challenging.

There is also an emerging body of literature that analyzes the training and generalization error of deep supervised learning with overparameterized neural networks (Daniely, 2017; Jacot et al., 2018; Wu et al., 2018; Allen-Zhu et al., 2018a;b; Du et al., 2018a;b; Zou et al., 2018; Chizat and Bach, 2018; Jacot et al., 2018; Li and Liang, 2018; Cao and Gu, 2019a;b; Arora et al., 2019; Lee et al., 2019), especially when they are trained using stochastic gradient. See Fan et al. (2019) for a detailed survey. In comparison, our focus is on deep reinforcement learning with policy gradient methods. In particular, the policy evaluation steps are based on the TD(0) algorithm, which uses stochastic semigradient (Sutton, 1988) rather than stochastic gradient. Moreover, the interplay between the actor and critic makes our analysis even more challenging than that of deep supervised learning.

**Notation.** For distribution $\mu$ on $\Omega$ and $p > 0$, we define $\|f(\cdot)\|_{\mu,p} = (\int_{\Omega} |f|^p \mathrm{d}\mu)^{1/p}$ as the $L_p(\mu)$ norm of $f$. We define $\|f(\cdot)\|_{\mu,\infty} = \inf\{C \geq 0 : |f(x)| \leq C \text{ for } \mu\text{-almost every } x\}$ as the $L_\infty(\mu)$-norm of $f$. We write $\|f\|_{\mu,p}$ for notational simplicity when the variable of $f$ is clear from the context. We further denote by $\|\cdot\|_\mu$ the $L_2(\mu)$-norm for notational simplicity. For a vector $\phi \in \mathbb{R}^n$ and $p > 0$, we denote by $\|\phi\|_p$ the $\ell_p$-norm of $\phi$. We denote by $x = ([x]_1^\top, \ldots, [x]_m^\top)^\top$ a vector in $\mathbb{R}^{md}$, where $[x]_i \in \mathbb{R}^d$ is the $i$-th block of $x$ for $i \in [m]$.

## 2 BACKGROUND

In this section, we introduce the background of reinforcement learning and policy gradient methods.

**Reinforcement Learning.** A discounted Markov decision process (MDP) is defined by tuple $(\mathcal{S}, \mathcal{A}, \mathcal{P}, \zeta, r, \gamma)$. Here $\mathcal{S}$ and $\mathcal{A}$ are the sets of all possible states and actions, respectively. Meanwhile, $\mathcal{P}$ is the Markov transition kernel and $r$ is the reward function, which is possibly stochastic. Specifically, when taking action $a \in \mathcal{A}$ at state $s \in \mathcal{S}$, the agent receives reward $r(s,a)$ and the environment transits into a new state according to transition probability $\mathcal{P}(\cdot \,|\, s, a)$. Meanwhile, $\zeta$ is the distribution of initial state $S_0 \in \mathcal{S}$ and $\gamma \in (0,1)$ is the discount factor. In addition, policy $\pi(a \,|\, s)$ gives the probability of taking action $a$ at state $s$. We denote the state- and action-value functions associated with $\pi$ by $V^\pi \colon \mathcal{S} \to \mathbb{R}$ and $Q^\pi \colon \mathcal{S} \times \mathcal{A} \to \mathbb{R}$, which are defined respectively as

$$V^\pi(s) = (1 - \gamma) \cdot \mathbb{E}\left[\sum_{t=0}^{\infty} \gamma^t \cdot r(S_t, A_t) \,\Big|\, S_0 = s\right], \quad \forall s \in \mathcal{S}, \tag{2.1}$$

$$Q^\pi(s,a) = (1 - \gamma) \cdot \mathbb{E}\left[\sum_{t=0}^{\infty} \gamma^t \cdot r(S_t, A_t) \,\Big|\, S_0 = s, A_0 = a\right], \quad \forall (s,a) \in \mathcal{S} \times \mathcal{A}, \tag{2.2}$$

where $S_0 \sim \zeta(\cdot)$, $A_t \sim \pi(\cdot \,|\, S_t)$, and $S_{t+1} \sim \mathcal{P}(\cdot \,|\, S_t, A_t)$ for all $t \geq 0$. Also, we define the advantage function of policy $\pi$ as the difference between $Q^\pi$ and $V^\pi$, i.e., $A^\pi(s,a) = Q^\pi(s,a) - V^\pi(s)$. By the definitions in (2.1) and (2.2), $V^\pi$ and $Q^\pi$ are related via

$$V^\pi(s) = \mathbb{E}_\pi\left[Q^\pi(s,a)\right] = \langle Q^\pi(s,\cdot), \pi(\cdot \,|\, s)\rangle,$$

where $\langle \cdot, \cdot \rangle$ is the inner product in $\mathbb{R}^{|\mathcal{A}|}$. Here we write $\mathbb{E}_{a \sim \pi(\cdot \,|\, s)}[Q^\pi(s,a)]$ as $\mathbb{E}_\pi[Q^\pi(s,a)]$ for notational simplicity. Note that policy $\pi$ together with transition probability $\mathcal{P}$ induces a Markov

chain over state space $\mathcal{S}$. We denote by $\varrho_\pi$ the stationary state distribution of the Markov chain induced by $\pi$. We further define $\varsigma_\pi(s,a) = \pi(a \mid s) \cdot \varrho_\pi(s)$ as the stationary state-action distribution over $\mathcal{S} \times \mathcal{A}$. Meanwhile, policy $\pi$ induces a state visitation measure over $\mathcal{S}$ and a state-action visitation measure over $\mathcal{S} \times \mathcal{A}$, which are denoted by $\nu_\pi$ and $\sigma_\pi$, respectively. Specifically, for all $(s,a) \in \mathcal{S} \times \mathcal{A}$, we define

$$\nu_\pi(s) = (1-\gamma) \cdot \sum_{t=0}^\infty \gamma^t \cdot \mathbb{P}(S_t = s), \qquad \sigma_\pi(s,a) = (1-\gamma) \cdot \sum_{t=0}^\infty \gamma^t \cdot \mathbb{P}(S_t = s, A_t = a), \tag{2.3}$$

where $S_0 \sim \zeta(\cdot)$, $A_t \sim \pi(\cdot \mid S_t)$, and $S_{t+1} \sim \mathcal{P}(\cdot \mid S_t, A_t)$ for all $t \geq 0$. By definition, we have $\sigma_\pi(s,a) = \pi(a \mid s) \cdot \nu_\pi(s)$. We define the expected total reward function $J(\pi)$ by

$$J(\pi) = (1-\gamma) \cdot \mathbb{E}\left[\sum_{t=0}^\infty \gamma^t \cdot r(S_t, A_t)\right] = \mathbb{E}_\zeta\left[V^\pi(s)\right] = \mathbb{E}_{\sigma_\pi}\left[r(s,a)\right], \quad \forall \pi, \tag{2.4}$$

where we write $\mathbb{E}_{\sigma_\pi}[r(s,a)] = \mathbb{E}_{(s,a)\sim\sigma_\pi(\cdot,\cdot)}[r(s,a)]$ for notational simplicity. The goal of reinforcement learning is to find the optimal policy that maximizes $J(\pi)$, which is denoted by $\pi^*$. When state space $\mathcal{S}$ is large, a popular approach is to find the maximizer of $J(\pi)$ over a class of parameterized policies $\{\pi_\theta : \theta \in \mathcal{B}\}$, where $\theta \in \mathcal{B}$ is the parameter and $\mathcal{B}$ is the parameter space. In this case, we obtain the optimization problem $\max_{\theta \in \mathcal{B}} J(\pi_\theta)$.

**Policy Gradient Methods.** Policy gradient methods maximize $J(\pi_\theta)$ using $\nabla_\theta J(\pi_\theta)$. These methods are based on the policy gradient theorem (Sutton and Barto, 2018), which states that

$$\nabla_\theta J(\pi_\theta) = \mathbb{E}_{\sigma_{\pi_\theta}}\left[Q^{\pi_\theta}(s,a) \cdot \nabla_\theta \log \pi_\theta(a \mid s)\right], \tag{2.5}$$

where $\sigma_{\pi_\theta}$ is the state-action visitation measure defined in (2.3). Based on (2.5), (vanilla) policy gradient maximizes the expected total reward via gradient ascent. Specifically, we generate a sequence of policy parameters $\{\theta_i\}_{i \in [T]}$ via

$$\theta_{i+1} \leftarrow \theta_i + \eta \cdot \nabla_\theta J(\pi_{\theta_i}), \tag{2.6}$$

where $\eta > 0$ is the learning rate. Meanwhile, natural policy gradient (Kakade, 2002) utilizes natural gradient ascent (Amari, 1998), which is invariant to the parameterization of policies. Specifically, let $F(\theta)$ be the Fisher information matrix corresponding to policy $\pi_\theta$, which is given by

$$F(\theta) = \mathbb{E}_{\sigma_{\pi_\theta}}\left[\nabla_\theta \log \pi_\theta(a \mid s)\left[\nabla_\theta \log \pi_\theta(a \mid s)\right]^\top\right]. \tag{2.7}$$

At each iteration, natural policy gradient performs

$$\theta_{i+1} \leftarrow \theta_i + \eta \cdot F^{-1}(\theta_i) \cdot \nabla_\theta J(\pi_{\theta_i}), \tag{2.8}$$

where $F^{-1}(\theta_i)$ is the inverse of $F(\theta_i)$ and $\eta$ is the learning rate. In practice, both $Q^{\pi_\theta}$ in (2.5) and $F(\theta)$ in (2.7) remain to be estimated, which yields approximations of the policy improvement steps in (2.6) and (2.8).

## 3 NEURAL POLICY GRADIENT METHODS

In this section, we represent $\pi_\theta$ by a two-layer neural network and study neural policy gradient methods, which estimate the policy gradient and natural policy gradient using the actor-critic scheme (Konda and Tsitsiklis, 2000).

### 3.1 OVERPARAMETERIZED NEURAL POLICY

We now introduce the parameterization of policies. For notational simplicity, we assume that $\mathcal{S} \times \mathcal{A} \subseteq \mathbb{R}^d$ with $d \geq 2$. Without loss of generality, we further assume that $\|(s,a)\|_2 = 1$ for all $(s,a) \in \mathcal{S} \times \mathcal{A}$. A two-layer neural network $f((s,a); W, b)$ with input $(s,a)$ and width $m$ takes the form of

$$f\left((s,a); W, b\right) = \frac{1}{\sqrt{m}} \sum_{r=1}^m b_r \cdot \text{ReLU}\left((s,a)^\top [W]_r\right), \quad \forall (s,a) \in \mathcal{S} \times \mathcal{A}. \tag{3.1}$$

Here $\text{ReLU}\colon \mathbb{R} \to \mathbb{R}$ is the rectified linear unit (ReLU) activation function, which is defined as $\text{ReLU}(u) = \mathbb{1}\{u > 0\} \cdot u$. Also, $\{b_r\}_{r \in [m]}$ and $W = ([W]_1^\top, \ldots, [W]_m^\top)^\top \in \mathbb{R}^{md}$ in (3.1) are the parameters. When training the two-layer neural network, we initialize the parameters via $[W_{\text{init}}]_r \sim N(0, I_d/d)$ and $b_r \sim \text{Unif}(\{-1, 1\})$ for all $r \in [m]$. Note that the ReLU activation function satisfies $\text{ReLU}(c \cdot u) = c \cdot \text{ReLU}(u)$ for all $c > 0$ and $u \in \mathbb{R}$. Hence, without loss of generality, we keep $b_r$ fixed at the initial parameter throughout training and only update $W$ in the sequel. See, e.g., Allen-Zhu et al. (2018b) for a detailed argument. For notational simplicity, we write $f((s,a); W, b)$ as $f((s,a); W)$ hereafter.

Using the two-layer neural network in (3.1), we define

$$\pi_\theta(a \,|\, s) = \frac{\exp\bigl[\tau \cdot f\bigl((s,a); \theta\bigr)\bigr]}{\sum_{a' \in \mathcal{A}} \exp\bigl[\tau \cdot f\bigl((s,a'); \theta\bigr)\bigr]}, \quad \forall (s,a) \in \mathcal{S} \times \mathcal{A}, \tag{3.2}$$

where $f((\cdot, \cdot); \theta)$ is defined in (3.1) with $\theta \in \mathbb{R}^{md}$ playing the role of $W$. Note that $\pi_\theta$ defined in (3.2) takes the form of an energy-based policy (Haarnoja et al., 2017). With a slight abuse of terminology, we call $\tau$ the temperature parameter and $f((\cdot, \cdot); \theta)$ the energy function in the sequel.

In the sequel, we investigate policy gradient methods for the class of neural policies defined in (3.2). We define the feature mapping $\phi_\theta = ([\phi_\theta]_1^\top, \ldots, [\phi_\theta]_m^\top)^\top \colon \mathbb{R}^d \to \mathbb{R}^{md}$ of a two-layer neural network $f((\cdot, \cdot); \theta)$ as

$$[\phi_\theta]_r(s,a) = \frac{b_r}{\sqrt{m}} \cdot \mathbb{1}\bigl\{(s,a)^\top [\theta]_r > 0\bigr\} \cdot (s,a), \quad \forall (s,a) \in \mathcal{S} \times \mathcal{A},\ \forall r \in [m]. \tag{3.3}$$

By (3.1), it holds that $f((\cdot, \cdot); \theta) = \phi_\theta(\cdot, \cdot)^\top \theta$. Meanwhile, $f((\cdot, \cdot); \theta)$ is almost everywhere differentiable with respect to $\theta$, and it holds that $\nabla_\theta f((\cdot, \cdot); \theta) = \phi_\theta(\cdot, \cdot)$. In the following proposition, we calculate the closed forms of the policy gradient $\nabla_\theta J(\pi_\theta)$ and the Fisher information matrix $F(\theta)$ for $\pi_\theta$ defined in (3.2).

**Proposition 3.1** (Policy Gradient and Fisher Information Matrix)**.** For $\pi_\theta$ defined in (3.2), we have

$$\nabla_\theta J(\pi_\theta) = \tau \cdot \mathbb{E}_{\sigma_{\pi_\theta}}\Bigl[Q^{\pi_\theta}(s,a) \cdot \bigl(\phi_\theta(s,a) - \mathbb{E}_{\pi_\theta}\bigl[\phi_\theta(s,a')\bigr]\bigr)\Bigr], \tag{3.4}$$

$$F(\theta) = \tau^2 \cdot \mathbb{E}_{\sigma_{\pi_\theta}}\Bigl[\bigl(\phi_\theta(s,a) - \mathbb{E}_{\pi_\theta}\bigl[\phi_\theta(s,a')\bigr]\bigr)\bigl(\phi_\theta(s,a) - \mathbb{E}_{\pi_\theta}\bigl[\phi_\theta(s,a')\bigr]\bigr)^\top\Bigr], \tag{3.5}$$

where $\phi_\theta(\cdot, \cdot)$ is the feature mapping defined in (3.3), $\tau$ is the temperature parameter, and $\sigma_{\pi_\theta}$ is the state-action visitation measure defined in (2.3). Here we write $\mathbb{E}_{\pi_\theta}[\phi_\theta(s,a')] = \mathbb{E}_{a' \sim \pi_\theta(\cdot \,|\, s)}[\phi_\theta(s,a')]$ for notational simplicity.

*Proof.* See §H.1 for a detailed proof. $\qquad\qquad\qquad\qquad\qquad\qquad\qquad\qquad\qquad\qquad\quad \square$

Since the action-value function $Q^{\pi_\theta}$ in (3.4) is unknown, to obtain the policy gradient, we use another two-layer neural network to track the action-value function of policy $\pi_\theta$. Specifically, we use a two-layer neural network $Q_\omega(\cdot, \cdot) = f((\cdot, \cdot); \omega)$ defined in (3.1) to represent the action-value function $Q^{\pi_\theta}$, where $\omega$ plays the same role as $W$ in (3.1). Such an approach is known as the actor-critic scheme (Konda and Tsitsiklis, 2000). We call $\pi_\theta$ and $Q_\omega$ the actor and critic, respectively. We highlight that in the overparameterized regime where the width of two-layer neural networks $m$ is large, a shared architecture and random initialization between the actor $Q_\omega$ and the energy function of critic $\pi_\theta$ ensures approximate compatible function approximations. See §B for details.

### 3.2 NEURAL POLICY GRADIENT METHODS

Now we present neural policy gradient and neural natural policy gradient. Following the actor-critic scheme, they generate a sequence of policies $\{\pi_{\theta_i}\}_{i \in [T+1]}$ and action-value functions $\{Q_{\omega_i}\}_{i \in [T]}$.

#### 3.2.1 ACTOR UPDATE

As introduced in §2, we aim to solve the optimization problem $\max_{\theta \in \mathcal{B}} J(\pi_\theta)$ iteratively via gradient-based methods, where $\mathcal{B}$ is the parameter space. We set $\mathcal{B} = \{\alpha \in \mathbb{R}^{md} : \|\alpha - W_{\text{init}}\|_2 \leq R\}$, where $R > 1$ and $W_{\text{init}}$ is the initial parameter defined in §3.1. For all $i \in [T]$, let $\theta_i$ be the

policy parameter at the $i$-th iteration. For notational simplicity, in the sequel, we denote by $\sigma_i$ and $\varsigma_i$ the state-action visitation measure $\sigma_{\pi_{\theta_i}}$ and the stationary state-action distribution $\varsigma_{\pi_{\theta_i}}$, respectively, which are defined in §2. Similarly, we write $\nu_i = \nu_{\pi_{\theta_i}}$ and $\varrho_i = \varrho_{\pi_{\theta_i}}$. To update $\theta_i$, we set

$$\theta_{i+1} \leftarrow \Pi_{\mathcal{B}}\big(\theta_i + \eta \cdot G(\theta_i) \cdot \widehat{\nabla}_\theta J(\pi_{\theta_i})\big), \tag{3.6}$$

where we define $\Pi_{\mathcal{B}} \colon \mathbb{R}^{md} \to \mathcal{B}$ as the projection operator onto the parameter space $\mathcal{B} \subseteq \mathbb{R}^{md}$. Here $G(\theta_i) \in \mathbb{R}^{md \times md}$ is a matrix specific to each algorithm. Specifically, we have $G(\theta_i) = I_{md}$ for policy gradient and $G(\theta_i) = (F(\theta_i))^{-1}$ for natural policy gradient, where $F(\theta_i)$ is the Fisher information matrix in (3.5). Meanwhile, $\eta$ is the learning rate and $\widehat{\nabla}_\theta J(\pi_{\theta_i})$ is an estimator of $\nabla_\theta J(\pi_{\theta_i})$, which takes the form of

$$\widehat{\nabla}_\theta J(\pi_{\theta_i}) = \frac{1}{B} \cdot \sum_{\ell=1}^{B} Q_{\omega_i}(s_\ell, a_\ell) \cdot \nabla_\theta \log \pi_{\theta_i}(a_\ell \,|\, s_\ell). \tag{3.7}$$

Here $\tau_i$ is the temperature parameter of $\pi_{\theta_i}$, $\{(s_\ell, a_\ell)\}_{\ell \in [B]}$ is sampled from the state-action visitation measure $\sigma_i$ corresponding to the current policy $\pi_{\theta_i}$, and $B > 0$ is the batch size. Also, $Q_{\omega_i}$ is the critic obtained by Algorithm 2. Here we omit the dependency of $\widehat{\nabla}_\theta J(\pi_{\theta_i})$ on $\omega_i$ for notational simplicity. See §C for the sampling from visitation measures.

**Inverting Fisher Information Matrix.** Recall that $G(\theta_i)$ is the inverse of the Fisher information matrix used in natural policy gradient. In the overparameterized regime, inverting an estimator $\widehat{F}(\theta_i)$ of $F(\theta_i)$ can be infeasible as $\widehat{F}(\theta_i)$ is a high-dimensional matrix, which is possibly not invertible. To resolve this issue, we estimate the natural policy gradient $G(\theta_i) \cdot \nabla_\theta J(\pi_{\theta_i})$ by solving

$$\min_{\alpha \in \mathcal{B}} \|\widehat{F}(\theta_i) \cdot \alpha - \tau_i \cdot \widehat{\nabla}_\theta J(\pi_{\theta_i})\|_2, \tag{3.8}$$

where $\widehat{\nabla}_\theta J(\pi_{\theta_i})$ is defined in (3.7), $\tau_i$ is the temperature parameter in $\pi_{\theta_i}$, and $\mathcal{B}$ is the parameter space. Meanwhile, $\widehat{F}(\theta_i)$ is an unbiased estimator of $F(\theta_i)$ based on $\{(s_\ell, a_\ell)\}_{\ell \in [B]}$ sampled from $\sigma_i$, which is defined as

$$\widehat{F}(\theta_i) = \frac{\tau_i^2}{B} \cdot \sum_{\ell=1}^{B} \Big(\phi_{\theta_i}(s_\ell, a_\ell) - \mathbb{E}_{\pi_{\theta_i}}\big[\phi_{\theta_i}(s_\ell, a_\ell')\big]\Big)\Big(\phi_{\theta_i}(s_\ell, a_\ell) - \mathbb{E}_{\pi_{\theta_i}}\big[\phi_{\theta_i}(s_\ell, a_\ell')\big]\Big)^\top, \tag{3.9}$$

where $a_\ell' \sim \pi_{\theta_i}(\cdot \,|\, s_\ell)$ and $\phi_{\theta_i}$ is defined in (3.3) with $\theta = \theta_i$. The actor update of neural natural policy gradient takes the form of

$$\tau_{i+1} \leftarrow \tau_i + \eta, \qquad \tau_{i+1} \cdot \theta_{i+1} \leftarrow \tau_i \cdot \theta_i + \eta \cdot \operatorname*{argmin}_{\alpha \in \mathcal{B}} \|\widehat{F}(\theta_i) \cdot \alpha - \tau_i \cdot \widehat{\nabla}_\theta J(\pi_{\theta_i})\|_2, \tag{3.10}$$

where we use an arbitrary minimizer of (3.8) if it is not unique. Note that we also update the temperature parameter by $\tau_{i+1} \leftarrow \tau_i + \eta$, which ensures $\theta_{i+1} \in \mathcal{B}$. It is worth mentioning that up to minor modifications, our analysis allows for approximately solving (3.8), which is the common practice of approximate second-order optimization (Martens and Grosse, 2015; Wu et al., 2017).

To summarize, at the $i$-th iteration, neural policy gradient obtains $\theta_{i+1}$ via projected gradient ascent using $\widehat{\nabla}_\theta J(\pi_{\theta_i})$ defined in (3.7). Meanwhile, neural natural policy gradient solves (3.8) and obtains $\theta_{i+1}$ according to (3.10).

### 3.2.2 CRITIC UPDATE

To obtain $\widehat{\nabla}_\theta J(\pi_\theta)$, it remains to obtain the critic $Q_{\omega_i}$ in (3.7). For any policy $\pi$, the action-value function $Q^\pi$ is the unique solution to the Bellman equation $Q = \mathcal{T}^\pi Q$ (Sutton and Barto, 2018). Here $\mathcal{T}^\pi$ is the Bellman operator that takes the form of

$$\mathcal{T}^\pi Q(s, a) = \mathbb{E}\big[(1 - \gamma) \cdot r(s, a) + \gamma \cdot Q(s', a')\big], \quad \forall (s, a) \in \mathcal{S} \times \mathcal{A},$$

where $s' \sim \mathcal{P}(\cdot \,|\, s, a)$ and $a' \sim \pi(\cdot \,|\, s')$. Correspondingly, we aim to solve the following optimization problem

$$\omega_i \leftarrow \operatorname*{argmin}_{\omega \in \mathcal{B}} \mathbb{E}_{\varsigma_i}\Big[\big(Q_\omega(s, a) - \mathcal{T}^{\pi_{\theta_i}} Q_\omega(s, a)\big)^2\Big], \tag{3.11}$$

where $\varsigma_i$ and $\mathcal{T}^{\pi_{\theta_i}}$ are the stationary state-action distribution and the Bellman operator associated with $\pi_{\theta_i}$, respectively, and $\mathcal{B}$ is the parameter space. We adopt neural temporal-difference learning (TD) studied in Cai et al. (2019), which solves the optimization problem in (3.11) via stochastic semigradient descent (Sutton, 1988). Specifically, an iteration of neural TD takes the form of

$$\omega(t+1/2)$$
$$\leftarrow \omega(t) - \eta_{\mathrm{TD}} \cdot \big(Q_{\omega(t)}(s,a) - (1-\gamma) \cdot r(s,a) - \gamma Q_{\omega(t)}(s',a')\big) \cdot \nabla_\omega Q_{\omega(t)}(s,a), \quad (3.12)$$
$$\omega(t+1) \leftarrow \underset{\alpha \in \mathcal{B}}{\mathrm{argmin}} \, \|\alpha - \omega(t+1/2)\|_2, \quad (3.13)$$

where $(s,a) \sim \varsigma_i(\cdot)$, $s' \sim \mathcal{P}(\cdot \,|\, s, a)$, $a' \sim \pi(\cdot \,|\, s')$, and $\eta_{\mathrm{TD}}$ is the learning rate of neural TD. Here (3.12) is the stochastic semigradient step, and (3.13) projects the parameter obtained by (3.12) back to the parameter space $\mathcal{B}$. Meanwhile, the state-action pairs in (3.12) are sampled from the stationary state-action distribution $\varsigma_i$, which is achieved by sampling from the Markov chain induced by $\pi_{\theta_i}$ until it mixes. See Algorithm 2 in §F for details. Finally, combining the actor updates and the critic update described in (3.6), (3.10), and (3.11), respectively, we obtain neural policy gradient and natural policy gradient, which are described in Algorithm 1.

---

**Algorithm 1** Neural Policy Gradient Methods

---

**Require:** Number of iterations $T$, number of TD iterations $T_{\mathrm{TD}}$, learning rate $\eta$, learning rate $\eta_{\mathrm{TD}}$ of neural TD, temperature parameters $\{\tau_i\}_{i \in [T+1]}$, batch size $B$.
1: **Initialization:** Initialize $b_r \sim \mathrm{Unif}(\{-1,1\})$ and $[W_{\mathrm{init}}]_r \sim N(0, I_d/d)$ for all $r \in [m]$. Set $\mathcal{B} \leftarrow \{\alpha \in \mathbb{R}^{md} : \|\alpha - W_{\mathrm{init}}\|_2 \le R\}$ and $\theta_1 \leftarrow W_{\mathrm{init}}$.
2: **for** $i \in [T]$ **do**
3:    Update $\omega_i$ using Algorithm 2 with $\pi_{\theta_i}$ as the input, $\omega(0) \leftarrow W_{\mathrm{init}}$ and $\{b_r\}_{r \in [m]}$ as the initialization, $T_{\mathrm{TD}}$ as the number of iterations, and $\eta_{\mathrm{TD}}$ as the learning rate.
4:    Sample $\{(s_\ell, a_\ell)\}_{\ell \in [B]}$ from the visitation measure $\sigma_i$, and estimate $\widehat{\nabla}_\theta J(\pi_\theta)$ and $\widehat{F}(\theta_i)$ using (3.7) and (3.9), respectively.
5:    If using policy gradient, update $\theta_{i+1}$ by

$$\theta_{i+1} \leftarrow \Pi_{\mathcal{B}}\big(\theta_i + \eta \cdot \widehat{\nabla}_\theta J(\pi_{\theta_i})\big).$$

   If using natural policy gradient, update $\theta_{i+1}$ and $\tau_{i+1}$ by

$$\tau_{i+1} \leftarrow \tau_i + \eta, \qquad \tau_{i+1} \cdot \theta_{i+1} \leftarrow \tau_i \cdot \theta_i + \eta \cdot \underset{\alpha \in \mathcal{B}}{\mathrm{argmin}} \|\widehat{F}(\theta_i) \cdot \alpha - \tau_i \cdot \widehat{\nabla}_\theta J(\pi_{\theta_i})\|_2.$$

6: **end for**
7: **Output:** $\{\pi_{\theta_i}\}_{i \in [T+1]}$.

---

## 4 MAIN RESULTS

In this section, we establish the global optimality and convergence for neural policy gradient methods. Hereafter, we assume that the absolute value of the reward function $r$ is upper bounded by an absolute constant $Q_{\max} > 0$. As a result, we obtain from (2.1) and (2.2) that $|V^\pi(s,a)| \le Q_{\max}$, $|Q^\pi(s,a)| \le Q_{\max}$, and $|A^\pi(s,a)| \le 2Q_{\max}$ for all $\pi$ and $(s,a) \in \mathcal{S} \times \mathcal{A}$. In what follows, we show that neural policy gradient converges to a stationary point of $J(\pi_\theta)$ with respect to $\theta$ at a sublinear rate. We further characterize the geometry of $J(\pi_\theta)$ and establish the global optimality of the obtained stationary point. We defer the global optimality and convergence of neural natural policy gradient to §A.

In the sequel, we study the convergence of neural policy gradient, i.e., Algorithm 1 with (3.6) as the actor update, where $G(\theta) = I_{md}$. In what follows, we lay out a regularity condition on the action-value function $Q^\pi$.

**Assumption 4.1** (Action-Value Function Class). We define

$$\mathcal{F}_{R,\infty} = \left\{ f(s,a) = f_0(s,a) + \int \mathbb{1}\{w^\top(s,a) > 0\} \cdot (s,a)^\top \iota(w) \mathrm{d}\mu(w) : \|\iota(w)\|_\infty \le R/\sqrt{d} \right\},$$

where $\mu \colon \mathbb{R}^d \to \mathbb{R}$ is the density function of the Gaussian distribution $N(0, I_d/d)$, $f_0(\cdot, \cdot) = f((\cdot, \cdot); W_{\text{init}})$ is the two-layer neural network corresponding to the initial parameter $W_{\text{init}}$, and $\iota \colon \mathbb{R}^d \to \mathbb{R}^d$ together with $f_0$ parameterizes the element of $\mathcal{F}_{R,\infty}$. We assume that $Q^\pi \in \mathcal{F}_{R,\infty}$ for all $\pi$.

Assumption 4.1 is a mild regularity condition on $Q^\pi$, as $\mathcal{F}_{R,\infty}$ captures a sufficiently general family of functions, which constitute a subset of the reproducing kernel Hilbert space (RKHS) induced by the random feature $\mathbb{1}\{w^\top(s,a) > 0\} \cdot (s,a)$ with $w \sim N(0, I_d/d)$ (Rahimi and Recht, 2008; 2009) up to the shift of $f_0$. Similar assumptions are imposed in the analysis of batch reinforcement learning in RKHS (Farahmand et al., 2016).

In what follows, we lay out a regularity condition on the state visitation measure $\nu_\pi$ and the stationary state distribution $\varrho_\pi$.

**Assumption 4.2** (Regularity Condition on $\nu_\pi$ and $\varrho_\pi$)**.** Let $\pi$ and $\widetilde{\pi}$ be two arbitrary policies. We assume that there exists an absolute constant $c > 0$ such that

$$\mathbb{E}_{\widetilde{\pi} \cdot \nu_\pi}\Big[\mathbb{1}\big\{|y^\top(s,a)| \leq u\big\}\Big] \leq c \cdot u/\|y\|_2,$$

$$\mathbb{E}_{\widetilde{\pi} \cdot \varrho_\pi}\Big[\mathbb{1}\big\{|y^\top(s,a)| \leq u\big\}\Big] \leq c \cdot u/\|y\|_2, \quad \forall y \in \mathbb{R}^d, \ \forall u > 0.$$

Here the expectations are taken over the joint distributions $\widetilde{\pi}(\cdot \mid \cdot) \cdot \nu_\pi(\cdot)$ and $\widetilde{\pi}(\cdot \mid \cdot) \cdot \varrho_\pi(\cdot)$ over $\mathcal{S} \times \mathcal{A}$, respectively.

Assumption 4.2 essentially imposes a regularity condition on the Markov transition kernel $\mathcal{P}$ of the MDP as $\mathcal{P}$ determines $\nu_\pi$ and $\varrho_\pi$ for all $\pi$. Such a regularity condition holds if both $\nu_\pi$ and $\varrho_\pi$ have upper-bounded density functions for all $\pi$.

After introducing these regularity conditions, we present the following proposition adapted from Cai et al. (2019), which characterizes the convergence of neural TD for the critic update.

**Proposition 4.3** (Convergence of Critic Update)**.** We set $\eta_{\text{TD}} = \min\{(1-\gamma)/8, 1/\sqrt{T_{\text{TD}}}\}$ in Algorithm 1. Let $Q_{\omega_i}$ be the output of the $i$-th critic update in Line 3 of Algorithm 1, which is an estimator of $Q^{\pi_{\theta_i}}$ obtained by Algorithm 2 with $T_{\text{TD}}$ iterations. Under Assumptions 4.1 and 4.2, it holds for $T_{\text{TD}} = \Omega(m)$ that

$$\mathbb{E}_{\text{init}}\big[\|Q_{\omega_i} - Q^{\pi_{\theta_i}}\|_{\varsigma_i}^2\big] = \mathcal{O}(R^3 \cdot m^{-1/2} + R^{5/2} \cdot m^{-1/4}), \tag{4.1}$$

where $\varsigma_i$ is the stationary state-action distribution corresponding to $\pi_{\theta_i}$. Here the expectation is taken over the random initialization.

*Proof.* See §F.1 for a detailed proof. $\qquad\square$

Cai et al. (2019) show that the error of the critic update consists of two parts, namely the approximation error of two-layer neural networks and the algorithmic error of neural TD. The former decays as the width $m$ grows, while the latter decays as the number of neural TD iterations $T_{\text{TD}}$ in Algorithm 2 grows. By setting $T_{\text{TD}} = \Omega(m)$, the algorithmic error in (4.1) of Proposition 4.3 is dominated by the approximation error. In contrast with Cai et al. (2019), we obtain a more refined convergence characterization under the more restrictive assumption that $Q^\pi \in \mathcal{F}_{R,\infty}$. Specifically, such a restriction allows us to obtain the upper bound of the mean squared error in (4.1) of Proposition 4.3.

It now remains to establish the convergence of the actor update, which involves the estimator $\widehat{\nabla}_\theta J(\pi_{\theta_i})$ of the policy gradient $\nabla_\theta J(\pi_{\theta_i})$ based on $\{(s_\ell, a_\ell)\}_{\ell \in [B]}$. We introduce the following regularity condition on the variance of $\widehat{\nabla}_\theta J(\pi_{\theta_i})$.

**Assumption 4.4** (Variance Upper Bound)**.** Recall that $\sigma_i$ is the state-action visitation measure corresponding to $\pi_{\theta_i}$ for all $i \in [T]$. Let $\xi_i = \widehat{\nabla}_\theta J(\pi_{\theta_i}) - \mathbb{E}[\widehat{\nabla}_\theta J(\pi_{\theta_i})]$, where $\widehat{\nabla}_\theta J(\pi_{\theta_i})$ is defined in (3.7). We assume that there exists an absolute constant $\sigma_\xi > 0$ such that $\mathbb{E}[\|\xi_i\|_2^2] \leq \tau_i^2 \cdot \sigma_\xi^2/B$ for all $i \in [T]$. Here the expectations are taken over $\sigma_i$ given $\theta_i$ and $\omega_i$.

Assumption 4.4 is a mild regularity condition. Such a regularity condition holds if the Markov chain that generates $\{(s_\ell, a_\ell)\}_{\ell \in [B]}$ mixes sufficiently fast and $Q_{\omega_i}(s,a)$ with $(s,a) \sim \sigma_i$ have upper

bounded second moments for all $i \in [T]$. Zhang et al. (2019) verify that under certain regularity conditions, similar unbiased policy gradient estimators have almost surely upper bounded norms, which implies Assumption 4.4. Similar regularity conditions are also imposed in the analysis of policy gradient methods by Xu et al. (2019a;b).

In what follows, we impose a regularity condition on the discrepancy between the state-action visitation measure and the stationary state-action distribution corresponding to the same policy.

**Assumption 4.5** (Regularity Condition on $\sigma_i$ and $\varsigma_i$). We assume that there exists an absolute constant $\kappa > 0$ such that

$$\left\{ \mathbb{E}_{\varsigma_i} \left[ \left( \frac{\mathrm{d}\sigma_i}{\mathrm{d}\varsigma_i}(s, a) \right)^2 \right] \right\}^{1/2} \leq \kappa, \quad \forall i \in [T]. \tag{4.2}$$

Here $\mathrm{d}\sigma_i / \mathrm{d}\varsigma_i$ is the Radon-Nikodym derivative of $\sigma_i$ with respect to $\varsigma_i$.

We highlight that if the MDP is initialized at the stationary distribution $\varsigma_i$, the state-action visitation measure $\sigma_i$ is the same as $\varsigma_i$. Meanwhile, if the induced Markov state-action chain mixes sufficiently fast, such an assumption also holds. A similar regularity condition is imposed by Scherrer (2013), which assumes that the $L_\infty$-norm of $\mathrm{d}\sigma_i / \mathrm{d}\varsigma_i$ is upper bounded, whereas we only assume that its $L_2$-norm is upper bounded.

Meanwhile, we impose the following regularity condition on the smoothness of the expected total reward $J(\pi_\theta)$ with respect to $\theta$.

**Assumption 4.6** (Lipschitz Continuous Policy Gradient). We assume that $\nabla_\theta J(\pi_\theta)$ is $L$-Lipschitz continuous with respect to $\theta$, where $L > 0$ is an absolute constant.

Such an assumption holds when the transition probability $\mathcal{P}(\cdot \mid s, a)$ and the reward function $r$ are both Lipschitz continuous with respect to their inputs (Pirotta et al., 2015). Also, Karimi et al. (2019); Zhang et al. (2019); Xu et al. (2019b); Agarwal et al. (2019) verify the Lipschitz continuity of the policy gradient under certain regularity conditions.

Note that we restrict $\theta$ to the parameter space $\mathcal{B}$. Here we call $\widehat{\theta} \in \mathcal{B}$ a stationary point of $J(\pi_\theta)$ if it holds for all $\theta \in \mathcal{B}$ that $\nabla_\theta J(\pi_{\widehat{\theta}})^\top (\theta - \widehat{\theta}) \leq 0$. We now show that the sequence $\{\theta_i\}_{i \in [T+1]}$ generated by neural policy gradient converges to a stationary point at a sublinear rate.

**Theorem 4.7** (Convergence to Stationary Point). We set $\tau_i = 1$, $\eta = 1/\sqrt{T}$, $\eta_{\mathrm{TD}} = \min\{(1 - \gamma)/8, 1/\sqrt{T_{\mathrm{TD}}}\}$, $T_{\mathrm{TD}} = \Omega(m)$, and $\mathcal{B} = \{\alpha : \|\alpha - W_{\mathrm{init}}\|_2 \leq R\}$ by Algorithm 1, where the actor update is given in (3.6) with $G(\theta) = I_{md}$. For all $i \in [T]$, we define

$$\rho_i = \eta^{-1} \cdot \left[ \Pi_{\mathcal{B}} \big( \theta_i + \eta \cdot \nabla_\theta J(\pi_{\theta_i}) \big) - \theta_i \right] \in \mathbb{R}^{md}, \tag{4.3}$$

where $\Pi_{\mathcal{B}} \colon \mathbb{R}^{md} \to \mathcal{B}$ is the projection operator onto $\mathcal{B} \subseteq \mathbb{R}^{md}$. Under the assumptions of Proposition 4.3 and Assumptions 4.4-4.6, for $T \geq 4L^2$ we have

$$\min_{i \in [T]} \mathbb{E}\big[\|\rho_i\|_2^2\big] \leq 8/\sqrt{T} \cdot \mathbb{E}\big[ J(\pi_{\theta_{T+1}}) - J(\pi_{\theta_1}) \big] + 8\sigma_\xi^2/B + \varepsilon_Q(T),$$

where $\kappa$ is defined in (4.2) of Assumption 4.2 and $\varepsilon_Q(T) = \kappa \cdot \mathcal{O}(R^{5/2} \cdot m^{-1/4} \cdot T^{1/2} + R^{9/4} \cdot m^{-1/8} \cdot T^{1/2})$. Here the expectations are taken over all the randomness.

*Proof.* See §D.1 for a detailed proof. □

By Theorem 4.7 with $m = \Omega(T^8 \cdot R^{18})$ and $B = \Omega(\sqrt{T})$, we obtain $\min_{i \in [T]} \mathbb{E}[\|\rho_i\|_2^2] = \mathcal{O}(1/\sqrt{T})$. Therefore, when the two-layer neural networks are sufficiently wide and the batch size $B$ is sufficiently large, neural policy gradient achieves a $1/\sqrt{T}$-rate of convergence. Moreover, $\rho_i$ defined in (4.3) is known as the gradient mapping at $\theta_i$ (Nesterov, 2018). It is known that $\widehat{\theta} \in \mathcal{B}$ is a stationary point if and only if the gradient mapping at $\widehat{\theta}$ is a zero vector. Therefore, (a subsequence of) $\{\theta_i\}_{i \in [T+1]}$ converges to a stationary point $\widehat{\theta} \in \mathcal{B}$ as $\min_{i \in [T]} \mathbb{E}[\|\rho_i\|_2^2]$ converges to zero. In other words, neural policy gradient converges to a stationary point at a $1/\sqrt{T}$-rate. Also, we remark

that the projection operator in the actor update is adopted only for the purpose of simplicity, which can be removed with more refined analysis. Moreover, the projection-free version of neural policy gradient converges to a stationary point at a similar sublinear rate. See §G for details.

We now characterize the global optimality of the obtained stationary point $\widehat{\theta}$. To this end, we compare the expected total reward of $\pi_{\widehat{\theta}}$ with that of the global optimum $\pi^*$ of $J(\pi)$.

**Theorem 4.8** (Global Optimality of Stationary Point). Let $\widehat{\theta} \in \mathcal{B}$ be a stationary point of $J(\pi_\theta)$. It holds that

$$(1 - \gamma) \cdot \big(J(\pi^*) - J(\pi_{\widehat{\theta}})\big) \leq 2Q_{\max} \cdot \inf_{\theta \in \mathcal{B}} \|u_{\widehat{\theta}}(\cdot, \cdot) - \phi_{\widehat{\theta}}(\cdot, \cdot)^\top \theta\|_{\sigma_{\pi_{\widehat{\theta}}}},$$

where $Q_{\max}$ is the upper bound of $|r|$ and $u_{\widehat{\theta}} \colon \mathcal{S} \times \mathcal{A} \to \mathbb{R}$ is defined as

$$u_{\widehat{\theta}}(s, a) = \frac{\mathrm{d}\sigma_{\pi^*}}{\mathrm{d}\sigma_{\pi_{\widehat{\theta}}}}(s, a) - \frac{\mathrm{d}\nu_{\pi^*}}{\mathrm{d}\nu_{\pi_{\widehat{\theta}}}}(s) + \phi_{\widehat{\theta}}(s, a)^\top \widehat{\theta}, \quad \forall (s, a) \in \mathcal{S} \times \mathcal{A}. \tag{4.4}$$

Here $\mathrm{d}\sigma_{\pi^*}/\mathrm{d}\sigma_{\pi_{\widehat{\theta}}}$ and $\mathrm{d}\nu_{\pi^*}/\mathrm{d}\nu_{\pi_{\widehat{\theta}}}$ are the Radon-Nikodym derivatives, and $\|\cdot\|_{\sigma_{\pi_{\widehat{\theta}}}}$ is the $L_2(\sigma_{\pi_{\widehat{\theta}}})$-norm.

*Proof.* See §D.2 for a detailed proof. $\qquad\square$

To understand Theorem 4.8, we highlight that for $\theta, \widehat{\theta} \in \mathcal{B}$, the function $\phi_{\widehat{\theta}}(\cdot, \cdot)^\top \theta$ is well approximated by the overparameterized two-layer neural network $f((\cdot, \cdot); \theta)$. See Corollary E.4 for details. Therefore, the global optimality of $\pi_{\widehat{\theta}}$ depends on the error of approximating $u_{\widehat{\theta}}$ with an overparameterized two-layer neural network. Specifically, if $u_{\widehat{\theta}}$ is well approximated by an overparameterized two-layer neural network, then $\pi_{\widehat{\theta}}$ is nearly as optimal as $\pi^*$. In the following corollary, we formally establish a sufficient condition for any stationary point $\widehat{\theta}$ to be globally optimal.

**Theorem 4.9** (Global Optimality of Stationary Point). Let $\widehat{\theta} \in \mathcal{B}$ be a stationary point of $J(\pi_\theta)$. We assume that $u_{\widehat{\theta}} \in \mathcal{F}_{R,\infty}$ in Theorem 4.8. Under Assumption 4.2, it holds that

$$(1 - \gamma) \cdot \mathbb{E}_{\text{init}}\big[J(\pi^*) - J(\pi_{\widehat{\theta}})\big] = \mathcal{O}(R^{3/2} \cdot m^{-1/4}).$$

More generally, without assuming $u_{\widehat{\theta}} \in \mathcal{F}_{R,\infty}$ in Theorem 4.8, under Assumption 4.2, it holds that

$$(1 - \gamma) \cdot \mathbb{E}_{\text{init}}\big[J(\pi^*) - J(\pi_{\widehat{\theta}})\big] = \mathcal{O}(R^{3/2} \cdot m^{-1/4}) + \mathbb{E}_{\text{init}}\big[\|\Pi_{\mathcal{F}_{R,\infty}} u_{\widehat{\theta}} - u_{\widehat{\theta}}\|_{\sigma_{\pi_{\widehat{\theta}}}}\big].$$

Here the expectations are taken over the random initialization, and $\Pi_{\mathcal{F}_{R,\infty}}$ is the projection operator onto $\mathcal{F}_{R,\infty}$ with respect to the $L_2(\sigma_{\pi_{\widehat{\theta}}})$-norm.

*Proof.* See §H.2 for a detailed proof. $\qquad\square$

By Theorem 4.9, a stationary point $\widehat{\theta}$ is globally optimal if $u_{\widehat{\theta}} \in \mathcal{F}_{R,\infty}$ and $m \to \infty$. Moreover, following from the definition of $\rho_i$ in (4.3) of Theorem 4.7, we obtain that

$$\nabla_\theta J(\pi_{\theta_i})^\top (\theta - \theta_i) \leq (2R + 2\eta \cdot Q_{\max}) \cdot \|\rho_i\|_2, \quad \forall \theta \in \mathcal{B}. \tag{4.5}$$

See §H.3 for a detailed proof of (4.5). Since $\|\rho_i\|_2 = 0$ implies that $\theta_i$ is a stationary point, the right-hand side of (4.5) quantifies the deviation of $\theta_i$ from a stationary point $\widehat{\theta}$. Following similar analysis to §D.2 and §H.2, if $u_{\theta_i} \in \mathcal{F}_{R,\infty}$ for all $i \in [T]$, we obtain that

$$(1 - \gamma) \cdot \min_{i \in [T]} \mathbb{E}\big[J(\pi^*) - J(\pi_{\theta_i})\big] = \mathcal{O}(R^{3/2} \cdot m^{-1/4}) + (2R + 2\eta \cdot Q_{\max}) \cdot \min_{i \in [T]} \mathbb{E}\big[\|\rho_i\|_2\big].$$

Thus, by invoking Theorem 4.7, it holds for sufficiently large $m$ and $B$ that the expected total reward $J(\pi_{\theta_i})$ converges to the global optimum $J(\pi^*)$ at a $1/T^{1/4}$-rate.

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

## A    NEURAL NATURAL POLICY GRADIENT

In the sequel, we study the convergence of neural natural policy gradient. As shown in Algorithm 1, neural natural policy gradient uses neural TD for policy evaluation and updates the actor using (3.10), where $\theta_i$ and $\tau_i$ in (3.2) are both updated. To analyze the critic update, we impose Assumptions 4.1 and 4.2, which guarantee that Proposition 4.3 holds. Meanwhile, to analyze the actor update, we impose the following regularity conditions.

In parallel to Assumption 4.4, we lay out the following regularity condition on the variance of the estimators of the policy gradient and the Fisher information matrix.

**Assumption A.1** (Variance Upper Bound). Let $\mathcal{B} = \{\alpha \in \mathbb{R}^{md} : \|\alpha - W_{\mathrm{init}}\|_2 \leq R\}$, where $W_{\mathrm{init}}$ is the initial parameter. We define

$$\delta_i = (\tau_{i+1} \cdot \theta_{i+1} - \tau_i \cdot \theta_i)/\eta = \operatorname*{argmin}_{\alpha \in \mathcal{B}} \|\widehat{F}(\theta_i) \cdot \alpha - \tau_i \cdot \widehat{\nabla}_\theta J(\pi_{\theta_i})\|_2, \quad \forall i \in [T],$$

where $\widehat{\nabla}_\theta J(\pi_{\theta_i})$ and $\widehat{F}(\theta_i)$ are defined in (3.7) and (3.9), respectively. With slight abuse of notation, for all $i \in [T]$, we define the function $\xi_i : \mathbb{R}^{md} \to \mathbb{R}^{md}$ as

$$\xi_i(\alpha) = \widehat{F}(\theta_i) \cdot \alpha - \tau_i \cdot \widehat{\nabla}_\theta J(\pi_{\theta_i}) - \mathbb{E}\big[\widehat{F}(\theta_i) \cdot \alpha - \tau_i \cdot \widehat{\nabla}_\theta J(\pi_{\theta_i})\big].$$

We assume that there exists an absolute constant $\sigma_\xi > 0$ such that

$$\mathbb{E}\big[\|\xi_i(\delta_i)\|_2^2\big] \leq \tau_i^4 \cdot \sigma_\xi^2/B, \quad \mathbb{E}\big[\|\xi_i(\omega_i)\|_2^2\big] \leq \tau_i^4 \cdot \sigma_\xi^2/B, \quad \forall i \in [T].$$

Here the expectations are taken over $\sigma_i$ given $\theta_i$ and $\omega_i$.

Next, we lay out a regularity condition on the visitation measures $\sigma_i$, $\nu_i$ and the stationary distributions $\varsigma_i$, $\varrho_i$, respectively.

**Assumption A.2** (Upper Bounded Concentrability Coefficient). We denote by $\nu_*$ and $\sigma_*$ the state and state-action visitation measures corresponding to the global optimum $\pi^*$. For all $i \in [T]$, we define the concentrability coefficients $\varphi_i$, $\psi_i$, $\varphi_i'$, and $\psi_i'$ as

$$\varphi_i = \left\{ \mathbb{E}_{\sigma_i}\big[(\mathrm{d}\sigma_*/\mathrm{d}\sigma_i)^2\big] \right\}^{1/2}, \quad \psi_i = \left\{ \mathbb{E}_{\nu_i}\big[(\mathrm{d}\nu_*/\mathrm{d}\nu_i)^2\big] \right\}^{1/2},$$

$$\varphi_i' = \left\{ \mathbb{E}_{\varsigma_i}\big[(\mathrm{d}\sigma_*/\mathrm{d}\varsigma_i)^2\big] \right\}^{1/2}, \quad \psi_i' = \left\{ \mathbb{E}_{\varrho_i}\big[(\mathrm{d}\nu_*/\mathrm{d}\varrho_i)^2\big] \right\}^{1/2}, \tag{A.1}$$

where $\mathrm{d}\sigma_*/\mathrm{d}\sigma_i$, $\mathrm{d}\nu_*/\mathrm{d}\nu_i$, $\mathrm{d}\sigma_*/\mathrm{d}\varsigma_i$, and $\mathrm{d}\nu_*/\mathrm{d}\varrho_i$ are the Radon-Nikodym derivatives. We assume that the concentrability coefficients defined in (A.1) are uniformly upper bounded by an absolute constant $c_0 > 0$.

The regularity condition on upper bounded concentrability coefficients is commonly imposed in the reinforcement learning literature and is standard for theoretical analysis (Szepesvári and Munos, 2005; Munos and Szepesvári, 2008; Antos et al., 2008; Lazaric et al., 2016; Farahmand et al., 2010; 2016; Scherrer, 2013; Scherrer et al., 2015; Yang et al., 2019b; Chen and Jiang, 2019).

Finally, we introduce the following regularity condition on the initial parameter $W_{\mathrm{init}}$ in Algorithm 1.

**Assumption A.3** (Upper Bounded Moment at Random Initialization). Let $\phi_0(s, a) \in \mathbb{R}^{md}$ be the feature mapping defined in (3.3) with $\theta = W_{\mathrm{init}}$. We assume that there exists an absolute constant $M > 0$ such that

$$\mathbb{E}_{\mathrm{init}}\left[ \sup_{(s,a) \in \mathcal{S} \times \mathcal{A}} \left|f\big((s,a); W_{\mathrm{init}}\big)\right|^2 \right] = \mathbb{E}_{\mathrm{init}}\left[ \sup_{(s,a) \in \mathcal{S} \times \mathcal{A}} |\phi_0(s,a)^\top W_{\mathrm{init}}|^2 \right] \leq M^2.$$

Here the expectations are taken over the random initialization.

Note that as $m \to \infty$, the two-layer neural network $\phi_0(s, a)^\top W_{\mathrm{init}}$ converges to a Gaussian process indexed by $(s, a)$ (Lee et al., 2018), which lies in a compact subset of $\mathbb{R}^d$. It is known that under certain regularity conditions, the maximum of a Gaussian process over a compact index set is a sub-Gaussian random variable (van Handel, 2014). Therefore, the regularity condition that $\max_{(s,a)} |\phi_0(s,a)^\top W_{\mathrm{init}}|$ has a finite second moment is mild.

We now establish the global optimality and rate of convergence of neural natural policy gradient.

**Theorem A.4** (Global Optimality and Convergence). We set $\eta = 1/\sqrt{T}$, $\eta_{\mathrm{TD}} = \min\{(1 - \gamma)/8, 1/\sqrt{T_{\mathrm{TD}}}\}$, $T_{\mathrm{TD}} = \Omega(m)$, $\tau_i = (i - 1) \cdot \eta$, and $\mathcal{B} = \{\alpha : \|\alpha - W_{\mathrm{init}}\|_2 \leq R\}$ in Algorithm 1, where the actor update is given in (3.10). Under the assumptions of Proposition 4.3 and Assumptions A.1-A.3, we have

$$\min_{i \in [T]} \mathbb{E}\big[J(\pi^*) - J(\pi_{\theta_i})\big] \leq \frac{\log |\mathcal{A}| + 9R^2 + M}{(1 - \gamma) \cdot \sqrt{T}} + \frac{1}{(1 - \gamma) \cdot T} \cdot \sum_{i=1}^{T} \bar{\epsilon}_i(T). \tag{A.2}$$

Here $M$ is defined in Assumption A.3 and $\bar{\epsilon}_i(T)$ satisfies

$$\bar{\epsilon}_i(T) = \underbrace{\sqrt{8}c_0 \cdot R^{1/2} \cdot (\sigma_\xi^2/B)^{1/4}}_{(a)} \tag{A.3}$$
$$+ \underbrace{\mathcal{O}\big((\tau_{i+1} \cdot T^{1/2} + 1) \cdot R^{3/2} \cdot m^{-1/4} + R^{5/4} \cdot m^{-1/8}\big)}_{(b)} + \underbrace{\varepsilon_{Q,i}}_{(c)},$$

where $c_0$ is defined in Assumption A.2 and $\varepsilon_{Q,i} = c_0 \cdot \mathcal{O}(R^{3/2} \cdot m^{-1/4} + R^{5/4} \cdot m^{-1/8})$. Here the expectation is taken over all the randomness.

*Proof.* See §D.3 for a detailed proof. □

As shown in (A.2) of Theorem A.4, the optimality gap $\min_{i \in [T]} \mathbb{E}[J(\pi^*) - J(\pi_{\theta_i})]$ is upper bounded by two terms. Intuitively, the first $\mathcal{O}(1/\sqrt{T})$ term characterizes the convergence of neural natural policy gradient as $m, B \to \infty$. Meanwhile, the second term aggregates the errors incurred by both the actor update and the critic update due to finite $m$ and $B$. Specifically, in (A.3) of Theorem A.4, (a) corresponds to the estimation error of $\widehat{F}(\theta)$ and $\widehat{\nabla}_\theta J(\pi_\theta)$ due to the finite batch size $B$, which vanishes as $B \to \infty$. Also, (b) corresponds to the incompatibility between the parameterizations of the actor and critic. As introduced in §3.1, we use shared architecture and random initialization to ensure approximately compatible function approximations. In particular, (b) vanishes as $m \to \infty$. Meanwhile, (c) corresponds to the policy evaluation error, i.e., the error of approximating $Q^{\pi_{\theta_i}}$ using $Q_{\omega_i}$. As shown in Proposition 4.3, such an error is sufficiently small when both $m$ and $T_{\mathrm{TD}}$ are sufficiently large. To conclude, when $m$, $B$, and $T_{\mathrm{TD}}$ are sufficiently large, the expected total reward of (a subsequence of) $\{\pi_{\theta_i}\}_{i \in [T+1]}$ obtained from the neural natural policy gradient converges to the global optimum $J(\pi^*)$ at a $1/\sqrt{T}$-rate. Formally, we have the following corollary.

**Corollary A.5** (Global Optimality and Convergence). Under the same assumptions of Theorem A.4, it holds for $m = \Omega(R^{10} \cdot T^6)$ and $B = \Omega(R^2 \cdot T^2 \cdot \sigma_\xi^2)$ that

$$\min_{i \in [T]} \mathbb{E}\big[J(\pi^*) - J(\pi_{\theta_i})\big] = \mathcal{O}\left(\frac{\log |\mathcal{A}|}{(1 - \gamma) \cdot \sqrt{T}}\right).$$

Here the expectation is taken over all the randomness.

*Proof.* See §H.4 for a detailed proof. □

Corollary A.5 establishes both the global optimality and rate of convergence of neural natural policy gradient. Combining Theorem 4.7 and Corollary A.5, we conclude that when we use overparameterized two-layer neural networks, both neural policy gradient and neural natural policy gradient converge at $1/\sqrt{T}$-rates. In comparison, when $m$ and $B$ are sufficiently large, neural policy gradient is only shown to converge to a stationary point under the additional regularity condition that $\nabla_\theta J(\pi_\theta)$ is Lipschitz continuous (Assumption 4.6). Moreover, by Theorem 4.8, the global optimality of such a stationary point hinges on the representation power of the overparameterized two-layer neural network. In contrast, neural natural policy gradient is shown to attain the global optimum when both $m$ and $B$ are sufficiently large without additional regularity conditions such as Assumption 4.6, which reveals the benefit of incorporating more sophisticated optimization techniques to reinforcement learning. A similar phenomenon is observed in the LQR setting (Fazel et al., 2018; Malik et al., 2018; Tu and Recht, 2018), where natural policy gradient enjoys an improved rate of convergence.

In recent work, Liu et al. (2019) study the global optimality and rates of convergence of neural proximal policy optimization (PPO) and trust region policy optimization (TRPO) (Schulman et al., 2015; 2017). Although Liu et al. (2019) establish a similar $1/\sqrt{T}$-rate of convergence to the global optimum, neural PPO is different from neural natural policy gradient, as it requires solving a sub-problem of policy improvement in the functional space by fitting an overparameterized two-layer neural network using multiple stochastic gradient steps in the parameter space. In contrast, neural natural policy gradient only requires a single stochastic natural gradient step in the parameter space, which makes the analysis even more challenging.

## B    SHARED INITIALIZATION AND COMPATIBLE FUNCTION APPROXIMATION.

Sutton et al. (2000) introduce the notion of compatible function approximations. Specifically, the action-value function $Q_\omega$ is compatible with $\pi_\theta$ if we have $\nabla_\omega A_\omega(s, a) = \nabla_\theta \log \pi_\theta(a \,|\, s)$ for all $(s, a) \in \mathcal{S} \times \mathcal{A}$, where $A_\omega(s, a) = Q_\omega(s, a) - \langle Q_\omega(s, \cdot), \pi_\theta(\cdot \,|\, s) \rangle$ is the advantage function corresponding to $Q_\omega$. Compatible function approximations enable us to construct unbiased estimators of the policy gradient, which are essential for the optimality and convergence of policy gradient methods (Konda and Tsitsiklis, 2000; Sutton et al., 2000; Kakade, 2002; Peters and Schaal, 2008; Wagner, 2011; 2013).

To approximately obtain compatible function approximations when both the actor and critic are represented by neural networks, we use a shared architecture between the action-value function $Q_\omega$ and the energy function of $\pi_\theta$, and initialize $Q_\omega$ and $\pi_\theta$ with the same parameter $W_{\text{init}}$, where $[W_{\text{init}}]_r \sim N(0, I_d/d)$ for all $r \in [m]$. We show that in the overparameterized regime where $m$ is large, the shared architecture and random initialization ensure $Q_\omega$ to be approximately compatible with $\pi_\theta$ in the following sense. We define $\overline{\phi}_0 = ([\overline{\phi}_0]_1^\top, \ldots, [\overline{\phi}_0]_m^\top)^\top : \mathbb{R}^d \to \mathbb{R}^{md}$ as the centered feature mapping corresponding to the initialization, which takes the form of

$$[\overline{\phi}_0]_r(s, a) = \frac{b_r}{\sqrt{m}} \cdot \mathbb{1}\big\{(s, a)^\top [W_{\text{init}}]_r > 0\big\} \cdot (s, a) \tag{B.1}$$

$$- \mathbb{E}_{\pi_\theta}\left[\frac{b_r}{\sqrt{m}} \cdot \mathbb{1}\big\{(s, a')^\top [W_{\text{init}}]_r > 0\big\} \cdot (s, a')\right], \quad \forall (s, a) \in \mathcal{S} \times \mathcal{A},$$

where $W_{\text{init}}$ is the initialization shared by both the actor and critic, and we omit the dependency on $\theta$ for notational simplicity. Similarly, we define for all $(s, a) \in \mathcal{S} \times \mathcal{A}$ the following centered feature mappings,

$$\overline{\phi}_\theta(s, a) = \phi_\theta(s, a) - \mathbb{E}_{\pi_\theta}\big[\phi_\theta(s, a')\big], \qquad \overline{\phi}_\omega(s, a) = \phi_\omega(s, a) - \mathbb{E}_{\pi_\theta}\big[\phi_\omega(s, a')\big]. \tag{B.2}$$

Here $\phi_\theta(s, a)$ and $\phi_\omega(s, a)$ are the feature mappings defined in (3.3), which correspond to $\theta$ and $\omega$, respectively. By (3.1), we have

$$A_\omega(s, a) = Q_\omega(s, a) - \mathbb{E}_{\pi_\theta}\big[Q_\omega(s, a')\big] = \overline{\phi}_\omega(s, a)^\top \omega, \qquad \nabla_\theta \log \pi_\theta(a \,|\, s) = \overline{\phi}_\theta(s, a), \tag{B.3}$$

which holds almost everywhere for $\theta \in \mathbb{R}^{md}$. As shown in Corollary E.3 in §E, when the width $m$ is sufficiently large, in policy gradient methods, both $\overline{\phi}_\theta$ and $\overline{\phi}_\omega$ are well approximated by $\overline{\phi}_0$ defined in (B.1). Therefore, by (B.3), we conclude that in the overparameterized regime with shared architecture and random initialization, $Q_\omega$ is approximately compatible with $\pi_\theta$.

## C    SAMPLING FROM VISITATION MEASURE.

Recall that the policy gradient $\nabla_\theta J(\pi_\theta)$ in (3.4) involves an expectation taken over the state-action visitation measure $\sigma_{\pi_\theta}$. Thus, to obtain an unbiased estimator of the policy gradient, we need to sample from the visitation measure $\sigma_{\pi_\theta}$. To achieve such a goal, we introduce an artificial MDP $(\mathcal{S}, \mathcal{A}, \widetilde{\mathcal{P}}, \zeta, r, \gamma)$. Such an MDP only differs from the original MDP in the Markov transition kernel $\widetilde{\mathcal{P}}$, which is defined as

$$\widetilde{\mathcal{P}}(s' \,|\, s, a) = \gamma \cdot \mathcal{P}(s' \,|\, s, a) + (1 - \gamma) \cdot \zeta(s'), \quad \forall (s, a, s') \in \mathcal{S} \times \mathcal{A} \times \mathcal{S}.$$

Here $\mathcal{P}$ is the Markov transition kernel of the original MDP. That is, at each state transition of the artificial MDP, the next state is sampled from the initial state distribution $\zeta$ with probability $1 - \gamma$.

In other words, at each state transition, we restart the original MDP with probability $1 - \gamma$. As shown in Konda (2002), the stationary state distribution of the induced Markov chain is exactly the state visitation measure $\nu_{\pi_\theta}$. Therefore, when we sample a trajectory $\{(S_t, A_t)\}_{t \geq 0}$, where $S_0 \sim \zeta(\cdot)$, $A_t \sim \pi(\cdot \mid S_t)$, and $S_{t+1} \sim \widetilde{\mathcal{P}}(\cdot \mid S_t, A_t)$ for all $t \geq 0$, the marginal distribution of $(S_t, A_t)$ converges to the state-action visitation measure $\sigma_{\pi_\theta}$.

## D  PROOF OF MAIN RESULTS

In this section, we present the proof of Theorems 4.7, 4.8, and A.4. Our proof utilizes the following lemma, which establishes the one-point convexity of $J(\pi)$ at the global optimum $\pi^*$. Such a lemma is adapted from Kakade and Langford (2002).

**Lemma D.1** (Performance Difference (Kakade and Langford, 2002))**.** It holds for all $\pi$ that

$$J(\pi^*) - J(\pi) = (1 - \gamma)^{-1} \cdot \mathbb{E}_{\nu_*}\big[\langle Q^\pi(s, \cdot), \pi^*(\cdot \mid s) - \pi(\cdot \mid s)\rangle\big],$$

where $\nu_*$ is the state visitation measure corresponding to $\pi^*$.

*Proof.* Following from Lemma J.1, which is Lemma 6.1 in Kakade and Langford (2002), it holds for all $\pi$ that

$$J(\pi^*) - J(\pi) = (1 - \gamma)^{-1} \cdot \mathbb{E}_{\sigma_*}\big[A^\pi(s, a)\big], \tag{D.1}$$

where $\sigma_*$ is the state-action visitation measure corresponding to $\pi^*$, and $A^\pi$ is the advantage function associated with $\pi$. By definition, we have $\sigma_*(\cdot, \cdot) = \pi^*(\cdot \mid \cdot) \cdot \nu_*(\cdot)$. Meanwhile, it holds for all $s \in \mathcal{S}$ that

$$
\begin{aligned}
\mathbb{E}_{\pi^*}\big[A^\pi(s, a)\big] &= \mathbb{E}_{\pi^*}\big[Q^\pi(s, a)\big] - V^\pi(s) = \langle Q^\pi(s, \cdot), \pi^*(\cdot \mid s)\rangle - \langle Q^\pi(s, \cdot), \pi(\cdot \mid s)\rangle \\
&= \langle Q^\pi(s, \cdot), \pi^*(\cdot \mid s) - \pi(\cdot \mid s)\rangle.
\end{aligned} \tag{D.2}
$$

Combining (D.1) and (D.2), we conclude that

$$J(\pi^*) - J(\pi) = (1 - \gamma)^{-1} \cdot \mathbb{E}_{\nu_*}\big[\langle Q^\pi(s, \cdot), \pi^*(\cdot \mid s) - \pi(\cdot \mid s)\rangle\big],$$

which concludes the proof of Lemma D.1. $\qquad\square$

### D.1  PROOF OF THEOREM 4.7

*Proof.* We first lower bound the difference between the expected total rewards of $\pi_{\theta_{i+1}}$ and $\pi_{\theta_i}$. By Assumption 4.6, $\nabla_\theta J(\pi_\theta)$ is $L$-Lipschitz continuous. Thus, it holds that

$$J(\pi_{\theta_{i+1}}) - J(\pi_{\theta_i}) \geq \eta \cdot \nabla_\theta J(\pi_{\theta_i})^\top \delta_i - L/2 \cdot \|\theta_{i+1} - \theta_i\|_2^2, \tag{D.3}$$

where $\delta_i = (\theta_{i+1} - \theta_i)/\eta$. Recall that $\xi_i = \widehat{\nabla}_\theta J(\pi_{\theta_i}) - \mathbb{E}[\widehat{\nabla}_\theta J(\pi_{\theta_i})]$, where the expectation is taken over $\sigma_i$ given $\theta_i$ and $\omega_i$. It holds that

$$\nabla_\theta J(\pi_{\theta_i})^\top \delta_i = \Big(\nabla_\theta J(\pi_{\theta_i}) - \mathbb{E}\big[\widehat{\nabla}_\theta J(\pi_{\theta_i})\big]\Big)^\top \delta_i - \xi_i^\top \delta_i + \widehat{\nabla}_\theta J(\pi_{\theta_i})^\top \delta_i. \tag{D.4}$$

On the right-hand side of (D.4), the first term represents the error of estimating $\nabla_\theta J(\pi_{\theta_i})$ using $\mathbb{E}[\widehat{\nabla}_\theta J(\pi_{\theta_i})] = \mathbb{E}_{\sigma_i}[\nabla_\theta \log \pi_{\theta_i}(a \mid s) \cdot Q_{\omega_i}(s, a)]$, the second term is related to the variance of the estimator $\widehat{\nabla}_\theta J(\pi_{\theta_i})$ of the policy gradient $\nabla_\theta J(\pi_{\theta_i})$, and the last term relates the increment $\delta_i$ of the actor update to $\widehat{\nabla}_\theta J(\pi_{\theta_i})$. In the following lemma, we establish a lower bound of the first term.

**Lemma D.2.** It holds that

$$\left| \Big(\nabla_\theta J(\pi_{\theta_i}) - \mathbb{E}\big[\widehat{\nabla}_\theta J(\pi_{\theta_i})\big]\Big)^\top \delta_i \right| \leq 4\kappa \cdot R/\eta \cdot \|Q^{\pi_{\theta_i}} - Q_{\omega_i}\|_{\varsigma_i},$$

where $\widehat{\nabla} J(\pi_{\theta_i})$ is defined in (3.7), $\varsigma_i$ is the stationary state-action distribution, and $\kappa$ is the absolute constant defined in Assumption 4.5. Here the expectation is taken over $\sigma_i$ given $\theta_i$ and $\omega_i$.

*Proof.* See §H.5 for a detailed proof. $\qquad\square$

For the second term on the right-hand side of (D.4), we have

$$-\xi_i^\top \delta_i \geq -\|\xi_i\|_2^2/2 - \|\delta_i\|_2^2/2. \tag{D.5}$$

Now it remains to lower bound the third term on the right-hand side of (D.4). For notational simplicity, we define

$$e_i = \theta_{i+1} - \big(\theta_i + \eta \cdot \widehat{\nabla} J(\pi_{\theta_i})\big) = \Pi_{\mathcal{B}}\big(\theta_i + \eta \cdot \widehat{\nabla} J(\pi_{\theta_i})\big) - \big(\theta_i + \eta \cdot \widehat{\nabla} J(\pi_{\theta_i})\big),$$

where $\Pi_{\mathcal{B}}$ is the projection operator onto $\mathcal{B}$. It then holds that

$$e_i^\top \Big[\Pi_{\mathcal{B}}\big(\theta_i + \eta \cdot \widehat{\nabla} J(\pi_{\theta_i})\big) - x\Big] = e_i^\top(\theta_{i+1} - x) \leq 0, \quad \forall x \in \mathcal{B}. \tag{D.6}$$

Specifically, setting $x = \theta_i$ in (D.6), we obtain that $e_i^\top \delta_i \leq 0$, which implies

$$\widehat{\nabla} J_\theta(\pi_{\theta_i})^\top \delta_i = (\delta_i - e_i/\eta)^\top \delta_i \geq \|\delta_i\|_2^2. \tag{D.7}$$

By plugging Lemma D.2, (D.5), and (D.7) into (D.4), we obtain that

$$\nabla_\theta J(\pi_{\theta_i})^\top \delta_i \geq -4\kappa \cdot R/\eta \cdot \|Q^{\pi_{\theta_i}} - Q_{\omega_i}\|_{\varsigma_i} + \|\delta_i\|_2^2/2 - \|\xi_i\|_2^2/2. \tag{D.8}$$

Thus, by plugging (D.8) and the definition that $\delta_i = (\theta_{i+1} - \theta_i)/\eta$ into (D.3), we obtain for all $i \in [T]$ that

$$\begin{aligned}
(1 - L \cdot \eta) &\cdot \mathbb{E}\big[\|\delta_i\|_2^2/2\big] \\
&\leq \eta^{-1} \cdot \mathbb{E}\big[J(\pi_{\theta_{i+1}}) - J(\pi_{\theta_i})\big] + 4\kappa \cdot R/\eta \cdot \mathbb{E}\big[\|Q^{\pi_{\theta_i}} - Q_{\omega_i}\|_{\varsigma_i}\big] + \mathbb{E}\big[\|\xi_i\|_2^2/2\big],
\end{aligned} \tag{D.9}$$

where the expectations are taking over all the randomness.

Now we turn to characterize $\|\rho_i - \delta_i\|_2$. By the definition of $\rho_i$ in (4.3), we have

$$\begin{aligned}
\|\rho_i - \delta_i\|_2 &= \eta^{-1} \cdot \Big\| \Pi_{\mathcal{B}}\big(\theta_i + \eta \cdot \nabla_\theta J(\pi_{\theta_i})\big) - \theta_i - \Big(\Pi_{\mathcal{B}}\big(\theta_i + \eta \cdot \widehat{\nabla}_\theta J(\pi_{\theta_i})\big) - \theta_i\Big)\Big\|_2 \\
&= \eta^{-1} \cdot \Big\| \Pi_{\mathcal{B}}\big(\theta_i + \eta \cdot \nabla_\theta J(\pi_{\theta_i})\big) - \Pi_{\mathcal{B}}\big(\theta_i + \eta \cdot \widehat{\nabla}_\theta J(\pi_{\theta_i})\big)\Big\|_2 \\
&\leq \|\nabla_\theta J(\pi_{\theta_i}) - \widehat{\nabla}_\theta J(\pi_{\theta_i})\|_2.
\end{aligned} \tag{D.10}$$

The following lemma further upper bounds the right-hand side of (D.10).

**Lemma D.3.** It holds for all $i \in [T]$ that

$$\mathbb{E}\big[\|\nabla_\theta J(\pi_{\theta_i}) - \widehat{\nabla}_\theta J(\pi_{\theta_i})\|_2^2\big] \leq 2\mathbb{E}\big[\|\xi_i\|_2^2\big] + 8\kappa^2 \cdot \mathbb{E}\big[\|Q^{\pi_{\theta_i}} - Q_{\omega_i}\|_{\varsigma_i}^2\big].$$

Here the expectations are taken over all the randomness.

*Proof.* See §H.6 for a detailed proof. □

Recall that we set $\eta = 1/\sqrt{T}$. Upon telescoping (D.9), it holds for $T \geq 4L^2$ that

$$\begin{aligned}
\min_{i \in [T]} \mathbb{E}\big[\|\rho_i\|_2^2\big] &\leq 1/T \cdot \sum_{i=1}^T \mathbb{E}\big[\|\rho_i\|_2^2\big] \\
&\leq 1/T \cdot \sum_{i=1}^T \Big(2\mathbb{E}\big[\|\delta_i\|_2^2\big] + 2\mathbb{E}\big[\|\rho_i - \delta_i\|_2^2\big]\Big) \\
&\leq 1/T \cdot \sum_{i=1}^T 4(1 - L \cdot \eta) \cdot \mathbb{E}\big[\|\delta_i\|_2^2\big] + 2\mathbb{E}\big[\|\rho_i - \delta_i\|_2^2\big] \\
&\leq 8/\sqrt{T} \cdot \mathbb{E}\big[J(\pi_{\theta_{T+1}}) - J(\pi_{\theta_1})\big] + 8/T \cdot \sum_{i=1}^T \mathbb{E}\big[\|\xi_i\|_2^2\big] + \varepsilon_Q(T), \tag{D.11}
\end{aligned}$$

where the third inequality follows from the fact that $1 - L \cdot \eta \geq 1/2$, while the fourth inequality follows from (D.9), (D.10), and Lemma D.3. Here the expectations are taken over all the randomness, and $\varepsilon_Q(T)$ is defined as

$$\varepsilon_Q(T) = 32\kappa \cdot R/\sqrt{T} \cdot \sum_{i=1}^{T} \mathbb{E}\big[\|Q^{\pi_{\theta_i}} - Q_{\omega_i}\|_{\varsigma_i}\big] + 16\kappa^2/T \cdot \sum_{i=1}^{T} \mathbb{E}\big[\|Q^{\pi_{\theta_i}} - Q_{\omega_i}\|_{\varsigma_i}^2\big].$$

By Proposition 4.3 and Assumption 4.4, it holds for all $i \in [T]$ that

$$\mathbb{E}\big[\|Q^{\pi_{\theta_i}} - Q_{\omega_i}\|_{\varsigma_i}^2\big] = \mathcal{O}(R^3 \cdot m^{-1/2} + R^{5/2} \cdot m^{-1/4}), \qquad \mathbb{E}\big[\|\xi_i\|_2^2\big] \leq \sigma_\xi^2/B. \tag{D.12}$$

By plugging (D.12) into (D.11), we conclude that

$$\min_{i \in [T]} \mathbb{E}\big[\|\rho_i\|_2^2\big] \leq 8/\sqrt{T} \cdot \mathbb{E}\big[J(\pi_{\theta_{T+1}}) - J(\pi_{\theta_1})\big] + 8\sigma_\xi^2/B + \varepsilon_Q(T),$$

where

$$\varepsilon_Q(T) = \kappa \cdot \mathcal{O}(R^{5/2} \cdot m^{-1/4} \cdot T^{1/2} + R^{9/4} \cdot m^{-1/8} \cdot T^{1/2}).$$

Thus, we complete the proof of Theorem 4.7. $\qquad\square$

## D.2 PROOF OF THEOREM 4.8

*Proof.* Since $\widehat{\theta}$ is a stationary point of $J(\pi_\theta)$, it holds that

$$\nabla_\theta J(\pi_{\widehat{\theta}})^\top (\theta - \widehat{\theta}) \leq 0, \quad \forall \theta \in \mathcal{B}. \tag{D.13}$$

Therefore, by Proposition 3.1, we obtain from (D.13) that

$$\nabla_\theta J(\pi_{\widehat{\theta}})^\top (\theta - \widehat{\theta}) = \mathbb{E}_{\sigma_{\pi_{\widehat{\theta}}}}\big[\overline{\phi}_{\widehat{\theta}}(s,a)^\top (\theta - \widehat{\theta}) \cdot Q^{\pi_{\widehat{\theta}}}(s,a)\big] \leq 0, \quad \forall \theta \in \mathcal{B}. \tag{D.14}$$

Here $\phi_{\widehat{\theta}}$ and $\overline{\phi}_{\widehat{\theta}}$ are defined in (3.3) and (B.2) with $\theta = \widehat{\theta}$, respectively. Note that

$$\mathbb{E}_{\sigma_{\pi_{\widehat{\theta}}}}\big[\overline{\phi}_{\widehat{\theta}}(s,a)^\top (\theta - \widehat{\theta}) \cdot V^{\pi_{\widehat{\theta}}}(s)\big] = \mathbb{E}_{\nu_{\pi_{\widehat{\theta}}}}\Big[\mathbb{E}_{\pi_{\widehat{\theta}}}\big[\overline{\phi}_{\widehat{\theta}}(s,a)\big]^\top (\theta - \widehat{\theta}) \cdot V^{\pi_{\widehat{\theta}}}(s)\Big] = 0,$$

$$\mathbb{E}_{\sigma_{\pi_{\widehat{\theta}}}}\Big[\mathbb{E}_{\pi_{\widehat{\theta}}}\big[\phi_{\widehat{\theta}}(s,a')^\top (\theta - \widehat{\theta})\big] \cdot A^{\pi_{\widehat{\theta}}}(s,a)\Big] = \mathbb{E}_{\nu_{\pi_{\widehat{\theta}}}}\Big[\mathbb{E}_{\pi_{\widehat{\theta}}}\big[\phi_{\widehat{\theta}}(s,a')^\top (\theta - \widehat{\theta})\big] \cdot \mathbb{E}_{\pi_{\widehat{\theta}}}\big[A^{\pi_{\widehat{\theta}}}(s,a)\big]\Big] = 0,$$

which holds since $\mathbb{E}_{\pi_{\widehat{\theta}}}[\overline{\phi}_{\widehat{\theta}}(s,a)] = \mathbb{E}_{\pi_{\widehat{\theta}}}[A^{\pi_{\widehat{\theta}}}(s,a)] = 0$ for all $s \in \mathcal{S}$. Thus, by (D.14), we have

$$\mathbb{E}_{\sigma_{\pi_{\widehat{\theta}}}}\big[\overline{\phi}_{\widehat{\theta}}(s,a)^\top (\theta - \widehat{\theta}) \cdot Q^{\pi_{\widehat{\theta}}}(s,a)\big]$$

$$= \mathbb{E}_{\sigma_{\pi_{\widehat{\theta}}}}\big[\phi_{\widehat{\theta}}(s,a)^\top (\theta - \widehat{\theta}) \cdot A^{\pi_{\widehat{\theta}}}(s,a)\big] - \mathbb{E}_{\sigma_{\pi_{\widehat{\theta}}}}\Big[\mathbb{E}_{\pi_{\widehat{\theta}}}\big[\phi_{\widehat{\theta}}(s,a')^\top (\theta - \widehat{\theta})\big] \cdot A^{\pi_{\widehat{\theta}}}(s,a)\Big]$$

$$\quad + \mathbb{E}_{\sigma_{\pi_{\widehat{\theta}}}}\big[\overline{\phi}_{\widehat{\theta}}(s,a)^\top (\theta - \widehat{\theta}) \cdot V^{\pi_{\widehat{\theta}}}(s)\big]$$

$$= \mathbb{E}_{\sigma_{\pi_{\widehat{\theta}}}}\big[\phi_{\widehat{\theta}}(s,a)^\top (\theta - \widehat{\theta}) \cdot A^{\pi_{\widehat{\theta}}}(s,a)\big] \leq 0, \quad \forall \theta \in \mathcal{B}. \tag{D.15}$$

Meanwhile, by Lemma D.1 we have

$$(1 - \gamma) \cdot \big(J(\pi^*) - J(\pi_{\widehat{\theta}})\big) = \mathbb{E}_{\nu_*}\big[\langle A^{\pi_{\widehat{\theta}}}(s,\cdot), \pi^*(\cdot \,|\, s) - \pi_{\widehat{\theta}}(\cdot \,|\, s)\rangle\big]. \tag{D.16}$$

In what follows, we write $\Delta_\theta = \theta - \widehat{\theta}$. Combining (D.15) and (D.16), we obtain that

$$(1 - \gamma) \cdot \big(J(\pi^*) - J(\pi_{\widehat{\theta}})\big)$$

$$\leq \mathbb{E}_{\nu_*}\big[\langle A^{\pi_{\widehat{\theta}}}(s,\cdot), \pi^*(\cdot \,|\, s) - \pi_{\widehat{\theta}}(\cdot \,|\, s)\rangle\big] - \mathbb{E}_{\sigma_{\pi_{\widehat{\theta}}}}\big[\phi_{\widehat{\theta}}(s,a)^\top \Delta_\theta \cdot A^{\pi_{\widehat{\theta}}}(s,a)\big]$$

$$= \mathbb{E}_{\nu_*}\big[\langle A^{\pi_{\widehat{\theta}}}(s,\cdot), \pi^*(\cdot \,|\, s) - \pi_{\widehat{\theta}}(\cdot \,|\, s)\rangle\big] - \mathbb{E}_{\nu_{\pi_{\widehat{\theta}}}}\big[\langle A^{\pi_{\widehat{\theta}}}(s,\cdot), \phi_{\widehat{\theta}}(s,\cdot)^\top \Delta_\theta \cdot \pi_{\widehat{\theta}}(\cdot \,|\, s)\rangle\big], \tag{D.17}$$

where we use the fact that $\sigma_{\pi_{\widehat{\theta}}}(\cdot, \cdot) = \pi_{\widehat{\theta}}(\cdot \,|\, \cdot) \cdot \nu_{\pi_{\widehat{\theta}}}(\cdot)$. It remains to upper bound the right-hand side of (D.17). By calculation, it holds for all $(s,a) \in \mathcal{S} \times \mathcal{A}$ that

$$\big(\pi^*(a \,|\, s) - \pi_{\widehat{\theta}}(a \,|\, s)\big)\mathrm{d}\nu_*(s) - \phi_{\widehat{\theta}}(s,a)^\top \Delta_\theta \cdot \pi_{\widehat{\theta}}(a \,|\, s)\mathrm{d}\nu_{\widehat{\theta}}(s)$$

$$= \left(\frac{\pi^*(a \,|\, s) - \pi_{\widehat{\theta}}(a \,|\, s)}{\pi_{\widehat{\theta}}(a \,|\, s)} \cdot \frac{\mathrm{d}\nu_*}{\mathrm{d}\nu_{\widehat{\theta}}}(s) - \phi_{\widehat{\theta}}(s,a)^\top \Delta_\theta\right) \cdot \pi_{\widehat{\theta}}(a \,|\, s)\mathrm{d}\nu_{\pi_{\widehat{\theta}}}(s)$$

$$= \big(u_{\widehat{\theta}}(s,a) - \phi_{\widehat{\theta}}(s,a)^\top \theta\big)\mathrm{d}\sigma_{\pi_{\widehat{\theta}}}(s,a), \tag{D.18}$$

where $u_{\widehat{\theta}}$ is defined as

$$u_{\widehat{\theta}}(s, a) = \frac{\mathrm{d}\sigma_{\pi^*}}{\mathrm{d}\sigma_{\pi_{\widehat{\theta}}}}(s, a) - \frac{\mathrm{d}\nu_{\pi^*}}{\mathrm{d}\nu_{\pi_{\widehat{\theta}}}}(s) + \phi_{\widehat{\theta}}(s, a)^\top \widehat{\theta}, \quad \forall (s, a) \in \mathcal{S} \times \mathcal{A}.$$

Here $\mathrm{d}\sigma_{\pi^*}/\mathrm{d}\sigma_{\pi_{\widehat{\theta}}}$ and $\mathrm{d}\nu_{\pi^*}/\mathrm{d}\nu_{\pi_{\widehat{\theta}}}$ are the Radon-Nikodym derivatives. By plugging (D.18) into (D.17), we obtain that

$$
\begin{aligned}
(1 - \gamma) &\cdot \big( J(\pi^*) - J(\pi_{\widehat{\theta}}) \big) \\
&\leq \mathbb{E}_{\nu_{\pi_{\widehat{\theta}}}}\big[ \langle A^{\pi_{\widehat{\theta}}}(s, \cdot), \pi^*(\cdot \,|\, s) - \pi_{\widehat{\theta}}(\cdot \,|\, s) \rangle \big] - \mathbb{E}_{\nu_{\pi_{\widehat{\theta}}}}\big[ \langle A^{\pi_{\widehat{\theta}}}(s, \cdot), \phi_{\widehat{\theta}}(s, \cdot)^\top \Delta_\theta \cdot \pi_{\widehat{\theta}}(\cdot \,|\, s) \rangle \big] \\
&= \int_{\mathcal{S}} \sum_{a \in \mathcal{A}} A^{\pi_{\widehat{\theta}}}(s, a) \cdot \Big( \big( \pi^*(a \,|\, s) - \pi_{\widehat{\theta}}(a \,|\, s) \big) \mathrm{d}\nu_*(s) - \phi_{\widehat{\theta}}(s, a)^\top \Delta_\theta \cdot \pi_{\widehat{\theta}}(a \,|\, s) \mathrm{d}\nu_{\widehat{\theta}}(s) \Big) \\
&= \int_{\mathcal{S} \times \mathcal{A}} A^{\pi_{\widehat{\theta}}}(s, a) \cdot \big( u_{\widehat{\theta}}(s, a) - \phi_{\widehat{\theta}}(s, a)^\top \Delta_\theta \big) \mathrm{d}\sigma_{\pi_{\widehat{\theta}}}(s, a) \\
&\leq \| A^{\pi_{\widehat{\theta}}}(\cdot, \cdot) \|_{\sigma_{\pi_{\widehat{\theta}}}} \cdot \| u_{\widehat{\theta}}(\cdot, \cdot) - \phi_{\widehat{\theta}}(\cdot, \cdot)^\top \theta \|_{\sigma_{\pi_{\widehat{\theta}}}},
\end{aligned}
\tag{D.19}
$$

where the second equality follows from (D.18) and the last inequality is from the Cauchy-Schwartz inequality. Note that $|A^{\pi_{\widehat{\theta}}}(s, a)| \leq 2Q_{\max}$ for all $(s, a) \in \mathcal{S} \times \mathcal{A}$. Therefore, it follows from (D.19) that

$$(1 - \gamma) \cdot \big( J(\pi^*) - J(\pi_{\widehat{\theta}}) \big) \leq 2Q_{\max} \cdot \| u_{\widehat{\theta}}(\cdot, \cdot) - \phi_{\widehat{\theta}}(\cdot, \cdot)^\top \theta \|_{\sigma_{\pi_{\widehat{\theta}}}}, \quad \forall \theta \in \mathcal{B}. \tag{D.20}$$

Finally, by taking the infimum of the right-hand side of (D.20) with respect to $\theta \in \mathcal{B}$, we obtain that

$$(1 - \gamma) \cdot \big( J(\pi^*) - J(\pi_{\widehat{\theta}}) \big) \leq 2Q_{\max} \cdot \inf_{\theta \in \mathcal{B}} \| u_{\widehat{\theta}}(\cdot, \cdot) - \phi_{\widehat{\theta}}(\cdot, \cdot)^\top \theta \|_{\sigma_{\pi_{\widehat{\theta}}}},$$

which concludes the proof of Theorem 4.8. $\qquad \square$

## D.3 Proof of Theorem A.4

*Proof.* For notational simplicity, we write $\pi_i = \pi_{\theta_i}$ hereafter. In the following lemma, we characterize the performance difference $J(\pi^*) - J(\pi_i)$ based on Lemma D.1.

**Lemma D.4.** It holds that

$$
\begin{aligned}
(1 - \gamma) \cdot \eta \cdot \big( J(\pi^*) - J(\pi_i) \big) = \mathbb{E}_{\nu_*}\Big[ &D_{\mathrm{KL}}\big( \pi^*(\cdot \,|\, s) \big\| \pi_i(\cdot \,|\, s) \big) - D_{\mathrm{KL}}\big( \pi^*(\cdot \,|\, s) \big\| \pi_{i+1}(\cdot \,|\, s) \big) \\
&- D_{\mathrm{KL}}\big( \pi_{i+1}(\cdot \,|\, s) \big\| \pi_i(\cdot \,|\, s) \big) \Big] - H_i,
\end{aligned}
$$

where $H_i$ is defined as

$$
H_i = \underbrace{\mathbb{E}_{\nu_*}\Big[ \big\langle \log\big( \pi_{i+1}(\cdot \,|\, s)/\pi_i(\cdot \,|\, s) \big) - \eta \cdot Q_{\omega_i}(s, \cdot), \pi^*(\cdot \,|\, s) - \pi_i(\cdot \,|\, s) \big\rangle \Big]}_{(i)} \tag{D.21}
$$

$$
+ \underbrace{\eta \cdot \mathbb{E}_{\nu_*}\big[ \langle Q_{\omega_i}(s, \cdot) - Q^{\pi_i}(s, \cdot), \pi^*(\cdot \,|\, s) - \pi_i(\cdot \,|\, s) \rangle \big]}_{(ii)}
$$

$$
+ \underbrace{\mathbb{E}_{\nu_*}\Big[ \big\langle \log\big( \pi_i(\cdot \,|\, s)/\pi_{i+1}(\cdot \,|\, s) \big), \pi_{i+1}(\cdot \,|\, s) - \pi_i(\cdot \,|\, s) \big\rangle \Big]}_{(iii)}.
$$

*Proof.* See §H.7 for a detailed proof. $\qquad \square$

Here $H_i$ defined in (D.21) of Lemma D.4 consists of three terms. Specifically, (i) is related to the error of estimating the natural policy gradient using (3.8). Also, (ii) is related to the error of estimating $Q^{\pi_i}$ using $Q_{\omega_i}$. Meanwhile, (iii) is the remainder term. We upper bound these three terms in §H.8. Combining these upper bounds, we obtain the following lemma.

**Lemma D.5.** Under Assumptions 4.2 and A.3, we have

$$\mathbb{E}\left[\left|H_i\right| - \mathbb{E}_{\nu_*}\left[D_{\mathrm{KL}}\big(\pi_{i+1}(\cdot\,|\,s)\big\|\pi_i(\cdot\,|\,s)\big)\right]\right] \leq \eta^2 \cdot (9R^2 + M^2) + \eta \cdot (\varphi_i' + \psi_i') \cdot \varepsilon_{Q,i} + \varepsilon_i.$$

Here the expectation is taken over all the randomness. Meanwhile, $\varphi_i'$ and $\psi_i'$ are the concentrability coefficients defined in (A.1) of Assumption A.2, $\varepsilon_{Q,i}$ is defined as $\varepsilon_{Q,i} = \mathbb{E}[\|Q^{\pi_i} - Q_{\omega_i}\|_{\varsigma_i}]$, $M$ is the absolute constant defined in Assumption A.3, and $\varepsilon_i$ is defined as

$$\varepsilon_i = \sqrt{2} \cdot R^{1/2} \cdot \eta \cdot (\varphi_i + \psi_i) \cdot \tau_i^{-1} \cdot \left\{\mathbb{E}\big[\|\xi_i(\delta_i)\|_2\big] + \mathbb{E}\big[\|\xi_i(\omega_i)\|_2\big]\right\}^{1/2} \quad \text{(D.22)}$$
$$+ \mathcal{O}\big((\tau_{i+1} + \eta) \cdot R^{3/2} \cdot m^{-1/4} + \eta \cdot R^{5/4} \cdot m^{-1/8}\big).$$

Here $\xi_i(\delta_i)$ and $\xi_i(\omega_i)$ are defined in Assumption A.1, where $\delta_i = \eta^{-1} \cdot (\tau_{i+1} \cdot \theta_{i+1} - \tau_i \cdot \theta_i)$, while $\varphi_i$ and $\psi_i$ are the concentrability coefficients defined in (A.1) of Assumption A.2.

*Proof.* See §H.8 for a detailed proof. □

By Lemmas D.4 and D.5, we obtain that

$$(1 - \gamma) \cdot \mathbb{E}\big[J(\pi^*) - J(\pi_i)\big] \leq \eta^{-1} \cdot \mathbb{E}\bigg[\mathbb{E}_{\nu_*}\Big[D_{\mathrm{KL}}\big(\pi^*(\cdot\,|\,s)\big\|\pi_i(\cdot\,|\,s)\big) \quad \text{(D.23)}$$
$$- D_{\mathrm{KL}}\big(\pi^*(\cdot\,|\,s)\big\|\pi_{i+1}(\cdot\,|\,s)\big)\Big]\bigg]$$
$$+ \eta \cdot (9R^2 + M^2) + \eta^{-1} \cdot \varepsilon_i + (\varphi_i' + \psi_i') \cdot \varepsilon_{Q,i},$$

where $\varepsilon_{Q,i}$ is defined as $\varepsilon_{Q,i} = \mathbb{E}[\|Q^{\pi_i} - Q_{\omega_i}\|_{\varsigma_i}]$, $M$ is the absolute constant defined in Assumption A.3, $\varepsilon_i$ is defined in (D.22) of Lemma D.5, and the expectations are taken over all the randomness. Recall that we set $\eta = 1/\sqrt{T}$. Upon telescoping (D.23), we obtain that

$$(1 - \gamma) \cdot \min_{i \in [T]} \mathbb{E}\big[J(\pi^*) - J(\pi_i)\big] \leq \frac{1 - \gamma}{T} \cdot \sum_{i=1}^{T} \mathbb{E}\big[J(\pi^*) - J(\pi_i)\big] \quad \text{(D.24)}$$
$$\leq \frac{1}{\sqrt{T}} \cdot \left(\mathbb{E}\bigg[\mathbb{E}_{\nu_*}\Big[D_{\mathrm{KL}}\big(\pi^*(\cdot\,|\,s)\big\|\pi_1(\cdot\,|\,s)\big)\Big]\bigg] + 9R^2 + M^2\right)$$
$$+ \frac{1}{T} \cdot \sum_{i=1}^{T}\big(\sqrt{T} \cdot \varepsilon_i + (\varphi_i' + \psi_i') \cdot \varepsilon_{Q,i}\big),$$

where the expectations are taken over all the randomness and the last inequality follows from the fact that

$$D_{\mathrm{KL}}\big(\pi^*(\cdot\,|\,s)\big\|\pi_{T+1}(\cdot\,|\,s)\big) \geq 0, \quad \forall s \in \mathcal{S}, \; \forall \theta_{T+1} \in \mathbb{R}^{md}.$$

In what follows, we upper bound the right-hand side of (D.24). Note that we set $\tau_1 = 0$. By the parameterization of policy in (3.2), it then holds that $\pi_1(\cdot\,|\,s)$ is uniform over $\mathcal{A}$ for all $s \in \mathcal{S}$ and $\theta_1 \in \mathbb{R}^{md}$. Therefore, we obtain that

$$D_{\mathrm{KL}}\big(\pi^*(\cdot\,|\,s)\big\|\pi_1(\cdot\,|\,s)\big) \leq \log|\mathcal{A}|, \quad \forall s \in \mathcal{S}, \; \forall \theta_1 \in \mathbb{R}^{md}. \quad \text{(D.25)}$$

Meanwhile, by Assumption A.1, we have

$$\mathbb{E}\big[\|\xi_i(\delta_i)\|_2\big] \leq \left\{\mathbb{E}\Big[\mathbb{E}_{\sigma_i}\big[\|\xi_i(\delta_i)\|_2^2\big]\Big]\right\}^{1/2} \leq \tau_i^2 \cdot \sigma_\xi \cdot B^{-1/2},$$

where the expectation $\mathbb{E}_{\sigma_i}[\|\xi_i(\delta_i)\|_2^2]$ is taken over $\sigma_i$ given $\theta_i$ and $\omega_i$, while the other expectations are taken over all the randomness. A similar upper bound holds for $\mathbb{E}[\|\xi_i(\omega_i)\|_2]$. Therefore, by plugging the upper bounds of $\mathbb{E}[\|\xi_i(\sigma_i)\|_2]$ and $\mathbb{E}[\|\xi_i(\omega_i)\|_2]$ into $\varepsilon_i$ defined in (D.22) of Lemma D.5, we obtain from Assumption A.2 that

$$\sqrt{T} \cdot \varepsilon_i \leq 2\sqrt{2}c_0 \cdot R^{1/2} \cdot \sigma_\xi^{1/2} \cdot B^{-1/4} \quad \text{(D.26)}$$
$$+ \mathcal{O}\big((\tau_{i+1} \cdot T^{1/2} + 1) \cdot R^{3/2} \cdot m^{-1/4} + R^{5/4} \cdot m^{-1/8}\big).$$

Also, combining Assumption A.2 and Proposition 4.3, it holds that

$$(\varphi_i' + \psi_i') \cdot \varepsilon_{Q,i} \leq 2c_0 \cdot \mathbb{E}\big[\|Q^{\pi_i} - Q_{\omega_i}\|_{\varsigma_i}\big] = c_0 \cdot \mathcal{O}(R^{3/2} \cdot m^{-1/4} + R^{5/4} \cdot m^{-1/8}). \quad \text{(D.27)}$$

Finally, by plugging (D.25), (D.26), and (D.27) into (D.24) and setting

$$\bar{\epsilon}_i(T) = \sqrt{T} \cdot \varepsilon_i + (\varphi_i' + \psi_i') \cdot \varepsilon_{Q,i},$$

we complete the proof of Theorem A.4. $\qquad\square$

## E  LINEARIZATION ERROR

In this section, we lay out a fundamental lemma that characterizes the distance between a two-layer neural network $\phi_\theta^\top \theta$ and its linearization $\phi_0^\top \theta$, where $\phi_\theta$ is the feature mapping of the two-layer neural network defined in (3.3) and $\phi_0$ is the feature mapping corresponding to the initial parameter $W_{\text{init}}$.

We first introduce a function class that consists of lineaizations of $f((\cdot,\cdot); W)$ defined in (3.1).

**Definition E.1** (Function Class). Let $R > 0$ be an absolute constant. For all $m \in \mathbb{N}$, we define

$$\widetilde{\mathcal{F}}_{R,m} = \bigg\{ \widehat{f}((s,a); W) = \frac{1}{\sqrt{m}} \cdot \sum_{r=1}^{m} b_r \cdot \mathbb{1}\big\{[W_{\text{init}}]_r^\top (s,a) > 0\big\} \cdot [W]_r^\top (s,a) \quad \text{(E.1)}$$

$$: \|W - W_{\text{init}}\|_2 \leq R \bigg\},$$

where $[W_{\text{init}}]_r \sim N(0, I_d/d)$ and $b_r \sim \text{Unif}(\{-1,1\})$ are the initial parameters of the two-layer neural network defined in (3.1).

Note that $\widetilde{\mathcal{F}}_{R,m}$ in (E.1) is a class of functions that are linear in $W$ but nonlinear in $(s,a)$. Meanwhile, it holds that $\nabla_W \widehat{f}((s,a); W) = \nabla_W f((s,a); W)|_{W=W_{\text{init}}}$ for all $(s,a) \in \mathcal{S} \times \mathcal{A}$, where $f((\cdot,\cdot); W)$ is the two-layer neural network defined in (3.1). Thus, $\widehat{f}((\cdot,\cdot); W)$ can be viewed as the linearization of $f((\cdot,\cdot); W)$ at the initial parameter $W_{\text{init}}$. Moreover, for a fixed $R$, the linearization error of $\widehat{f}((\cdot,\cdot); W)$ decays to zero as the width $m \to \infty$. Intuitively, since $\|W - W_{\text{init}}\|_2$ is upper bounded by $R$, the differences between blocks $\|[W]_r - [W_{\text{init}}]_r\|_2$ are sufficiently small for a sufficiently large $m$ and all $r \in [m]$. As a result, for a sufficiently large $m$, we have $\mathbb{1}\{[W_{\text{init}}]_r^\top (s,a) > 0\} = \mathbb{1}\{[W]_r^\top (s,a) > 0\}$ with high probability for all $r \in [m]$ and $(s,a) \in \mathcal{S} \times \mathcal{A}$, and thus $f((\cdot,\cdot); W)$ is well approximated by its linearization $\widehat{f}((\cdot,\cdot); W)$.

The following lemma formally characterizes the corresponding linearization error.

**Lemma E.2** (Linearization Error (Cai et al., 2019)). Let $W_{\text{init}}$ be the initial parameter of the two-layer neural network defined in (3.1). Let $\mathcal{B} = \{\alpha \in \mathbb{R}^{md} : \|\alpha - W_{\text{init}}\|_2 \leq R\}$. Under Assumption 4.2, it holds for all $\theta, \theta' \in \mathcal{B}$ that

$$\mathbb{E}_{\text{init}}\big[\|\phi_\theta(\cdot,\cdot)^\top \theta' - \phi_0(\cdot,\cdot)^\top \theta'\|_\sigma^2\big] = \mathcal{O}(R^3 \cdot m^{-1/2}),$$

where the expectation is taken over the random initialization. Here $\phi_\theta$ and $\phi_0$ are the feature mappings defined in (3.3), which correspond to $\theta$ and $W_{\text{init}}$, respectively, and $\sigma(\cdot,\cdot) = \pi(\cdot \,|\, \cdot) \cdot \nu(\cdot)$ is the distribution over $\mathcal{S} \times \mathcal{A}$ such that Assumption 4.2 holds.

*Proof.* By the definition of feature mapping in (3.3), we obtain that

$$\phi_\theta(s,a)^\top \theta' - \phi_0(s,a)^\top \theta'$$
$$= \frac{1}{\sqrt{m}} \cdot \sum_{r=1}^{m} \Big( \mathbb{1}\big\{(s,a)^\top [\theta]_r > 0\big\} - \mathbb{1}\big\{(s,a)^\top [W_{\text{init}}]_r > 0\big\} \Big) \cdot (s,a)^\top [\theta']_r. \quad \text{(E.2)}$$

Meanwhile, for $\mathbb{1}\{(s,a)^\top [\theta]_r > 0\} \neq \mathbb{1}\{(s,a)^\top [W_{\text{init}}]_r > 0\}$, we have

$$|(s,a)^\top [W_{\text{init}}]_r| \leq |(s,a)^\top [\theta]_r - (s,a)^\top [W_{\text{init}}]_r| \leq \|(s,a)\|_2 \cdot \|[\theta]_r - [W_{\text{init}}]_r\|_2, \quad \text{(E.3)}$$

where the last inequality follows from the Cauchy-Schwartz inequality. Recall that $\|(s,a)\|_2 \leq 1$ for all $(s,a) \in \mathcal{S} \times \mathcal{A}$. Thus, it follows from (E.3) that

$$
\begin{aligned}
&\big|\mathbb{1}\big\{(s,a)^\top[\theta]_r > 0\big\} - \mathbb{1}\big\{(s,a)^\top[W_{\text{init}}]_r > 0\big\}\big| \\
&\qquad \leq \mathbb{1}\big\{|(s,a)^\top[W_{\text{init}}]_r| \leq \|[\theta]_r - [W_{\text{init}}]_r\|_2\big\}.
\end{aligned}
\tag{E.4}
$$

By plugging (E.4) into (E.2), we obtain that

$$
\begin{aligned}
&|\phi_\theta(s,a)^\top\theta' - \phi_0(s,a)^\top\theta'| \\
&\quad\leq \frac{1}{\sqrt{m}} \cdot \sum_{r=1}^m \mathbb{1}\big\{|(s,a)^\top[W_{\text{init}}]_r| \leq \|[\theta]_r - [W_{\text{init}}]_r\|_2\big\} \cdot |(s,a)^\top[\theta']_r| \\
&\quad\leq \frac{1}{\sqrt{m}} \cdot \sum_{r=1}^m \mathbb{1}\big\{|(s,a)^\top[W_{\text{init}}]_r| \leq \|[\theta]_r - [W_{\text{init}}]_r\|_2\big\} \\
&\qquad\qquad \cdot \Big(|(s,a)^\top[W_{\text{init}}]_r| + \big|(s,a)^\top\big([\theta']_r - [W_{\text{init}}]_r\big)\big|\Big) \\
&\quad\leq \frac{1}{\sqrt{m}} \cdot \sum_{r=1}^m \mathbb{1}\big\{|(s,a)^\top[W_{\text{init}}]_r| \leq \|[\theta]_r - [W_{\text{init}}]_r\|_2\big\} \\
&\qquad\qquad \cdot \big(|(s,a)^\top[W_{\text{init}}]_r| + \|[\theta']_r - [W_{\text{init}}]_r\|_2\big),
\end{aligned}
\tag{E.5}
$$

where the last inequality follows from the Cauchy-Schwartz inequality and the fact that $\|(s,a)\|_2 \leq 1$. Following from the fact that $\mathbb{1}\{|x| \leq y\} \cdot |x| \leq \mathbb{1}\{|x| \leq y\} \cdot y$, we obtain from (E.5) that

$$
\begin{aligned}
&|\phi_\theta(s,a)^\top\theta' - \phi_0(s,a)^\top\theta'| \\
&\quad\leq \frac{1}{\sqrt{m}} \cdot \sum_{r=1}^m \mathbb{1}\big\{|(s,a)^\top[W_{\text{init}}]_r| \leq \|[\theta]_r - [W_{\text{init}}]_r\|_2\big\} \\
&\qquad\qquad \cdot \big(\|[\theta]_r - [W_{\text{init}}]_r\|_2 + \|[\theta']_r - [W_{\text{init}}]_r\|_2\big).
\end{aligned}
\tag{E.6}
$$

Therefore, following from the Cauchy-Schwartz inequality, we obtain from (E.6) that

$$
\begin{aligned}
&|\phi_\theta(s,a)^\top\theta' - \phi_0(s,a)^\top\theta'|^2 \\
&\quad\leq \frac{1}{m} \cdot \sum_{r=1}^m \mathbb{1}\big\{|(s,a)^\top[W_{\text{init}}]_r| \leq \|[\theta]_r - [W_{\text{init}}]_r\|_2\big\} \\
&\qquad\qquad \cdot \sum_{r=1}^m \big(2\|[\theta]_r - [W_{\text{init}}]_r\|_2^2 + 2\|[\theta']_r - [W_{\text{init}}]_r\|_2^2\big) \\
&\quad\leq \frac{1}{m} \cdot \sum_{r=1}^m \mathbb{1}\big\{|(s,a)^\top[W_{\text{init}}]_r| \leq \|[\theta]_r - [W_{\text{init}}]_r\|_2\big\} \cdot 2\big(\|\theta - W_{\text{init}}\|_2^2 + \|\theta' - W_{\text{init}}\|_2^2\big),
\end{aligned}
\tag{E.7}
$$

where the first inequality follows from the fact that $(x+y)^2 \leq 2x^2 + 2y^2$. Recall that $\theta, \theta' \in \mathcal{B}$, where $\mathcal{B} = \{\alpha \in \mathbb{R}^{md} : \|\alpha - W_{\text{init}}\|_2 \leq R\}$. Thus, following from (E.7), we have

$$
|\phi_\theta(s,a)^\top\theta' - \phi_0(s,a)^\top\theta'|^2 \leq \frac{4R^2}{m} \cdot \sum_{r=1}^m \mathbb{1}\big\{|(s,a)^\top[W_{\text{init}}]_r| \leq \|[\theta]_r - [W_{\text{init}}]_r\|_2\big\}.
\tag{E.8}
$$

By Assumption 4.2, we obtain from (E.8) that

$$
\begin{aligned}
\|\phi_\theta(\cdot,\cdot)^\top\theta' - \phi_0(\cdot,\cdot)^\top\theta'\|_\sigma^2 &= \mathbb{E}_\sigma\big[|\phi_\theta(s,a)^\top\theta' - \phi_0(s,a)^\top\theta'|^2\big] \\
&\leq \frac{4c \cdot R^2}{m} \cdot \sum_{r=1}^m \frac{\|[\theta]_r - [W_{\text{init}}]_r\|_2}{\|[W_{\text{init}}]_r\|_2},
\end{aligned}
\tag{E.9}
$$

where $c$ is the absolute constant defined by Assumption 4.2. It now suffices to take the expectation of the right-hand side of (E.9) over the random initialization. Following from the Cauchy-Schwartz

inequality, we obtain that

$$
\begin{aligned}
\left(\sum_{r=1}^{m} \frac{\|[\theta]_r - [W_{\text{init}}]_r\|_2}{\|[W_{\text{init}}]_r\|_2}\right)^2 &\leq \left(\sum_{r=1}^{m} \|[\theta]_r - [W_{\text{init}}]_r\|_2^2\right) \cdot \left(\sum_{r=1}^{m} 1/\|[W_{\text{init}}]_r\|_2^2\right) \\
&= \|\theta - W_{\text{init}}\|_2^2 \cdot \sum_{r=1}^{m} 1/\|[W_{\text{init}}]_r\|_2^2 \\
&\leq R^2 \cdot \sum_{r=1}^{m} 1/\|[W_{\text{init}}]_r\|_2^2,
\end{aligned}
\tag{E.10}
$$

where the last inequality follows from the fact that $\theta \in \mathcal{B}$. Therefore, combining (E.9) and (E.10), we conclude that

$$
\begin{aligned}
\mathbb{E}_{\text{init}}\left[\|\phi_\theta(\cdot,\cdot)^\top \theta' - \phi_0(\cdot,\cdot)^\top \theta'\|_\sigma^2\right] &\leq \frac{4c \cdot R^3}{m} \cdot \mathbb{E}_{\text{init}}\left[\left(\sum_{r=1}^{m} 1/\|[W_{\text{init}}]_r\|_2^2\right)^{1/2}\right] \\
&\leq \frac{4c \cdot R^3}{m} \cdot \left(\sum_{r=1}^{m} \mathbb{E}_{\text{init}}\left[1/\|[W_{\text{init}}]_r\|_2^2\right]\right)^{1/2} \\
&= 4c_1 \cdot R^3 \cdot m^{-1/2},
\end{aligned}
$$

where the second inequality follows from the Jensen's inequality and $c_1 = c \cdot \mathbb{E}_{x \sim N(0, I_d/d)}[1/\|x\|_2^2]$. Thus, we complete the proof of Lemma E.2. $\qquad\square$

By Lemma E.2, the linearization $\phi_0^\top \theta$ converges to the two-layer neural network $\phi_\theta^\top \theta$ as the width $m \to \infty$. Based on Lemma E.2, the following corollary characterizes a similar convergence where the feature mappings $\phi_\theta$ and $\phi_0$ are replaced by the centered feature mappings $\overline{\phi}_0$ and $\overline{\phi}_\theta$ defined in (B.1) and (B.2), respectively.

**Corollary E.3.** Let $W_{\text{init}}$ be the initial parameter and $\mathcal{B} = \{\alpha \in \mathbb{R}^{md} : \|\alpha - W_{\text{init}}\|_2 \leq R\}$ be the parameter space. Under Assumption 4.2, it holds for all $\theta, \theta' \in \mathcal{B}$ that

$$
\mathbb{E}_{\text{init}}\left[\|\overline{\phi}_\theta(\cdot,\cdot)^\top \theta' - \overline{\phi}_0(\cdot,\cdot)^\top \theta'\|_\sigma^2\right] = \mathcal{O}(R^3 \cdot m^{-1/2}),
$$

where the expectation is taken over the random initialization. Here $\overline{\phi}_0$ and $\overline{\phi}_\theta$ are the centered feature mappings defined in (B.1) and (B.2), respectively, and $\sigma(\cdot,\cdot) = \pi(\cdot\,|\,\cdot) \cdot \nu(\cdot)$ is the distribution over $\mathcal{S} \times \mathcal{A}$ such that Assumption 4.2 holds.

*Proof.* By the definitions of $\overline{\phi}_0$ and $\overline{\phi}_\theta$ in (B.1) and (B.2), respectively, we obtain that

$$
\begin{aligned}
\|\overline{\phi}_\theta(\cdot,\cdot)^\top \theta' - \overline{\phi}_0(\cdot,\cdot)^\top \theta'\|_\sigma^2 &= \left\|\phi_\theta(\cdot,\cdot)^\top \theta' - \phi_0(\cdot,\cdot)^\top \theta' - \mathbb{E}_{\pi_\theta}\left[\phi_\theta(\cdot, a')^\top \theta' - \phi_0(\cdot, a')^\top \theta'\right]\right\|_\sigma^2 \\
&\leq 2\|\phi_\theta(\cdot,\cdot)^\top \theta' - \phi_0(\cdot,\cdot)^\top \theta'\|_\sigma^2 + 2\|\phi_\theta(\cdot,\cdot)^\top \theta' - \phi_0(\cdot,\cdot)^\top \theta'\|_{\pi_\theta \cdot \nu}^2,
\end{aligned}
$$

where the second inequality follows from the Jensen's inequality and the fact that $\|x + y\|_2^2 \leq 2\|x\|_2^2 + 2\|y\|_2^2$. Therefore, by Assumption 4.2 and Lemma E.2, we obtain that

$$
\begin{aligned}
&\mathbb{E}_{\text{init}}\left[\|\overline{\phi}_\theta(\cdot,\cdot)^\top \theta' - \overline{\phi}_0(\cdot,\cdot)^\top \theta'\|_\sigma^2\right] \\
&\quad \leq 2\mathbb{E}_{\text{init}}\left[\|\phi_\theta(\cdot,\cdot)^\top \theta' - \phi_0(\cdot,\cdot)^\top \theta'\|_\sigma^2\right] + 2\mathbb{E}_{\text{init}}\left[\|\phi_\theta(\cdot,\cdot)^\top \theta' - \phi_0(\cdot,\cdot)^\top \theta'\|_{\pi_\theta \cdot \nu}^2\right] = \mathcal{O}(R^3 \cdot m^{-1/2}),
\end{aligned}
$$

which concludes the proof of Corollary E.3. $\qquad\square$

In what follows, we present a corollary that quantifies the difference between the function $\phi_{\widehat{\theta}}(\cdot,\cdot)^\top \theta$ and the two-layer neural network $f((\cdot,\cdot); \theta) = \phi_\theta(\cdot,\cdot)^\top \theta$ by the $L_2(\sigma)$-norm, where $\sigma(\cdot,\cdot) = \pi(\cdot\,|\,\cdot) \cdot \nu(\cdot)$ is the distribution over $\mathcal{S} \times \mathcal{A}$ such that Assumption 4.2 holds.

**Corollary E.4.** Let $\mathcal{B} = \{\alpha \in \mathbb{R}^{md} : \|\alpha - W_{\text{init}}\|_2 \leq R\}$. Under Assumption 4.2, it holds for all $\theta, \widehat{\theta} \in \mathcal{B}$ that

$$
\mathbb{E}_{\text{init}}\left[\|\phi_{\widehat{\theta}}(\cdot,\cdot)^\top \theta - \phi_\theta(\cdot,\cdot)^\top \theta\|_\sigma\right] = \mathcal{O}(R^{3/2} \cdot m^{-1/4}),
$$

where the expectation is taken over the random initialization. Here $\phi_\theta$ is the feature mapping defined in (3.3), and $\sigma(\cdot,\cdot) = \pi(\cdot\,|\,\cdot) \cdot \nu(\cdot)$ is the distribution over $\mathcal{S} \times \mathcal{A}$ such that Assumption 4.2 holds.

*Proof.* By the triangle inequality, we have

$$
\begin{aligned}
&\mathbb{E}_{\text{init}}\big[\|\phi_{\widehat{\theta}}(\cdot,\cdot)^\top\theta - \phi_\theta(\cdot,\cdot)^\top\theta\|_\sigma\big] \\
&\qquad \leq \mathbb{E}_{\text{init}}\big[\|\phi_{\widehat{\theta}}(\cdot,\cdot)^\top\theta - \phi_0(\cdot,\cdot)^\top\theta\|_\sigma\big] + \mathbb{E}_{\text{init}}\big[\|\phi_\theta(\cdot,\cdot)^\top\theta - \phi_0(\cdot,\cdot)^\top\theta\|_\sigma\big],
\end{aligned}
\tag{E.11}
$$

where $\phi_0$ is the feature mapping defined in (3.3) with $\theta = W_{\text{init}}$. Meanwhile, for all $\theta, \widehat{\theta} \in \mathcal{B} = \{\alpha \in \mathbb{R}^{md} : \|\alpha - W_{\text{init}}\|_2 \leq R\}$, it follows from Assumption 4.2 and Lemma E.2 that

$$
\begin{aligned}
\mathbb{E}_{\text{init}}\big[\|\phi_{\widehat{\theta}}(\cdot,\cdot)^\top\theta - \phi_0(\cdot,\cdot)^\top\theta\|_\sigma\big] &= \mathcal{O}(R^{3/2} \cdot m^{-1/4}), \\
\mathbb{E}_{\text{init}}\big[\|\phi_\theta(\cdot,\cdot)^\top\theta - \phi_0(\cdot,\cdot)^\top\theta\|_\sigma\big] &= \mathcal{O}(R^{3/2} \cdot m^{-1/4}),
\end{aligned}
\tag{E.12}
$$

where the expectations are taken over the random initialization. Combining (E.11) and (E.12), we obtain that

$$
\mathbb{E}_{\text{init}}\big[\|\phi_{\widehat{\theta}}(\cdot,\cdot)^\top\theta - \phi_\theta(\cdot,\cdot)^\top\theta\|_\sigma\big] = \mathcal{O}(R^{3/2} \cdot m^{-1/4}),
$$

which concludes the proof of Corollary E.4. $\qquad\square$

Corollary E.4 implies that when the width $m$ is sufficiently large, $\phi_{\widehat{\theta}}(\cdot,\cdot)^\top\theta$ is well approximated by the two-layer neural network $f((\cdot,\cdot);\theta)$ in $L_2(\sigma)$-norm, where $\sigma(\cdot,\cdot) = \pi(\cdot\,|\,\cdot) \cdot \nu(\cdot)$ is the distribution over $\mathcal{S} \times \mathcal{A}$ such that Assumption 4.2 holds.

# F  NEURAL TD

In this section, we introduce the details of neural TD (Cai et al., 2019) for critic update in Algorithm 1. Neural TD solves the optimization problem in (3.11) using the TD iterations defined in (3.12) and (3.13), which is summarized in Algorithm 2.

---

**Algorithm 2** Neural TD (Cai et al., 2019)

---

**Require:** The policy $\pi$, number of TD iterations $T_{\text{TD}}$, and learning rate $\eta_{\text{TD}}$ of neural TD.
1: **Initialization:**  Initialize $b_r \sim \text{Unif}(\{-1,1\})$ and $[W_{\text{init}}]_r \sim N(0, I_d/d)$. Set $\mathcal{B} \leftarrow \{\alpha \in \mathbb{R}^{md} : \|\alpha - W_{\text{init}}\|_2 \leq R\}$ and $\omega(0) \leftarrow W_{\text{init}}$.
2: **for** $t = 0, \ldots, T_{\text{TD}} - 1$ **do**
3: $\quad$ Sample a tuple $(s, a, r, s', a')$, where $(s,a) \sim \varsigma_i$, $s' \sim \mathcal{P}(\cdot\,|\,s,a)$, $r \leftarrow r(s,a)$, and $a' \sim \pi(\cdot\,|\,s')$.
4: $\quad$ Compute the Bellman residue $\delta \leftarrow Q_{\omega(t)}(s,a) - (1-\gamma) \cdot r - \gamma \cdot Q_{\omega(t)}(s',a')$.
5: $\quad$ Perform a TD update step: $\omega(t + 1/2) \leftarrow \omega(t) - \eta \cdot \delta \cdot \nabla_\omega Q_{\omega(t)}(s,a)$.
6: $\quad$ Perform a projection step: $\omega(t + 1) \leftarrow \Pi_{\mathcal{B}}(\omega(t + 1/2))$.
7: $\quad$ Perform an averaging step: $\overline{\omega} \leftarrow \frac{t+1}{t+2} \cdot \overline{\omega} + \frac{1}{t+2} \cdot \omega(t + 1)$.
8: **end for**
9: **Output:** $Q_{\text{out}}(\cdot) \leftarrow Q_{\overline{\omega}}(\cdot)$.

---

The following theorem by Cai et al. (2019) characterizes the rate of convergence of Algorithm 2.

**Theorem F.1** (Convergence of Neural TD (Cai et al., 2019)). We set $\eta_{\text{TD}} = \min\{(1 - \gamma)/8, 1/\sqrt{T_{\text{TD}}}\}$ in Algorithm 2. Under Assumption 4.2, it holds that

$$
\begin{aligned}
\mathbb{E}_{\text{init}}\big[\|Q_{\text{out}} - Q^\pi\|_{\varsigma_\pi}^2\big] \leq{}& 2\mathbb{E}_{\text{init}}\big[\|\Pi_{\widetilde{\mathcal{F}}_{R,m}} Q^\pi - Q^\pi\|_{\varsigma_\pi}^2\big] \\
&+ \mathcal{O}(R^2 \cdot T_{\text{TD}}^{-1/2} + R^3 \cdot m^{-1/2} + R^{5/2} \cdot m^{-1/4}),
\end{aligned}
\tag{F.1}
$$

where $\Pi_{\widetilde{\mathcal{F}}_{R,m}}$ is the projection operator onto $\widetilde{\mathcal{F}}_{R,m}$, and $\varsigma_\pi$ is the stationary state-action distribution corresponding to $\pi$.

*Proof.* See Proposition 4.7 in Cai et al. (2019) for a detailed proof. $\qquad\square$

## F.1 PROOF OF PROPOSITION 4.3

*Proof.* By Theorem F.1, to establish the rate of convergence of neural TD, it suffices to characterize the approximation error $\mathbb{E}_{\text{init}}[\|\Pi_{\widetilde{\mathcal{F}}_{R,m}} Q^\pi - Q^\pi\|_{\varsigma_\pi}^2]$ in (F.1). To this end, we first define a new function class

$$\overline{\mathcal{F}}_{R,m} = \left\{ \widehat{f}((s,a); W) = \frac{1}{\sqrt{m}} \cdot \sum_{r=1}^{m} b_r \cdot \mathbb{1}\big\{ [W_{\text{init}}]_r^\top (s,a) > 0 \big\} \cdot W_r^\top (s,a) \right.$$

$$\left. : \|[W]_r - [W_{\text{init}}]_r\|_\infty \leq R/\sqrt{md} \right\},$$

where $[W_{\text{init}}]_r \sim N(0, I_d/d)$ and $b_r \sim \text{Unif}(\{-1, 1\})$ are the initial parameters. By definition, $\overline{\mathcal{F}}_{R,m}$ is a subset of $\widetilde{\mathcal{F}}_{R,m}$ defined in Definition E.1. The following lemma obtained from Rahimi and Recht (2009) characterizes the deviation of $\overline{\mathcal{F}}_{R,m}$ from $\mathcal{F}_{R,\infty}$ given in Assumption 4.1.

**Lemma F.2** (Projection Error of $\overline{\mathcal{F}}_{R,m}$ (Rahimi and Recht, 2009)). *Let $f \in \mathcal{F}_{R,\infty}$, where $\mathcal{F}_{R,\infty}$ is defined in Assumption 4.1. For any $\delta > 0$, it holds with probability at least $1 - \delta$ that*

$$\|\Pi_{\overline{\mathcal{F}}_{R,m}} f - f\|_\varsigma \leq R \cdot m^{-1/2} \cdot \big[ 1 + \sqrt{2 \log(1/\delta)} \big], \tag{F.2}$$

*where $\varsigma$ is a distribution over $\mathcal{S} \times \mathcal{A}$.*

*Proof.* See Rahimi and Recht (2009) for a detailed proof. $\square$

Following from (F.2) in Lemma F.2, for all $f \in \mathcal{F}_{R,\infty}$ and $t > 0$, we have

$$\mathbb{P}\big( \|\Pi_{\overline{\mathcal{F}}_{R,m}} f - f\|_\varsigma \geq t \big) \leq \exp\big( -1/2 \cdot (t \cdot \sqrt{m}/R - 1)^2 \big). \tag{F.3}$$

Meanwhile, by Assumption 4.1, we have $Q^\pi \in \mathcal{F}_{R,\infty}$. Therefore, by setting $f = Q^\pi$ and $\varsigma = \varsigma_\pi$ in (F.3), we obtain that

$$\mathbb{E}_{\text{init}}\big[ \|\Pi_{\overline{\mathcal{F}}_{R,m}} Q^\pi - Q^\pi\|_{\varsigma_\pi}^2 \big] = \int_0^\infty \mathbb{P}\big( \|\Pi_{\overline{\mathcal{F}}_{R,m}} Q^\pi - Q^\pi\|_{\varsigma_\pi}^2 \geq t \big) \mathrm{d}t$$

$$\leq \int_0^\infty \exp\big( -1/2 \cdot (t \cdot \sqrt{m}/R - 1)^2 \big) \mathrm{d}t = \mathcal{O}(R \cdot m^{-1/2}), \tag{F.4}$$

where the expectation is taken over the random initialization. Also, note that $\overline{\mathcal{F}}_{R,m} \subseteq \widetilde{\mathcal{F}}_{R,m}$, where $\widetilde{\mathcal{F}}_{R,m}$ is defined in Definition E.1. Therefore, it follows from (F.4) that

$$\mathbb{E}_{\text{init}}\big[ \|\Pi_{\widetilde{\mathcal{F}}_{R,m}} Q^\pi - Q^\pi\|_{\varsigma_\pi}^2 \big] \leq \mathbb{E}_{\text{init}}\big[ \|\Pi_{\overline{\mathcal{F}}_{R,m}} Q^\pi - Q^\pi\|_{\varsigma_\pi}^2 \big] = \mathcal{O}(R \cdot m^{-1/2}). \tag{F.5}$$

Combining (F.5) and Theorem F.1, we obtain for $\eta_{\text{TD}} = \min\{(1-\gamma)/8, 1/\sqrt{T_{\text{TD}}}\}$ that

$$\mathbb{E}_{\text{init}}\big[ \|Q_{\text{out}} - Q^\pi\|_{\sigma_\pi}^2 \big] = \mathcal{O}(R \cdot m^{-1/2} + R^2 \cdot T_{\text{TD}}^{-1/2} + R^3 \cdot m^{-1/2} + R^{5/2} \cdot m^{-1/4}). \tag{F.6}$$

Specifically, $Q_{\omega_i}$ is the output of Algorithm 2 with $\pi_{\theta_i}$ as the input. Finally, by setting $T_{\text{TD}} = \Omega(m)$ in (F.6), we obtain

$$\mathbb{E}_{\text{init}}\big[ \|Q_{\omega_i} - Q^{\pi_{\theta_i}}\|_{\varsigma_i}^2 \big] = \mathcal{O}(R^3 \cdot m^{-1/2} + R^{5/2} \cdot m^{-1/4}),$$

which concludes the proof of Proposition 4.3. $\square$

## G PROJECTION-FREE NEURAL POLICY GRADIENT

In this section, we study the convergence of neural policy gradient where we do not impose the projection in the actor update. Specifically, the projection-free actor update takes the form of

$$\theta_{i+1} \leftarrow \theta_i + \eta \cdot \widetilde{\nabla}_\theta J(\pi_{\theta_i}).$$

Here $\widetilde{\nabla}_\theta J(\pi_{\theta_i})$ is an estimator of the policy gradient $\nabla_\theta J(\pi_{\theta_i})$, which takes the form of

$$\widetilde{\nabla}_\theta J(\pi_{\theta_i}) = \frac{\tau_i}{B} \cdot \sum_{\ell=1}^{B} \widetilde{Q}_{\omega_i}(s_\ell, a_\ell) \cdot \nabla_\theta \log \pi_{\theta_i}(a_\ell \,|\, s_\ell). \tag{G.1}$$

Here $\tau_i$ is the temperature parameter of $\pi_{\theta_i}$, $\{(s_\ell, a_\ell)\}_{\ell \in [B]}$ is sampled from the state-action visitation measure $\sigma_i$ corresponding to the current policy $\pi_{\theta_i}$, and $B > 0$ is the batch size. Also, $\widetilde{Q}_{\omega_i}$ is the modified critic. Specifically, for all $(s, a) \in \mathcal{S} \times \mathcal{A}$, we define

$$\begin{aligned}
\widetilde{Q}_{\omega_i}(s, a) = &\, Q_{\max} \cdot \mathbb{1}\{Q_{\omega_i}(s, a) \geq Q_{\max}\} - Q_{\max} \cdot \mathbb{1}\{Q_{\omega_i}(s, a) \leq -Q_{\max}\} \\
&+ Q_{\omega_i}(s, a) \cdot \mathbb{1}\{-Q_{\max} < Q_{\omega_i}(s, a) < Q_{\max}\},
\end{aligned} \tag{G.2}$$

where $Q_{\omega_i}$ is obtained from Algorithm 2 with $\pi_{\theta_i}$ as the input. We summarize projection-free neural policy gradient in Algorithm 3.

---

**Algorithm 3** Projection-Free Neural Policy Gradient

---

**Require:** Number of iterations $T$, number of TD iterations $T_{\text{TD}}$, learning rate $\eta$, learning rate $\eta_{\text{TD}}$ of neural TD, temperature parameters $\{\tau_i\}_{i \in [T+1]}$, and batch size $B$.
1: **Initialization:** Initialize $b_r \sim \text{Unif}(\{-1, 1\})$ and $[W_{\text{init}}]_r \sim N(0, I_d/d)$ for all $r \in [m]$. Set $\mathcal{B} \leftarrow \{\alpha \in \mathbb{R}^{md} : \|\alpha - W_{\text{init}}\|_2 \leq R\}$ and $\theta_1 \leftarrow W_{\text{init}}$.
2: **for** $i \in [T]$ **do**
3:     Update $\omega_i$ using Algorithm 2 with $\pi_{\theta_i}$ as the input, $\omega(0) \leftarrow W_{\text{init}}$ and $\{b_r\}_{r \in [m]}$ as the initialization, $T_{\text{TD}}$ as the number of iterations, and $\eta_{\text{TD}}$ as the learning rate.
4:     Sample $\{(s_\ell, a_\ell)\}_{\ell \in [B]}$ from the visitation measure $\sigma_i$, and estimate $\widetilde{\nabla}_\theta J(\pi_\theta)$ using (G.1) and (G.2).
5:     Update $\theta_{i+1}$ by $\theta_{i+1} \leftarrow \theta_i + \eta \cdot \widetilde{\nabla}_\theta J(\pi_{\theta_i})$.
6: **end for**
7: **Output:** $\{\pi_{\theta_i}\}_{i \in [T+1]}$.

---

### G.1 CONVERGENCE OF PROJECTION-FREE NEURAL POLICY GRADIENT

In this section, we show that the sequence $\{\theta_i\}_{i \in [T+1]}$ generated by projection-free neural policy gradient converges to a stationary point at a sublinear rate. In parallel to Assumption 4.4, we lay out the following regularity condition on the moments of the estimator $\widetilde{\nabla}_\theta J(\pi_{\theta_i})$.

**Assumption G.1** (Moment Upper Bound). Recall that $\sigma_i$ is the state-action visitation measure corresponding to $\pi_{\theta_i}$ for all $i \in [T]$. Let $\widetilde{\xi}_i = \widetilde{\nabla}_\theta J(\pi_{\theta_i}) - \mathbb{E}[\widetilde{\nabla}_\theta J(\pi_{\theta_i})]$, where $\widetilde{\nabla}_\theta J(\pi_{\theta_i})$ is defined in (G.1). We assume that there exists absolute constants $\sigma_{\widetilde{\xi}}, \varsigma_{\widetilde{\xi}} > 0$ such that $\mathbb{E}[\|\widetilde{\xi}_i\|_2^2] \leq \tau_i^2 \cdot \sigma_{\widetilde{\xi}}^2 / B$ and $\mathbb{E}[\|\widetilde{\xi}_i\|_2^3] \leq \tau_i^3 \cdot \varsigma_{\widetilde{\xi}}^3 / B^{3/2}$ for all $i \in [T]$. Here the expectations are taken over $\sigma_i$ given $\theta_i$ and $\omega_i$.

Similar to Theorem 4.7, in the following theorem, we show that the sequence $\{\theta_i\}_{i \in [T+1]}$ generated by Algorithm 3 converges to a stationary point $\widehat{\theta}$ with $\nabla_\theta J(\pi_{\widehat{\theta}}) = 0$ at a sublinear rate.

**Theorem G.2** (Convergence to Stationary Point). Let $\eta = 1/\sqrt{T}$, $\tau_i = 1$, $\eta_{\text{TD}} = \min\{(1 - \gamma)/8, 1/\sqrt{T_{\text{TD}}}\}$, and $T_{\text{TD}} = \Omega(m)$ in Algorithm 3. Under the assumptions of Proposition 4.3 and Assumptions 4.5, 4.6, and G.1, it holds for $T \geq 4L^2$ and $B = \Omega(\sigma_{\widetilde{\xi}}^2 \cdot T^{1/2})$ that

$$\min_{i \in [T]} \mathbb{E}\big[\|\nabla_\theta J(\pi_{\theta_i})\|_2^2\big] \leq 8/\sqrt{T} \cdot \mathbb{E}\big[J(\pi_{\theta_{T+1}}) - J(\pi_{\theta_1})\big] + \epsilon_{\text{PG}},$$

where

$$\epsilon_{\text{PG}} = \mathcal{O}(T^{-1/2} + R^{3/2} \cdot m^{-1/4} \cdot T + R^{5/4} \cdot m^{-1/8} \cdot T).$$

Here the expectations are taken over all the randomness.

*Proof.* Our proof aligns closely to that of Theorem 4.7 in §D.1. We first lower bound the difference $J(\pi_{\theta_{i+1}}) - J(\pi_{\theta_i})$. By Assumption 4.6, we have

$$J(\pi_{\theta_{i+1}}) - J(\pi_{\theta_i}) \geq \eta \cdot \nabla_\theta J(\pi_{\theta_i})^\top \delta_i - L/2 \cdot \|\theta_{i+1} - \theta_i\|_2^2, \tag{G.3}$$

where

$$\delta_i = (\theta_{i+1} - \theta_i)/\eta = \widetilde{\nabla}_\theta J(\pi_{\theta_i}), \quad \forall i \in [T].$$

Following the proof of Lemma D.2 in §H.5, we obtain that

$$\left| \left( \nabla_\theta J(\pi_{\theta_i}) - \mathbb{E}\big[\widetilde{\nabla}_\theta J(\pi_{\theta_i})\big] \right)^\top \delta_i \right| \leq \kappa/\eta \cdot 2\|\theta_{i+1} - \theta_i\|_2 \cdot \|Q^{\pi_{\theta_i}} - \widetilde{Q}_{\omega_i}\|_{\varsigma_i}, \tag{G.4}$$

where the expectation is taken over $\sigma_i$ given $\theta_i$ and $\omega_i$. Recall that $\widetilde{\xi}_i = \widetilde{\nabla}_\theta J(\pi_{\theta_i}) - \mathbb{E}\big[\widetilde{\nabla}_\theta J(\pi_{\theta_i})\big]$, where the expectation is taken over $\sigma_i$ given $\theta_i$ and $\omega_i$. Following from (G.4), we obtain that

$$\nabla_\theta J(\pi_{\theta_i})^\top \delta_i = \left( \nabla_\theta J(\pi_{\theta_i}) - \mathbb{E}\big[\widetilde{\nabla}_\theta J(\pi_{\theta_i})\big] \right)^\top \delta_i - (\widetilde{\xi}_i)^\top \delta_i + \widetilde{\nabla}_\theta J(\pi_{\theta_i})^\top \delta_i$$

$$\geq -2\kappa \cdot \|\theta_{i+1} - \theta_i\|_2/\eta \cdot \|Q^{\pi_{\theta_i}} - \widetilde{Q}_{\omega_i}\|_{\varsigma_i} - \|\widetilde{\xi}_i\|_2^2/2 + \|\delta_i\|_2^2/2, \tag{G.5}$$

where the second inequality follows similar analysis to §D.1. Hence, by plugging (G.5) into (G.3), we have

$$J(\pi_{\theta_{i+1}}) - J(\pi_{\theta_i})$$

$$\geq (\eta - L \cdot \eta^2)/2 \cdot \|\delta_i\|_2^2 - \eta \cdot \|\widetilde{\xi}_i\|_2^2/2 - 2\kappa \cdot \|\theta_{i+1} - \theta_i\|_2 \cdot \|Q^{\pi_{\theta_i}} - \widetilde{Q}_{\omega_i}\|_{\varsigma_i}. \tag{G.6}$$

It remains to upper bound $\|\theta_{i+1} - \theta_i\|_2$. To this end, we use the fact that

$$\|\theta_{i+1} - \theta_i\|_2 \leq \|\theta_i - W_{\text{init}}\|_2 + \|\theta_{i+1} - W_{\text{init}}\|_2,$$

and upper bound $\|\theta_i - W_{\text{init}}\|_2$ and $\|\theta_{i+1} - W_{\text{init}}\|_2$. By the actor update in Algorithm 3, we obtain for all $i > 1$ that

$$\|\theta_i - W_{\text{init}}\|_2 \leq \sum_{j=1}^{i-1} \eta \cdot \|\widetilde{\nabla}_\theta J(\pi_{\theta_j})\|_2 \leq \sum_{j=1}^{i-1} \eta \cdot \left( \left\|\mathbb{E}\big[\widetilde{\nabla}_\theta J(\pi_{\theta_j})\big]\right\|_2 + \|\widetilde{\xi}_j\|_2 \right), \tag{G.7}$$

where the expectation is taken over $\sigma_i$ given $\theta_i$ and $\omega_i$. Meanwhile, it holds that

$$\left\|\mathbb{E}\big[\widetilde{\nabla}_\theta J(\pi_{\theta_j})\big]\right\|_2 = \left\|\mathbb{E}_{\sigma_j}\big[\overline{\phi}_{\theta_j}(s,a) \cdot \widetilde{Q}_{\omega_j}(s,a)\big]\right\|_2 \leq \mathbb{E}_{\sigma_j}\big[\|\overline{\phi}_{\theta_j}(s,a)\|_2 \cdot |\widetilde{Q}_{\omega_j}(s,a)|\big], \tag{G.8}$$

where $\overline{\phi}_{\theta_j}$ is the centered feature mapping defined in (B.2), and the last inequality follows from the Jensen's inequality. We now upper bound the right-hand side of (G.8). Note that $\|\overline{\phi}_{\theta_j}(s,a)\|_2 \leq 2$ for all $(s,a) \in \mathcal{S} \times \mathcal{A}$. Meanwhile, by (G.2), we obtain that

$$|\widetilde{Q}_{\omega_j}(s,a)| \leq Q_{\max}, \quad \forall (s,a) \in \mathcal{S} \times \mathcal{A}. \tag{G.9}$$

By plugging (G.9) into (G.8), we obtain for all $j \in [T]$ that

$$\left\|\mathbb{E}\big[\widetilde{\nabla}_\theta J(\pi_{\theta_j})\big]\right\|_2 \leq 2Q_{\max}. \tag{G.10}$$

By further plugging (G.10) into (G.7), we obtain for all $i > 1$ that

$$\|\theta_i - W_{\text{init}}\|_2 \leq 2Q_{\max} \cdot \eta \cdot T + \sum_{j=1}^{i-1} \eta \cdot \|\widetilde{\xi}_j\|_2. \tag{G.11}$$

We now lower bound the right-hand side of (G.6) based on (G.11). Following from the Cauchy-Schwartz inequality and Assumption G.1, we obtain that

$$\mathbb{E}\big[\|\theta_i - W_{\text{init}}\|_2 \cdot \|Q^{\pi_{\theta_i}} - \widetilde{Q}_{\omega_i}\|_{\varsigma_i}\big]$$

$$\leq 2Q_{\max} \cdot \eta \cdot T \cdot \left\{\mathbb{E}\big[\|Q^{\pi_{\theta_i}} - \widetilde{Q}_{\omega_i}\|_{\varsigma_i}^2\big]\right\}^{1/2}$$

$$+ \sum_{j=1}^{i-1} \eta \cdot \left\{\mathbb{E}\big[\|\widetilde{\xi}_i\|_2^2\big]\right\}^{1/2} \cdot \left\{\mathbb{E}\big[\|Q^{\pi_{\theta_i}} - \widetilde{Q}_{\omega_i}\|_{\varsigma_i}^2\big]\right\}^{1/2}$$

$$\leq (2Q_{\max} \cdot \eta \cdot T + \sigma_{\widetilde{\xi}} \cdot \eta \cdot T \cdot B^{-1/2}) \cdot \left\{\mathbb{E}\big[\|Q^{\pi_{\theta_i}} - \widetilde{Q}_{\omega_i}\|_{\varsigma_i}^2\big]\right\}^{1/2}, \tag{G.12}$$

where the expectations are taken over all the randomness, and $\sigma_{\widetilde{\xi}}$ is the absolute constant defined in Assumptions G.1. By plugging (G.12) into (G.6), we obtain that

$$(\eta - L \cdot \eta^2)/2 \cdot \mathbb{E}\big[\|\delta_i\|_2^2\big]$$
$$\leq \mathbb{E}\big[J(\pi_{\theta_{i+1}}) - J(\pi_{\theta_i})\big] + \eta \cdot \sigma_{\widetilde{\xi}}^2/(2B) + R_0(T) \cdot \Big\{\mathbb{E}\big[\|Q^{\pi_{\theta_i}} - Q_{\omega_i}\|_{\varsigma_i}^2\big]\Big\}^{1/2}, \qquad \text{(G.13)}$$

where we use the fact that $\|\theta_{i+1} - \theta_i\|_2 \leq \|\theta_{i+1} - W_{\text{init}}\|_2 + \|\theta_i - W_{\text{init}}\|_2$. Here the expectations are taken over all the randomness, and $R_0(T)$ is defined by

$$R_0(T) = 4Q_{\max} \cdot \eta \cdot T + 2\sigma_{\widetilde{\xi}} \cdot \eta \cdot T \cdot B^{-1/2}.$$

By Proposition 4.3 and Assumption 4.2, we obtain for $\eta = 1/\sqrt{T}$, $B = \Omega(\sigma_{\widetilde{\xi}}^2 \cdot T^{1/2})$, and $T_{\text{TD}} = \Omega(m)$ that

$$R_0(T) = \mathcal{O}(\sqrt{T}), \qquad \mathbb{E}\big[\|Q^{\pi_{\theta_i}} - \widetilde{Q}_{\omega_i}\|_{\varsigma_i}^2\big] \leq \mathbb{E}\big[\|Q^{\pi_{\theta_i}} - Q_{\omega_i}\|_{\varsigma_i}^2\big]$$
$$= \mathcal{O}(R^3 \cdot m^{-1/2} + R^{5/2} \cdot m^{-1/4}), \qquad \text{(G.14)}$$

where the inequality holds since $|Q^{\pi_{\theta_i}}(s,a)| \leq Q_{\max}$ for all $(s,a) \in \mathcal{S} \times \mathcal{A}$. By plugging (G.14) into (G.13) with $\eta = 1/\sqrt{T}$ and $B = \Omega(\sigma_{\widetilde{\xi}}^2 \cdot T^{1/2})$, we obtain that

$$(1 - L/\sqrt{T})/2 \cdot \mathbb{E}\big[\|\delta_i\|_2^2\big] \leq \sqrt{T} \cdot \mathbb{E}\big[J(\pi_{\theta_{i+1}}) - J(\pi_{\theta_i})\big] + \epsilon_{\text{PG}}, \qquad \text{(G.15)}$$

where

$$\epsilon_{\text{PG}} = \mathcal{O}(T^{-1/2} + R^{3/2} \cdot m^{-1/4} \cdot T + R^{5/4} \cdot m^{-1/8} \cdot T). \qquad \text{(G.16)}$$

It remains to upper bound $\|\delta_i - \nabla_\theta J(\pi_{\theta_i})\|_2$, where $\delta_i = \widetilde{\nabla} J(\pi_{\theta_i})$. Following from similar analysis to §H.6, we obtain that

$$\mathbb{E}\big[\|\nabla_\theta J(\pi_{\theta_i}) - \widetilde{\nabla}_\theta J(\pi_{\theta_i})\|_2^2\big] \leq 2\mathbb{E}\big[\|\widetilde{\xi}_i\|_2^2\big] + 8\kappa^2 \cdot \mathbb{E}\big[\|Q^{\pi_{\theta_i}} - \widetilde{Q}_{\omega_i}\|_{\varsigma_i}^2\big],$$

where the expectations are taken over all the randomness. Therefore, following from Proposition 4.3 and Assumption G.1, it holds for $\eta = 1/\sqrt{T}$, $B = \Omega(\sigma_{\widetilde{\xi}}^2 \cdot T^{1/2})$, and $T_{\text{TD}} = \Omega(m)$ that

$$\mathbb{E}\big[\|\nabla_\theta J(\pi_{\theta_i}) - \widetilde{\nabla}_\theta J(\pi_{\theta_i})\|_2^2\big] = \mathcal{O}(T^{-1/2} + R^3 \cdot m^{-1/2} + R^{5/2} \cdot m^{-1/4}). \qquad \text{(G.17)}$$

Thus, combining (G.15) and (G.17), we obtain for all $i \in [T]$ that

$$\mathbb{E}\big[\|\nabla_\theta J(\pi_{\theta_i})\|_2^2\big] \leq 2\mathbb{E}\big[\|\delta_i\|_2^2\big] + 2\mathbb{E}\big[\|\nabla_\theta J(\pi_{\theta_i}) - \widetilde{\nabla}_\theta J(\pi_{\theta_i})\|_2^2\big]$$
$$\leq 4(1 - L/\sqrt{T}) \cdot \mathbb{E}\big[\|\delta_i\|_2^2\big] + 2\mathbb{E}\big[\|\nabla_\theta J(\pi_{\theta_i}) - \widetilde{\nabla}_\theta J(\pi_{\theta_i})\|_2^2\big]$$
$$\leq 8\sqrt{T} \cdot \mathbb{E}\big[J(\pi_{\theta_{i+1}}) - J(\pi_{\theta_i})\big] + \epsilon_{\text{PG}}, \qquad \text{(G.18)}$$

where we use the fact that $T \geq 4L^2$ and we define $\epsilon_{\text{PG}}$ in (G.16). Finally, by telescoping (G.18), we obtain that

$$\min_{i \in [T]} \mathbb{E}\big[\|\nabla_\theta J(\pi_{\theta_i})\|_2^2\big] \leq \frac{1}{T} \cdot \sum_{i=1}^T \mathbb{E}\big[\|\nabla_\theta J(\pi_{\theta_i})\|_2^2\big] \leq 8\mathbb{E}\big[J(\pi_{\theta_{T+1}}) - J(\pi_{\theta_1})\big]/\sqrt{T} + \epsilon_{\text{PG}},$$

where

$$\epsilon_{\text{PG}} = \mathcal{O}(T^{-1/2} + R^{3/2} \cdot m^{-1/4} \cdot T + R^{5/4} \cdot m^{-1/8} \cdot T).$$

Here the expectations are taken over all the randomness. Thus, we complete the proof of Theorem G.2. $\qquad\square$

Following from Theorem G.2, it holds for $m = \Omega(R^{10} \cdot T^{12})$ that

$$\min_{i \in [T]} \mathbb{E}\big[\|\nabla_\theta J(\pi_{\theta_i})\|_2^2\big] = \mathcal{O}(1/\sqrt{T}).$$

Therefore, $\theta_i$ converges to a stationary point at a $1/\sqrt{T}$-rate if the width $m$ of the two-layer neural network and the batch size $B$ are sufficiently large. We highlight that compared with neural policy gradient with projection in the actor update, Algorithm 3 needs a larger width $m$ to achieve the $1/\sqrt{T}$-rate of convergence. Such a stronger requirement on $m$ is the extra price to pay for using the projection-free actor update.

### G.2 GLOBAL OPTIMALITY OF PROJECTION-FREE NEURAL POLICY GRADIENT

In this section, we characterize the global optimality of projection-free neural policy gradient. We define a sequence of parameter spaces $\{\mathcal{B}_i\}_{i\in[T]}$ as follows,

$$\mathcal{B}_i = \big\{\alpha \in \mathbb{R}^{md} : \|\alpha - \theta_i\|_2 \leq \overline{R}_0\big\}, \quad \forall i \in [T], \tag{G.19}$$

where $\overline{R}_0 \geq 1$ is an absolute constant. The sequence $\{\mathcal{B}_i\}_{i\in[T]}$ characterizes the global optimality of the parameter sequence $\{\theta_i\}_{i\in[T]}$. Specifically, similar to (4.5), we have

$$\nabla_\theta J(\pi_{\theta_i})^\top (\theta - \theta_i) \leq \|\theta - \theta_i\|_2 \cdot \|\nabla_\theta J(\pi_{\theta_i})\|_2 \leq \overline{R}_0 \cdot \|\nabla_\theta J(\pi_{\theta_i})\|_2, \quad \forall \theta \in \mathcal{B}_i, \, \forall i \in [T],$$

where the first inequality follows from the Cauchy-Schwartz inequality. Following similar analysis to §D.2, we obtain for all $i \in [T]$ that

$$(1 - \gamma) \cdot \big(J(\pi^*) - J(\pi_{\theta_i})\big)$$
$$\leq 2Q_{\max} \cdot \inf_{\theta \in \mathcal{B}_i} \|u_{\theta_i}(\cdot, \cdot) - \phi_{\theta_i}(\cdot, \cdot)^\top \theta\|_{\sigma_i} + \overline{R}_0 \cdot \|\nabla_\theta J(\pi_{\theta_i})\|_2. \tag{G.20}$$

We now introduce the parameter space $\overline{\mathcal{B}}_T$ that includes the sequence $\{\theta_i\}_{i\in[T]}$ and the parameter space $\mathcal{B}_i$ as its subspace for all $i \in [T]$ as follows,

$$\overline{\mathcal{B}}_T = \big\{\alpha \in \mathbb{R}^{md} : \|\alpha - W_{\text{init}}\|_2 \leq R(T) + \overline{R}_0\big\}, \tag{G.21}$$

where

$$R(T) = 2Q_{\max} \cdot \eta \cdot T + \eta \cdot \sum_{i=1}^{T} \|\widetilde{\xi}_i\|_2. \tag{G.22}$$

Here $\widetilde{\xi}_i$ is defined in Assumption G.1. Following from (G.7) and (G.10) in the proof of Theorem G.2 in §G.1, we have $\theta_i \in \overline{\mathcal{B}}_T$ for all $i \in [T]$. By Corollary E.4, $\phi_{\theta_i}(\cdot, \cdot)^\top \theta$ is well approximated by $f((\cdot, \cdot); \theta)$ for $\theta, \theta_i \in \overline{\mathcal{B}}_T$ when the width $m$ is sufficiently large. Thus, following from (G.20), for a sufficiently large $m$, the suboptimality of $\theta_i$ is characterized by $\|\nabla_\theta J(\pi_{\theta_i})\|_2$, which is further quantified by Theorem G.2, and the approximation error $\inf_{\theta \in \mathcal{B}_i} \|u_{\theta_i}(\cdot, \cdot) - f((\cdot, \cdot); \theta)\|_{\sigma_i}$, which quantifies the representation power of the overparameterized two-layer neural networks. In the following theorem, we present a sufficient condition for the output of projection-free neural policy gradient to be globally optimal.

**Theorem G.3** (Global Optimality of Projection-Free Neural Policy Gradient)**.** Let $\eta = 1/\sqrt{T}$, $\tau_i = 1$, $\eta_{\text{TD}} = \min\{(1-\gamma)/8, 1/\sqrt{T_{\text{TD}}}\}$, and $T_{\text{TD}} = \Omega(m)$ in Algorithm 3. We define

$$\widetilde{u}_{\theta_i}(s, a) = u_{\theta_i}(s, a) + \phi_{\theta_i}(s, a)^\top (W_{\text{init}} - \theta_i), \quad \forall (s, a) \in \mathcal{S} \times \mathcal{A}.$$

Here $u_{\theta_i}$ is defined in (4.4) of Theorem 4.8 with $\widehat{\theta} = \theta_i$, and $\phi_{\theta_i}$ is the feature mapping defined in (3.3) with $\theta = \theta_i$. Under the assumptions of Theorem G.2, if it holds that

$$\widetilde{u}_{\theta_i} \in \mathcal{F}_{\overline{R}_0, \infty}, \quad \forall i \in [T],$$

then for $T \geq 4L^2$, $B = \Omega(T^{1/2})$, and $m = \Omega(R^{10} \cdot T^{12})$, we have

$$(1 - \gamma) \cdot \min_{i \in [T]} \mathbb{E}\big[J(\pi^*) - J(\pi_{\theta_i})\big] = \mathcal{O}(\overline{R}_0 \cdot T^{-1/4}).$$

Here the expectation is taken over all the randomness.

*Proof.* To prove Theorem G.3, it suffices to upper bound the expectation of the right-hand side of (G.20) over all the randomness. We first upper bound the following term,

$$\mathbb{E}\Big[\inf_{\theta \in \mathcal{B}_i} \|u_{\theta_i}(\cdot, \cdot) - \phi_{\theta_i}(\cdot, \cdot)^\top \theta\|_{\sigma_i}\Big],$$

where the expectation is taken over all the randomness. Note that

$$u_{\theta_i}(s, a) - \phi_{\theta_i}(s, a)^\top \theta = \widetilde{u}_{\theta_i}(s, a) + \phi_{\theta_i}(s, a)^\top \theta_i - \phi_{\theta_i}(s, a)^\top W_{\text{init}} - \phi_{\theta_i}(s, a)^\top \theta$$
$$= \widetilde{u}_{\theta_i}(s, a) - \phi_0(s, a)^\top (\theta - \theta_i + W_{\text{init}}) \tag{G.23}$$
$$\quad - \big(\phi_{\theta_i}(s, a) - \phi_0(s, a)\big)^\top (\theta - \theta_i + W_{\text{init}}),$$

which holds for all $(s, a) \in \mathcal{S} \times \mathcal{A}$ and $\theta \in \mathcal{B}_i$ with $\mathcal{B}_i$ defined in (G.19). Therefore, by the triangle inequality, we obtain from (G.23) that

$$\inf_{\theta \in \mathcal{B}_i} \|u_{\theta_i}(\cdot, \cdot) - \phi_{\theta_i}(\cdot, \cdot)^\top \theta\|_{\sigma_i} \leq \inf_{\theta \in \mathcal{B}_i} \left\{ \|\widetilde{u}_{\theta_i}(\cdot, \cdot) - \phi_0(\cdot, \cdot)^\top (\theta - \theta_i + W_{\text{init}})\|_{\sigma_i} \right. \tag{G.24}$$
$$\left. + \|(\phi_{\theta_i}(\cdot, \cdot) - \phi_0(\cdot, \cdot))^\top (\theta - \theta_i + W_{\text{init}})\|_{\sigma_i} \right\}.$$

We now upper bound the right-hand side of (G.24). In what follows, we define $\widetilde{\theta}_i$ by

$$\phi_0(\cdot, \cdot)^\top \widetilde{\theta}_i = \Pi_{\widetilde{\mathcal{F}}_{\overline{R}_0, m}} \widetilde{u}_{\theta_i}(\cdot, \cdot),$$

where $\Pi_{\widetilde{\mathcal{F}}_{\overline{R}_0, m}}$ is the projection operator onto $\widetilde{\mathcal{F}}_{\overline{R}_0, m}$. It then follows from the definition of $\widetilde{\mathcal{F}}_{\overline{R}_0, m}$ in Definition E.1 that $\widetilde{\theta}_i \in \mathcal{B}_1 = \{\alpha \in \mathbb{R}^{md} : \|\alpha - W_{\text{init}}\|_2 \leq \overline{R}_0\}$ for all $i \in [T]$. Meanwhile, by the definition of $\mathcal{B}_i$ in (G.19), we have

$$\widetilde{\theta}_i + \theta_i - W_{\text{init}} \in \mathcal{B}_i, \quad \forall i \in [T]. \tag{G.25}$$

Combining (G.24) and (G.25), we have

$$\inf_{\theta \in \mathcal{B}_i} \|u_{\theta_i}(\cdot, \cdot) - \phi_{\theta_i}(\cdot, \cdot)^\top \theta\|_{\sigma_i} \leq \|\widetilde{u}_{\theta_i}(\cdot, \cdot) - \phi_0(\cdot, \cdot)^\top \widetilde{\theta}_i\|_{\sigma_i} + \|(\phi_{\theta_i}(\cdot, \cdot) - \phi_0(\cdot, \cdot))^\top \widetilde{\theta}_i\|_{\sigma_i}. \tag{G.26}$$

Now, it suffices to upper bound the right-hand side of (G.26). Following from the proof of Proposition 4.3 in §F.1, we obtain for $\widetilde{u}_{\theta_i} \in \mathcal{F}_{\overline{R}_0, \infty}$ that

$$\mathbb{E}[\|\widetilde{u}_{\theta_i}(\cdot, \cdot) - \phi_0(\cdot, \cdot)^\top \widetilde{\theta}_i\|_{\sigma_i}] = \mathbb{E}[\|\widetilde{u}_{\theta_i}(\cdot, \cdot) - \Pi_{\widetilde{\mathcal{F}}_{\overline{R}_0, m}} \widetilde{u}_{\theta_i}(\cdot, \cdot)\|_{\sigma_i}] = \mathcal{O}(\overline{R}_0 \cdot m^{-1/2}), \tag{G.27}$$

where the expectations are taken over all the randomness. Meanwhile, note that

$$\|\widetilde{\theta}_i - W_{\text{init}}\|_2 \leq \overline{R}_0 \leq \overline{R}_0 + R(T),$$

where $R(T)$ is defined in (G.22). Therefore, we obain that $\theta_i, \widetilde{\theta}_i \in \overline{\mathcal{B}}_T$. By Assumption G.1, we obtain for $\eta = 1/\sqrt{T}$ and $B = \Omega(T^{1/2})$ that

$$\mathbb{E}[R(T)^2] = \mathcal{O}(T), \qquad \mathbb{E}[R(T)^3] = \mathcal{O}(T^{3/2}), \tag{G.28}$$

where the expectations are taken over all the randomness given $W_{\text{init}}$. Thus, following from (G.28), Assumption 4.2, and Lemma E.2, we obtain for all $\theta_i, \widetilde{\theta}_i \in \overline{\mathcal{B}}_T$ that

$$\mathbb{E}[\|\phi_0(\cdot, \cdot)^\top \widetilde{\theta}_i - \phi_{\theta_i}(\cdot, \cdot)^\top \widetilde{\theta}_i\|_{\sigma_i}]$$
$$\leq \left\{ \mathbb{E}[\|\phi_0(\cdot, \cdot)^\top \widetilde{\theta}_i - \phi_{\theta_i}(\cdot, \cdot)^\top \widetilde{\theta}_i\|_{\sigma_i}^2] \right\}^{1/2} = \mathcal{O}(T^{3/4} \cdot m^{-1/4}). \tag{G.29}$$

By plugging (G.27) and (G.29) into (G.26), we have

$$\mathbb{E}\left[ \inf_{\theta \in \mathcal{B}_i} \|u_{\theta_i}(\cdot, \cdot) - \phi_{\theta_i}(\cdot, \cdot)^\top \theta\|_{\sigma_i} \right]$$
$$\leq \mathbb{E}[\|\widetilde{u}_{\theta_i}(\cdot, \cdot) - \phi_0(\cdot, \cdot)^\top \widetilde{\theta}_i\|_{\sigma_i}] + \mathbb{E}[\|\phi_0(\cdot, \cdot)^\top \widetilde{\theta}_i - \phi_{\theta_i}(\cdot, \cdot)^\top \widetilde{\theta}_i\|_{\sigma_i}]$$
$$= \mathcal{O}(\overline{R}_0 \cdot m^{-1/2} + T^{3/4} \cdot m^{-1/4}), \tag{G.30}$$

which holds for all $i \in [T]$.

Meanwhile, by Theorem G.2, we obtain for $B = \Omega(T^{1/2})$ and $m = \Omega(R^{10} \cdot T^{12})$ that

$$\min_{i \in [T]} \mathbb{E}[\|\nabla_\theta J(\pi_{\theta_i})\|_2] = \mathcal{O}(T^{-1/4}). \tag{G.31}$$

Thus, by plugging (G.30) and (G.31) with $m = \Omega(R^{10} \cdot T^{12})$ into (G.20), we complete the proof of Theorem G.3. □

By Theorem G.3, it holds for sufficiently large width $m$ and batch size $B$ that the expected total reward $J(\pi_{\theta_i})$ converges to the global optimum $J(\pi^*)$ at a $1/T^{1/4}$-rate.

# H    PROOF OF AUXILIARY RESULTS

In this section, we lay out the proof of the auxiliary results.

## H.1    PROOF OF PROPOSITION 3.1

*Proof.* The proof is based on the policy gradient theorem (Sutton and Barto, 2018) in (2.5) and the definition of the Fisher information matrix in (2.7). It suffices to calculate $\nabla_\theta \log \pi_\theta(\cdot \,|\, \cdot)$. By the definition of $\pi_\theta(\cdot \,|\, \cdot)$ in (3.2), it holds for all $(s, a) \in \mathcal{S} \times \mathcal{A}$ that

$$
\nabla_\theta \log \pi_\theta(a \,|\, s) = \tau \cdot \nabla_\theta f\big((s, a); \theta\big) - \tau \cdot \frac{\sum_{a' \in \mathcal{A}} \nabla_\theta f\big((s, a'); \theta\big) \cdot \exp\big[\tau \cdot f\big((s, a'); \theta\big)\big]}{\sum_{a' \in \mathcal{A}} \exp\big[\tau \cdot f\big((s, a'); \theta\big)\big]}
$$

$$
= \tau \cdot \nabla_\theta f\big((s, a); \theta\big) - \tau \cdot \mathbb{E}_{\pi_\theta}\big[\nabla_\theta f\big((s, a'); \theta\big)\big], \tag{H.1}
$$

where we write $\mathbb{E}_{\pi_\theta}[\nabla_\theta f((s, a'); \theta)] = \mathbb{E}_{a' \sim \pi_\theta(\cdot \,|\, s)}[\nabla_\theta f((s, a'); \theta)]$ for notational simplicity. Meanwhile, recall that $\nabla_\theta f((\cdot, \cdot); \theta) = \phi_\theta(\cdot, \cdot)$, where $\phi_\theta$ is the feature mapping defined in (3.3). Thus, (H.1) implies that

$$
\nabla_\theta \log \pi_\theta(a \,|\, s) = \tau \cdot \phi_\theta(s, a) - \tau \cdot \mathbb{E}_{\pi_\theta}\big[\phi_\theta(s, a')\big]. \tag{H.2}
$$

Finally, by plugging (H.2) into (2.5) and (2.7), we have

$$
\nabla_\theta J(\pi_\theta) = \tau \cdot \mathbb{E}_{\sigma_{\pi_\theta}}\Big[Q^{\pi_\theta}(s, a) \cdot \Big(\phi_\theta(s, a) - \mathbb{E}_{\pi_\theta}\big[\phi_\theta(s, a')\big]\Big)\Big],
$$

$$
F(\theta) = \tau^2 \cdot \mathbb{E}_{\sigma_{\pi_\theta}}\Big[\Big(\phi_\theta(s, a) - \mathbb{E}_{\pi_\theta}\big[\phi_\theta(s, a')\big]\Big)\Big(\phi_\theta(s, a) - \mathbb{E}_{\pi_\theta}\big[\phi_\theta(s, a')\big]\Big)^\top\Big],
$$

which concludes the proof of Proposition 3.1. $\qquad\square$

## H.2    PROOF OF THEOREM 4.9

*Proof.* By Theorem 4.8, we have

$$
(1 - \gamma) \cdot \big(J(\pi^*) - J(\pi_{\widehat{\theta}})\big) \leq 2Q_{\max} \cdot \inf_{\theta \in \mathcal{B}} \|u_{\widehat{\theta}}(\cdot, \cdot) - \phi_{\widehat{\theta}}(\cdot, \cdot)^\top \theta\|_{\sigma_{\pi_{\widehat{\theta}}}}, \tag{H.3}
$$

where $u_{\widehat{\theta}}$ is defined in (4.4). It suffices to upper bound the right-hand side of (H.3) under the expectation over the random initialization. Following from the triangle inequality, we obtain that

$$
\inf_{\theta \in \mathcal{B}} \|u_{\widehat{\theta}}(\cdot, \cdot) - \phi_{\widehat{\theta}}(\cdot, \cdot)^\top \theta\|_{\sigma_{\pi_{\widehat{\theta}}}}
$$

$$
\leq \inf_{\theta \in \mathcal{B}}\Big\{\big\|u_{\widehat{\theta}}(\cdot, \cdot) - \Pi_{\widetilde{\mathcal{F}}_{R,m}} u_{\widehat{\theta}}(\cdot, \cdot)\big\|_{\sigma_{\pi_{\widehat{\theta}}}} + \big\|\Pi_{\widetilde{\mathcal{F}}_{R,m}} u_{\widehat{\theta}}(\cdot, \cdot) - \phi_{\widehat{\theta}}(\cdot, \cdot)^\top \theta\big\|_{\sigma_{\pi_{\widehat{\theta}}}}\Big\}
$$

$$
= \big\|u_{\widehat{\theta}}(\cdot, \cdot) - \Pi_{\widetilde{\mathcal{F}}_{R,m}} u_{\widehat{\theta}}(\cdot, \cdot)\big\|_{\sigma_{\pi_{\widehat{\theta}}}} + \inf_{\theta \in \mathcal{B}}\big\|\Pi_{\widetilde{\mathcal{F}}_{R,m}} u_{\widehat{\theta}}(\cdot, \cdot) - \phi_{\widehat{\theta}}(\cdot, \cdot)^\top \theta\big\|_{\sigma_{\pi_{\widehat{\theta}}}}, \tag{H.4}
$$

where $\widetilde{\mathcal{F}}_{R,m}$ is defined in Definition E.1. It remains to upper bound the right-hand side of (H.4). In what follows, we define $\widetilde{\theta}$ by

$$
\phi_0(\cdot, \cdot)^\top \widetilde{\theta} = \Pi_{\widetilde{\mathcal{F}}_{R,m}} u_{\widehat{\theta}}(\cdot, \cdot) \in \widetilde{\mathcal{F}}_{R,m},
$$

where $\phi_0$ is the feature mapping defined in (3.3) with $\theta = W_{\text{init}}$. By the definition of $\widetilde{\mathcal{F}}_{R,m}$ in Definition E.1, it holds that $\widetilde{\theta} \in \mathcal{B} = \{\alpha \in \mathbb{R}^{md} : \|\alpha - W_{\text{init}}\|_2 \leq R\}$. Thus, by (H.4) and the fact that $\widetilde{\theta} \in \mathcal{B}$, we have

$$
\inf_{\theta \in \mathcal{B}} \|u_{\widehat{\theta}}(\cdot, \cdot) - \phi_{\widehat{\theta}}(\cdot, \cdot)^\top \theta\|_{\sigma_{\pi_{\widehat{\theta}}}} \leq \|u_{\widehat{\theta}}(\cdot, \cdot) - \phi_0(\cdot, \cdot)^\top \widetilde{\theta}\|_{\sigma_{\pi_{\widehat{\theta}}}} + \|\phi_0(\cdot, \cdot)^\top \widetilde{\theta} - \phi_{\widehat{\theta}}(\cdot, \cdot)^\top \widetilde{\theta}\|_{\sigma_{\pi_{\widehat{\theta}}}}. \tag{H.5}
$$

Following from the proof of Proposition 4.3 in §F.1, it holds for $u_{\widehat{\theta}} \in \mathcal{F}_{R,\infty}$ that

$$
\mathbb{E}_{\text{init}}\big[\|u_{\widehat{\theta}}(\cdot, \cdot) - \phi_{\widehat{\theta}}(\cdot, \cdot)^\top \widetilde{\theta}\|_{\sigma_{\pi_{\widehat{\theta}}}}\big]
$$

$$
\leq \Big\{\mathbb{E}_{\text{init}}\big[\|u_{\widehat{\theta}}(\cdot, \cdot) - \phi_{\widehat{\theta}}(\cdot, \cdot)^\top \widetilde{\theta}\|_{\sigma_{\pi_{\widehat{\theta}}}}^2\big]\Big\}^{1/2} = \mathcal{O}(R \cdot m^{-1/2}), \tag{H.6}
$$

where the first inequality follows from the Jensen's inequality, and the expectations are taken over the random initialization. Meanwhile, following from Lemma E.2, we obtain for all $\widehat{\theta}, \widetilde{\theta} \in \mathcal{B}$ that

$$
\mathbb{E}_{\text{init}}\big[\|\phi_0(\cdot,\cdot)^\top\widetilde{\theta} - \phi_{\widehat{\theta}}(\cdot,\cdot)^\top\widetilde{\theta}\|_{\sigma_{\pi_{\widehat{\theta}}}}\big]
$$
$$
\leq \Big\{\mathbb{E}_{\text{init}}\big[\|\phi_0(\cdot,\cdot)^\top\widetilde{\theta} - \phi_{\widehat{\theta}}(\cdot,\cdot)^\top\widetilde{\theta}\|^2_{\sigma_{\pi_{\widehat{\theta}}}}\big]\Big\}^{1/2} = \mathcal{O}(R^{3/2}\cdot m^{-1/4}), \tag{H.7}
$$

where the expectations are taken over the random initialization. Finally, by plugging (H.6) and (H.7) into (H.5), we obtain that

$$
(1-\gamma)\cdot\mathbb{E}_{\text{init}}\big[J(\pi^*) - J(\pi_{\widehat{\theta}})\big] \leq 2Q_{\max}\cdot\mathbb{E}_{\text{init}}\Big[\inf_{\theta\in\mathcal{B}}\|u_{\widehat{\theta}}(\cdot,\cdot) - \phi_{\widehat{\theta}}(\cdot,\cdot)^\top\theta\|_{\sigma_{\pi_{\widehat{\theta}}}}\Big] = \mathcal{O}(R^{3/2}\cdot m^{-1/4}),
$$

where the first inequality follows from (H.3). Similarly, if the assumption that $u_{\widehat{\theta}} \in \mathcal{F}_{R,\infty}$ is not imposed, we conclude that

$$
(1-\gamma)\cdot\mathbb{E}_{\text{init}}\big[J(\pi^*) - J(\pi_{\widehat{\theta}})\big] \leq \mathcal{O}(R^{3/2}\cdot m^{-1/4}) + \mathbb{E}_{\text{init}}\big[\|\Pi_{\mathcal{F}_{R,\infty}}u_{\widehat{\theta}} - u_{\widehat{\theta}}\|_{\sigma_{\pi_{\widehat{\theta}}}}\big],
$$

which completes the proof of Theorem 4.9. $\qquad\square$

## H.3 PROOF OF INEQUALITY (4.5)

*Proof.* Recall that we define $\rho_i$ by

$$
\rho_i = \eta^{-1}\cdot\Big(\Pi_{\mathcal{B}}\big(\theta_i + \eta\cdot\nabla_\theta J(\pi_{\theta_i})\big) - \theta_i\Big), \tag{H.8}
$$

where $\Pi_{\mathcal{B}}$ is the projection operator onto $\mathcal{B}$. Following from (H.8) and the fact that $(\Pi_{\mathcal{B}}y - y)^\top(\Pi_{\mathcal{B}}y - x) \leq 0$ for all $x \in \mathcal{B}$, we have

$$
\big(\eta\cdot\rho_i - \eta\cdot\nabla_\theta J(\pi_{\theta_i})\big)^\top(\eta\cdot\rho_i + \theta_i - \theta) \leq 0, \quad \forall\theta\in\mathcal{B}. \tag{H.9}
$$

Thus, following from (H.9), we obtain that

$$
\nabla_\theta J(\pi_{\theta_i})^\top(\theta - \theta_i) \leq \rho_i^\top(\theta - \theta_i) - \eta\cdot\|\rho_i\|_2^2 + \eta\cdot\rho_i^\top\nabla_\theta J(\pi_{\theta_i})
$$
$$
\leq \|\rho_i\|_2\cdot\big(\|\theta - \theta_i\|_2 + \eta\cdot\|\nabla_\theta J(\pi_{\theta_i})\|_2\big), \quad \forall\theta\in\mathcal{B}, \tag{H.10}
$$

where the last inequality follows from the Cauchy-Schwartz inequality and the fact that $-\eta\cdot\|\rho_i\|_2^2 \leq 0$. It remains to upper bound the right-hand side of (H.10). For all $\theta, \theta_i \in \mathcal{B} = \{\alpha \in \mathbb{R}^{md} : \|\alpha - W_{\text{init}}\|_2 \leq R\}$, we have $\|\theta - \theta_i\|_2 \leq 2R$. Meanwhile, recall that we set $\tau_i = 1$. Therefore, following from Proposition 3.1, we obtain that

$$
\|\nabla_\theta J(\pi_{\theta_i})\|_2 \leq \mathbb{E}_{\sigma_i}\big[|Q^{\pi_{\theta_i}}(s,a)|\cdot\|\overline{\phi}_{\theta_i}(s,a)\|_2\big] \leq 2Q_{\max}, \tag{H.11}
$$

where the first inequality follows from the Jensen's inequality, and the second inequality follows from the facts that $|Q^{\pi_{\theta_i}}(s,a)| \leq Q_{\max}$ and $\|\overline{\phi}_{\theta_i}(s,a)\|_2 \leq 2$ for all $(s,a) \in \mathcal{S}\times\mathcal{A}$. By plugging (H.11) and the upper bound $\|\theta - \theta_i\|_2 \leq 2R$ into (H.10), we conclude that

$$
\nabla_\theta J(\pi_{\theta_i})^\top(\theta - \theta_i) \leq (2R + 2\eta\cdot Q_{\max})\cdot\|\rho_i\|_2, \quad \forall\theta\in\mathcal{B},
$$

which concludes the proof of (4.5). $\qquad\square$

## H.4 PROOF OF COROLLARY A.5

*Proof.* It suffices to calculate $\overline{\epsilon}_i(T)$ defined in (A.3) in Theorem A.4. Note that we set $\tau_i = (i-1)/\sqrt{T}$. Therefore, we have $\tau_i = \mathcal{O}(\sqrt{T})$ for all $i \in [T]$. Thus, it holds for $m = \Omega(R^{10}\cdot T^6)$ that

$$
\mathcal{O}\big((\tau_{i+1}\cdot T^{1/2} + 1)\cdot R^{3/2}\cdot m^{-1/4}\big) = \mathcal{O}(T^{-1/2}), \quad \forall i\in[T],
$$
$$
\mathcal{O}(R^{5/4}\cdot m^{-1/8}) = \mathcal{O}(T^{-1/2}). \tag{H.12}
$$

Meanwhile, it holds for $B = \Omega(R^2\cdot T^2\cdot\sigma_\xi^2)$ that

$$
R^{1/2}\cdot(\sigma_\xi^2/B)^{1/4} = \mathcal{O}(T^{-1/2}). \tag{H.13}
$$

Therefore, combining (H.12) and (H.13), we obtain that

$$\bar{\epsilon}_i(T) = \sqrt{8}c_0 \cdot R^{1/2} \cdot (\sigma_\xi^2/B)^{1/4} + \mathcal{O}\big((1 + \tau_{i+1} \cdot T^{1/2}) \cdot R^{3/2} \cdot m^{-1/4} + R^{5/4} \cdot m^{-1/8}\big)$$
$$= \mathcal{O}(T^{-1/2}).$$

By Theorem A.4, we have

$$\min_{i \in [T]} \mathbb{E}\big[J(\pi^*) - J(\pi_{\theta_i})\big] = \frac{\log |\mathcal{A}| + 9R^2 + M}{(1 - \gamma) \cdot \sqrt{T}} + \mathcal{O}\big((1 - \gamma)^{-1} \cdot T^{-1/2}\big)$$
$$= \mathcal{O}\bigg(\frac{\log |\mathcal{A}|}{(1 - \gamma) \cdot \sqrt{T}}\bigg),$$

which concludes the proof of Corollary A.5. $\qquad\square$

## H.5 PROOF OF LEMMA D.2

*Proof.* In the sequel, we write $g_i = \mathbb{E}[\widehat{\nabla}_\theta J(\pi_{\theta_i})]$ for notational simplicity, where $\widehat{\nabla}_\theta J(\pi_{\theta_i})$ is defined in (3.7), and the expectation is taken over $\sigma_i$ given $\theta_i$ and $\omega_i$. Recall that we set $\tau_i = 1$. By Proposition 3.1, we obtain that

$$|(\nabla_\theta J(\pi_{\theta_i}) - g_i)^\top \delta_i| = \Big|\mathbb{E}_{\sigma_i}\Big[\overline{\phi}_{\theta_i}(s, a) \cdot \big(Q^{\pi_{\theta_i}}(s, a) - Q_{\omega_i}(s, a)\big)\Big]^\top \delta_i\Big|$$
$$\leq \|\delta_i\|_2 \cdot \mathbb{E}_{\sigma_i}\big[\|\overline{\phi}_{\theta_i}(s, a)\|_2 \cdot |Q^{\pi_{\theta_i}}(s, a) - Q_{\omega_i}(s, a)|\big], \qquad (\text{H.14})$$

where $\overline{\phi}_{\theta_i}(s, a)$ is the centered feature mapping defined in (B.2) with $\theta = \theta_i$, and the inequality follows from the Jensen's inequality. Note that $\theta_i, \theta_{i+1} \in \mathcal{B}$. It holds that

$$\|\delta_i\|_2 = \|\theta_{i+1} - \theta_i\|_2/\eta \leq 2R/\eta.$$

Meanwhile, note that $\|\overline{\phi}_{\theta_i}(s, a)\|_2 \leq 2$ for all $(s, a) \in \mathcal{S} \times \mathcal{A}$. Therefore, it follows from Assumption 4.5 and (H.14) that

$$|(\nabla_\theta J(\pi_{\theta_i}) - g_i)^\top \delta_i| \leq 4R/\eta \cdot \mathbb{E}_{\sigma_i}\big[|Q^{\pi_{\theta_i}}(s, a) - Q_{\omega_i}(s, a)|\big]$$
$$\leq 4R/\eta \cdot \Big\{\mathbb{E}_{\varsigma_i}\big[(\mathrm{d}\sigma_i/\mathrm{d}\varsigma_i)^2\big]\Big\}^{1/2} \cdot \|Q^{\pi_{\theta_i}} - Q_{\omega_i}\|_{\varsigma_i}$$
$$\leq 4\kappa \cdot R/\eta \cdot \|Q^{\pi_{\theta_i}} - Q_{\omega_i}\|_{\varsigma_i},$$

where the second inequality follows from the Cauchy-Schwartz inequality, $\mathrm{d}\sigma_i/\mathrm{d}\varsigma_i$ is the Radon-Nikodym derivative, and $\kappa$ is defined in Assumption 4.5. Thus, we complete the proof of Lemma D.2. $\qquad\square$

## H.6 PROOF OF LEMMA D.3

*Proof.* In what follows, we write $g_i = \mathbb{E}[\widehat{\nabla} J(\pi_{\theta_i})]$ for notational simplicity, where the expectation is taken over $\sigma_i$ given $\theta_i$ and $\omega_i$. Note that

$$\mathbb{E}\big[\|\nabla_\theta J(\pi_{\theta_i}) - \widehat{\nabla}_\theta J(\pi_{\theta_i})\|_2^2\big] \leq 2\mathbb{E}\big[\|\xi_i\|_2^2\big] + 2\mathbb{E}\big[\|\nabla_\theta J(\pi_{\theta_i}) - g_i\|_2^2\big], \qquad (\text{H.15})$$

where we use the fact that $\|x + y\|_2^2 \leq 2\|x\|_2^2 + 2\|y\|_2^2$, and the expectations are taken over all the randomness. By Proposition 3.1, we have

$$\|\nabla_\theta J(\pi_{\theta_i}) - g_i\|_2 = \Big\|\mathbb{E}_{\sigma_i}\Big[\overline{\phi}_{\theta_i}(s, a) \cdot \big(Q^{\pi_{\theta_i}}(s, a) - Q_{\omega_i}(s, a)\big)\Big]\Big\|_2$$
$$\leq \mathbb{E}_{\sigma_i}\big[\|\overline{\phi}_{\theta_i}(s, a)\|_2 \cdot |Q^{\pi_{\theta_i}}(s, a) - Q_{\omega_i}(s, a)|\big], \qquad (\text{H.16})$$

where $\overline{\phi}_{\theta_i}$ is defined in (B.2) with $\theta = \theta_i$ and the second inequality follows from the Jensen's inequality. Since $\|\overline{\phi}_{\theta_i}(s, a)\|_2 \leq 2$ for all $(s, a) \in \mathcal{S} \times \mathcal{A}$, we obtain from (H.16) that

$$\|\nabla_\theta J(\pi_{\theta_i}) - g_i\|_2^2 \leq \Big\{\mathbb{E}_{\sigma_i}\big[\|\overline{\phi}_{\theta_i}(s, a)\|_2 \cdot |Q^{\pi_{\theta_i}}(s, a) - Q_{\omega_i}(s, a)|\big]\Big\}^2$$
$$\leq 4\kappa^2 \cdot \|Q^{\pi_{\theta_i}} - Q_{\omega_i}\|_{\varsigma_i}^2, \qquad (\text{H.17})$$

where $\kappa$ is defined in Assumption 4.5 and the inequality follows from the Cauchy-Schwartz inequality. By plugging (H.17) into (H.15), we obtain that

$$\mathbb{E}\big[\|\nabla_\theta J(\pi_{\theta_i}) - \widehat{\nabla}_\theta J(\pi_{\theta_i})\|_2^2\big] \leq 2\mathbb{E}\big[\|\xi_i\|_2^2\big] + 8\kappa^2 \cdot \mathbb{E}\big[\|Q^{\pi_{\theta_i}} - Q_{\omega_i}\|_{\sigma_i}^2\big],$$

which concludes the proof of Lemma D.3. $\qquad\square$

### H.7 PROOF OF LEMMA D.4

*Proof.* By the definition of the KL divergence, it holds for all $s \in \mathcal{S}$ that

$$D_{\mathrm{KL}}\big(\pi^*(\cdot\,|\,s)\big\|\pi_i(\cdot\,|\,s)\big) - D_{\mathrm{KL}}\big(\pi^*(\cdot\,|\,s)\big\|\pi_{i+1}(\cdot\,|\,s)\big)$$
$$= \big\langle \log\big(\pi_{i+1}(\cdot\,|\,s)/\pi_i(\cdot\,|\,s)\big), \pi^*(\cdot\,|\,s)\big\rangle. \tag{H.18}$$

Meanwhile, the right-hand side of (H.18) can be expanded as follows,

$$\big\langle \log\big(\pi_{i+1}(\cdot\,|\,s)/\pi_i(\cdot\,|\,s)\big), \pi^*(\cdot\,|\,s)\big\rangle$$
$$= \big\langle \log\big(\pi_{i+1}(\cdot\,|\,s)/\pi_i(\cdot\,|\,s)\big), \pi^*(\cdot\,|\,s) - \pi_{i+1}(\cdot\,|\,s)\big\rangle + \big\langle \log\big(\pi_{i+1}(\cdot\,|\,s)/\pi_i(\cdot\,|\,s)\big), \pi_{i+1}(\cdot\,|\,s)\big\rangle$$
$$= \underbrace{\big\langle \log\big(\pi_{i+1}(\cdot\,|\,s)/\pi_i(\cdot\,|\,s)\big), \pi^*(\cdot\,|\,s) - \pi_{i+1}(\cdot\,|\,s)\big\rangle}_{L_i} + D_{\mathrm{KL}}\big(\pi_{i+1}(\cdot\,|\,s)\big\|\pi_i(\cdot\,|\,s)\big). \tag{H.19}$$

Combining (H.18) and (H.19), we obtain that

$$L_i = D_{\mathrm{KL}}\big(\pi^*(\cdot\,|\,s)\big\|\pi_i(\cdot\,|\,s)\big) - D_{\mathrm{KL}}\big(\pi^*(\cdot\,|\,s)\big\|\pi_{i+1}(\cdot\,|\,s)\big) - D_{\mathrm{KL}}\big(\pi_{i+1}(\cdot\,|\,s)\big\|\pi_i(\cdot\,|\,s)\big). \tag{H.20}$$

In what follows, we calculate the difference

$$\mathbb{E}_{\nu_*}[L_i] - (1-\gamma)\cdot\eta\cdot\big(J(\pi^*) - J(\pi_i)\big).$$

By Lemma D.1, we have

$$J(\pi^*) - J(\pi_i) = (1-\gamma)^{-1}\cdot\mathbb{E}_{\nu_*}\big[\langle Q^{\pi_i}(s,\cdot), \pi^*(s,\cdot) - \pi_i(s,\cdot)\rangle\big]. \tag{H.21}$$

Meanwhile, for $L_i$ defined in (H.19), we obtain that

$$L_i - \eta\cdot\langle Q^{\pi_i}(s,\cdot), \pi^*(s,\cdot) - \pi_i(s,\cdot)\rangle$$
$$= \big\langle \log\big(\pi_{i+1}(\cdot\,|\,s)/\pi_i(\cdot\,|\,s)\big), \pi^*(\cdot\,|\,s) - \pi_{i+1}(\cdot\,|\,s)\big\rangle - \eta\cdot\langle Q^{\pi_i}(s,\cdot), \pi^*(s,\cdot) - \pi_i(s,\cdot)\rangle$$
$$= \big\langle \log\big(\pi_{i+1}(\cdot\,|\,s)/\pi_i(\cdot\,|\,s)\big) - \eta\cdot Q_{\omega_i}(s,\cdot), \pi^*(\cdot\,|\,s) - \pi_i(\cdot\,|\,s)\big\rangle \tag{H.22}$$
$$\quad + \eta\cdot\langle Q_{\omega_i}(s,\cdot) - Q^{\pi_i}(s,\cdot), \pi^*(\cdot\,|\,s) - \pi_i(\cdot\,|\,s)\rangle$$
$$\quad + \big\langle \log\big(\pi_{i+1}(\cdot\,|\,s)/\pi_i(\cdot\,|\,s)\big), \pi_i(\cdot\,|\,s) - \pi_{i+1}(\cdot\,|\,s)\big\rangle.$$

Note that upon taking the expectation over $s \sim \nu_*(\cdot)$ in (H.22), the right-hand side of (H.22) is equal to $H_i$ defined in (D.21) of Lemma D.4. Thus, combining (H.21) and (H.22), we obtain that

$$\mathbb{E}_{\nu_*}[L_i] - (1-\gamma)\cdot\eta\cdot\big(J(\pi^*) - J(\pi_i)\big) = H_i, \tag{H.23}$$

where $H_i$ is defined in (D.21). By plugging (H.20) into (H.23), we conclude that

$$(1-\gamma)\cdot\eta\cdot\big(J(\pi^*) - J(\pi_i)\big) = \mathbb{E}_{\nu_*}\Big[D_{\mathrm{KL}}\big(\pi^*(\cdot\,|\,s)\big\|\pi_i(\cdot\,|\,s)\big) - D_{\mathrm{KL}}\big(\pi^*(\cdot\,|\,s)\big\|\pi_{i+1}(\cdot\,|\,s)\big)$$
$$- D_{\mathrm{KL}}\big(\pi_{i+1}(\cdot\,|\,s)\big\|\pi_i(\cdot\,|\,s)\big)\Big] - H_i,$$

which concludes the proof of Lemma D.4. $\qquad\qquad\square$

### H.8 PROOF OF LEMMA D.5

*Proof.* By (D.21), we have

$$\mathbb{E}\big[|H_i|\big] \le \mathbb{E}\Big[\mathbb{E}_{\nu_*}\big[\langle \log\big(\pi_{i+1}(\cdot\,|\,s)/\pi_i(\cdot\,|\,s)\big) - \eta\cdot Q_{\omega_i}(s,\cdot), \pi^*(\cdot\,|\,s) - \pi_i(\cdot\,|\,s)\rangle\big]\Big] \tag{H.24}$$
$$\quad + \eta\cdot\mathbb{E}\Big[\mathbb{E}_{\nu_*}\big[|\langle Q_{\omega_i}(s,\cdot) - Q^{\pi_i}(s,\cdot), \pi^*(\cdot\,|\,s) - \pi_i(\cdot\,|\,s)\rangle|\big]\Big]$$
$$\quad + \mathbb{E}\Big[\mathbb{E}_{\nu_*}\big[|\langle \log\big(\pi_i(\cdot\,|\,s)/\pi_{i+1}(\cdot\,|\,s)\big), \pi_{i+1}(\cdot\,|\,s) - \pi_i(\cdot\,|\,s)\rangle|\big]\Big],$$

where the inequality follows from the Jensen's inequality, and the expectations are taken over all the randomness. To prove Lemma D.5, we establish the upper bounds of the three terms on the right-hand side of (H.24) respectively in the following lemmas.

**Lemma H.1.** It holds that

$$\mathbb{E}\Big[\mathbb{E}_{\nu_*}\big[|\langle Q_{\omega_i}(s,\cdot) - Q^{\pi_i}(s,\cdot), \pi^*(\cdot\,|\,s) - \pi_i(\cdot\,|\,s)\rangle|\big]\Big] \le (\phi_i' + \psi_i')\cdot\mathbb{E}\big[\|Q_{\omega_i} - Q^{\pi_i}\|_{\varsigma_i}\big],$$

where $\phi_i'$, $\psi_i'$ are the concentrability coefficients defined in (A.1) of Assumption A.2. Here the expectations are taken over all the randomness.

*Proof.* See §I.1 for a detailed proof. □

**Lemma H.2.** Under Assumptions 4.2 and A.3, it holds that

$$\mathbb{E}\Big[\mathbb{E}_{\nu_*}\big[|\langle \log(\pi_{i+1}(\cdot\,|\,s)/\pi_i(\cdot\,|\,s)), \pi_i(\cdot\,|\,s) - \pi_{i+1}(\cdot\,|\,s)\rangle|\big]\Big]$$

$$\le \mathbb{E}\Big[\mathbb{E}_{\nu_*}\big[D_{\mathrm{KL}}\big(\pi_{i+1}(\cdot\,|\,s)\big\|\pi_i(\cdot\,|\,s)\big)\big]\Big] + \eta^2\cdot(9R^2 + M^2) + \mathcal{O}(\tau_{i+1}\cdot R^{3/2}\cdot m^{-1/4}),$$

where $M$ is the absolute constant defined in Assumption A.3. Here the expectations are taken over all the randomness.

*Proof.* See §I.2 for a detailed proof. □

**Lemma H.3.** Under Assumption 4.2, it holds that

$$\mathbb{E}\Big[\mathbb{E}_{\nu_*}\big[|\langle \log(\pi_{i+1}(\cdot\,|\,s)/\pi_i(\cdot\,|\,s)) - \eta\cdot Q_{\omega_i}(s,\cdot), \pi^*(\cdot\,|\,s) - \pi_i(\cdot\,|\,s)\rangle|\big]\Big]$$

$$\le \sqrt{2}(\varphi_i + \psi_i)\cdot\eta\cdot R^{1/2}\cdot\tau_i^{-1}\cdot\Big\{\mathbb{E}\big[\|\xi_i(\delta_i)\|_2\big] + \mathbb{E}\big[\|\xi_i(\omega_i)\|_2\big]\Big\}^{1/2}$$

$$+ \mathcal{O}\big((\tau_{i+1} + \eta)\cdot R^{3/2}\cdot m^{-1/4} + \eta\cdot R^{5/4}\cdot m^{-1/8}\big),$$

where $\varphi_i$ and $\psi_i$ are the concentrability coefficients defined in (A.1) of Assumption A.2 and $\xi_i(\delta_i)$, $\xi_i(\omega_i)$ are defined in Assumption A.1. Here the expectations are taken over all the randomness.

*Proof.* See §I.3 for a detailed proof. □

Finally, applying Lemmas H.1, H.2, and H.3 to (H.24), it holds under Assumptions 4.2 and A.3 that

$$\mathbb{E}\Big[|H_i| - \mathbb{E}_{\nu_*}\big[D_{\mathrm{KL}}\big(\pi_{i+1}(\cdot\,|\,s)\big\|\pi_i(\cdot\,|\,s)\big)\big]\Big] \le \eta^2\cdot(6R^2 + M^2) + \eta\cdot(\varphi_i' + \psi_i')\cdot\varepsilon_{Q,i} + \varepsilon_i,$$

where

$$\varepsilon_{Q,i} = \mathbb{E}\big[\|Q^{\pi_i} - Q_{\omega_i}\|_{\varsigma_i}\big]$$

$$\varepsilon_i = \Big\{\mathbb{E}\big[\|\xi_i(\delta_i)\|_2 + \|\xi_i(\omega_i)\|_2\big]\Big\}^{1/2} + \mathcal{O}\big((\tau_{i+1} + \eta)\cdot R^{3/2}\cdot m^{-1/4} + \eta\cdot R^{5/4}\cdot m^{-1/8}\big).$$

Here the expectations are taken over all the randomness. Therefore, we complete the proof of Lemma D.5. □

# I PROOF OF SUPPORTING LEMMAS

In this section, we provide the proof of the lemmas in §H.

## I.1 PROOF OF LEMMA H.1

*Proof.* We define $\Delta_{Q,i}(s,a) = Q_{\omega_i}(s,a) - Q^{\pi_i}(s,a)$ for all $(s,a) \in \mathcal{S} \times \mathcal{A}$. It holds for all $i \in [T]$ that

$$
\mathbb{E}_{\nu^*}\big[|\langle \Delta_{Q,i}(s,\cdot), \pi^*(\cdot\,|\,s) - \pi_i(\cdot\,|\,s)\rangle|\big]
$$
$$
= \int_{\mathcal{S}} \bigg|\sum_{a \in \mathcal{A}} \Delta_{Q,i}(s,a) \cdot \big(\pi^*(a\,|\,s) - \pi_i(a\,|\,s)\big)\bigg| \mathrm{d}\nu_*(s). \tag{I.1}
$$

Meanwhile, it holds for any $s \in \mathcal{S}$ that

$$
\bigg|\sum_{a \in \mathcal{A}} \Delta_{Q,i}(s,a) \cdot \big(\pi^*(a\,|\,s) - \pi_i(a\,|\,s)\big)\bigg|
$$
$$
= \bigg|\int_{a \in \mathcal{A}} \Delta_{Q,i}(s,a) \cdot \big(\pi^*(a\,|\,s) - \pi_i(a\,|\,s)\big)/\pi_i(a\,|\,s)\mathrm{d}\pi_i(a\,|\,s)\bigg|
$$
$$
\leq \int_{a \in \mathcal{A}} \big|\Delta_{Q,i}(s,a) \cdot \big(\pi^*(a\,|\,s) - \pi_i(a\,|\,s)\big)/\pi_i(a\,|\,s)\big|\mathrm{d}\pi_i(a\,|\,s), \tag{I.2}
$$

where the inequality follows from the Jensen's inequality. By plugging (I.2) into (I.1), we have

$$
\mathbb{E}_{\nu^*}\big[|\langle \Delta_{Q,i}(s,\cdot), \pi^*(\cdot\,|\,s) - \pi_i(\cdot\,|\,s)\rangle|\big]
$$
$$
\leq \int_{\mathcal{S} \times \mathcal{A}} \big|\Delta_{Q,i}(s,a) \cdot \big(\pi^*(a\,|\,s) - \pi_i(a\,|\,s)\big)/\pi_i(a\,|\,s)\big|\mathrm{d}\widetilde{\sigma}(s,a), \tag{I.3}
$$

where we define $\widetilde{\sigma}(\cdot,\cdot) = \pi_i(\cdot\,|\,\cdot) \cdot \nu_*(\cdot)$. Recall that $\varsigma_i(\cdot,\cdot) = \pi_i(\cdot\,|\,\cdot) \cdot \varrho_i(\cdot)$ and $\sigma_*(\cdot,\cdot) = \pi^*(\cdot\,|\,\cdot) \cdot \nu_*(\cdot)$. Therefore, following from (I.3), it holds that

$$
\mathbb{E}_{\nu^*}\big[|\langle \Delta_{Q,i}(s,\cdot), \pi^*(\cdot\,|\,s) - \pi_i(\cdot\,|\,s)\rangle|\big]
$$
$$
\leq \int_{\mathcal{S} \times \mathcal{A}} |\Delta_{Q,i}(s,a)|\mathrm{d}\sigma_* + \int_{\mathcal{S} \times \mathcal{A}} |\Delta_{Q,i}(s,a)| \cdot \frac{\mathrm{d}\nu^*}{\mathrm{d}\varrho_i}(s)\,\mathrm{d}\varsigma_i(s,a). \tag{I.4}
$$

Finally, applying the Cauchy-Schwartz inequality to (I.4) yields that

$$
\mathbb{E}\Big[\mathbb{E}_{\nu_*}\big[|\langle \Delta_{Q,i}(s,\cdot), \pi^*(\cdot\,|\,s) - \pi_i(\cdot\,|\,s)\rangle|\big]\Big]
$$
$$
\leq \bigg(\big\{\mathbb{E}_{\varsigma_i}\big[(\mathrm{d}\sigma_*/\mathrm{d}\varsigma_i)^2\big]\big\}^{1/2} + \big\{\mathbb{E}_{\varrho_i}\big[(\mathrm{d}\nu_*/\mathrm{d}\varrho_i)^2\big]\big\}^{1/2}\bigg) \cdot \mathbb{E}\bigg[\big\{\mathbb{E}_{\varsigma_i}\big[|\Delta_{Q,i}(s,a)|^2\big]\big\}^{1/2}\bigg]
$$
$$
= (\varphi_i' + \psi_i') \cdot \mathbb{E}\bigg[\big\{\mathbb{E}_{\varsigma_i}\big[|\Delta_{Q,i}(s,a)|^2\big]\big\}^{1/2}\bigg] = (\varphi_i' + \psi_i') \cdot \mathbb{E}\big[\|\Delta_{Q,i}\|_{\varsigma_i}\big],
$$

where $\mathrm{d}\sigma_*/\mathrm{d}\varsigma_i$ and $\mathrm{d}\nu_*/\mathrm{d}\varrho_i$ are the Radon-Nikodym derivatives, $\varphi_i'$ and $\psi_i'$ are the concentrability coefficients defined in (A.1) of Assumption A.2, and the expectations are taken over all the randomness. Thus, we complete the proof of Lemma H.1. $\qquad\square$

## I.2 PROOF OF LEMMA H.2

*Proof.* Following from the definition of $\pi_\theta$ in (3.2), we obtain that

$$
\langle \log\big(\pi_{i+1}(\cdot\,|\,s)/\pi_i(\cdot\,|\,s)\big), \pi_i(\cdot\,|\,s) - \pi_{i+1}(\cdot\,|\,s)\rangle
$$
$$
= \langle \tau_{i+1} \cdot f\big((s,\cdot);\theta_{i+1}\big) - \tau_i \cdot f\big((s,\cdot);\theta_i\big), \pi_i(\cdot\,|\,s) - \pi_{i+1}(\cdot\,|\,s)\rangle \tag{I.5}
$$
$$
- \langle C_i(s), \pi_i(\cdot\,|\,s) - \pi_{i+1}(\cdot\,|\,s)\rangle,
$$

where $f((\cdot,\cdot);\theta)$ is the two-layer neural network defined in (3.1) and $C_i(s)$ is defined by

$$
C_i(s) = \log\bigg(\sum_{a \in \mathcal{A}} \exp\Big(\tau_i \cdot f\big((s,a);\theta_i\big)\Big)\bigg) - \log\bigg(\sum_{a \in \mathcal{A}} \exp\Big(\tau_{i+1} \cdot f\big((s,a);\theta_{i+1}\big)\Big)\bigg).
$$

Note that both $\pi_i(\cdot\,|\,s)$ and $\pi_{i+1}(\cdot\,|\,s)$ are distributions over $\mathcal{A}$, which implies that

$$
\langle C_i(s), \pi_i(\cdot\,|\,s) - \pi_{i+1}(\cdot\,|\,s)\rangle = C_i(s) - C_i(s) = 0, \quad \forall s \in \mathcal{S}. \tag{I.6}
$$

Meanwhile, recall that we define the feature mapping $\phi_\theta(s, a)$ in (3.3). For the two-layer neural network $f((\cdot, \cdot); \theta)$, we have

$$f\big((s, a); \theta\big) = \phi_\theta(s, a)^\top \theta, \quad \forall (s, a) \in \mathcal{S} \times \mathcal{A}. \tag{I.7}$$

In what follows, we write $\phi_i(s, a) = \phi_{\theta_i}(s, a)$ and $\Delta_i(a \,|\, s) = \pi_i(a \,|\, s) - \pi_{i+1}(a \,|\, s)$ for notational simplicity. By plugging (I.6) and (I.7) into (I.5), we obtain for all $s \in \mathcal{S}$ that

$$
\begin{aligned}
\big|\big\langle \log\big(\pi_{i+1}(\cdot \,|\, s) / \pi_i(\cdot \,|\, s)\big), \Delta_i(\cdot \,|\, s)\big\rangle\big| &= \big|\big\langle \tau_{i+1} \cdot \phi_{i+1}(s, \cdot)^\top \theta_{i+1} - \tau_i \cdot \phi_i(s, \cdot)^\top \theta_i, \Delta_i(\cdot \,|\, s)\big\rangle\big| \\
&\leq \big|\big\langle \phi_i(s, \cdot)^\top(\tau_{i+1} \cdot \theta_{i+1} - \tau_i \cdot \theta_i), \Delta_i(\cdot \,|\, s)\big\rangle\big| \\
&\quad + \tau_{i+1} \cdot \big|\big\langle \phi_{i+1}(s, \cdot)^\top \theta_{i+1} - \phi_i(s, \cdot)^\top \theta_{i+1}, \Delta_i(\cdot \,|\, s)\big\rangle\big| \\
&\leq \underbrace{\|\phi_i(s, \cdot)^\top(\tau_{i+1} \cdot \theta_{i+1} - \tau_i \cdot \theta_i)\|_{\infty, \mathcal{A}} \cdot \|\Delta_i(\cdot \,|\, s)\|_{1, \mathcal{A}}}_{\text{(i)}} \\
&\quad + \underbrace{\tau_{i+1} \cdot \big|\big\langle \phi_{i+1}(s, \cdot)^\top \theta_{i+1} - \phi_i(s, \cdot)^\top \theta_{i+1}, \Delta_i(\cdot \,|\, s)\big\rangle\big|}_{\text{(ii)}},
\end{aligned}
\tag{I.8}
$$

where the last inequality follows from the Hölder's inequality. Here we denote by $\|\cdot\|_{\infty, \mathcal{A}}$ and $\|\cdot\|_{1, \mathcal{A}}$ the $\ell_\infty$- and $\ell_1$-norms defined on $\mathbb{R}^{|\mathcal{A}|}$, respectively. In what follows, we upper bound (i) and (ii) on the right-hand side of (I.8) respectively.

**Upper Bounding (i) in** (I.8)**.** Recall that we define

$$\delta_i = \eta^{-1} \cdot (\tau_{i+1} \cdot \theta_{i+1} - \tau_i \cdot \theta_i) = \operatorname*{argmin}_{\alpha \in \mathcal{B}} \|\widehat{F}(\theta_i) \cdot \alpha - \tau_i \cdot \widehat{\nabla} J(\pi_{\theta_i})\|_2.$$

Thus, it holds that $\delta_i \in \mathcal{B}$ and $\|\delta_i - W_{\text{init}}\|_2 \leq R$, where $W_{\text{init}}$ is the initial parameter. In what follows, we denote by $\phi_0$ the feature mapping defined in (3.3) with $\theta = W_{\text{init}}$. Then for all $(s, a) \in \mathcal{S} \times \mathcal{A}$, we have

$$
\begin{aligned}
|\phi_i(s, a)^\top(\tau_{i+1} \cdot \theta_{i+1} - \tau_i \cdot \theta_i)| &= \eta \cdot |\phi_i(s, a)^\top \delta_i| \\
&\leq \eta \cdot \big(|\phi_0(s, a)^\top W_{\text{init}}| + |\phi_i(s, a)^\top \delta_i - \phi_i(s, a)^\top \theta_i| + |\phi_i(s, a)^\top \theta_i - \phi_0(s, a)^\top W_{\text{init}}|\big) \\
&\leq \eta \cdot \big(M_0 + \|\phi_i(s, a)\|_2 \cdot \|\delta_i - \theta_i\|_2 + |\phi_i(s, a)^\top \theta_i - \phi_0(s, a)^\top W_{\text{init}}|\big),
\end{aligned}
\tag{I.9}
$$

where the first inequality follows from the triangle inequality, the second inequality follows from the Cauchy-Schwartz inequality, and $M_0$ is defined by

$$M_0 = \sup_{(s, a) \in \mathcal{S} \times \mathcal{A}} |\phi_0(s, a)^\top W_{\text{init}}|. \tag{I.10}$$

In what follows, we upper bound the right-hand side of (I.9). Note that $\tau_{i-1} + \eta = \tau_i$. Therefore, we obtain that

$$\|\theta_i - W_{\text{init}}\|_2 \leq \tau_{i-1}/\tau_i \cdot \|\theta_{i-1} - W_{\text{init}}\|_2 + \eta/\tau_i \cdot \|\delta_{i-1} - W_{\text{init}}\|_2, \tag{I.11}$$

which holds for all $i > 1$. Recursively, since $\theta_1 = W_{\text{init}} \in \mathcal{B}$ and $\delta_i \in \mathcal{B}$ for all $i \in [T]$, it then follows from (I.11) that $\theta_i \in \mathcal{B}$ for all $i \in [T]$. Thus, it holds that $\|\delta_i - \theta_i\|_2 \leq 2R$. Meanwhile, following from (3.3), it holds for all $\theta \in \mathbb{R}^{md}$ and $(s, a) \in \mathcal{S} \times \mathcal{A}$ that $\|\phi_\theta(s, a)\|_2 \leq 1$. Therefore, we obtain that

$$\|\phi_i(s, a)\|_2 \cdot \|\delta_i - \theta_i\|_2 \leq 2R, \quad \forall (s, a) \in \mathcal{S} \times \mathcal{A}. \tag{I.12}$$

It remains to upper bound $|\phi_i(s, a)^\top \theta_i - \phi_0(s, a)^\top W_{\text{init}}|$ for all $(s, a) \in \mathcal{S} \times \mathcal{A}$, which is equal to $|f((s, a); \theta_i) - f((s, a); W_{\text{init}})|$ by (I.7). Recall that $f((\cdot, \cdot); \theta)$ is differentiable with respect to $\theta \in \mathbb{R}^{md}$ almost everywhere, and the gradient $\nabla_\theta f = ([\nabla_\theta f]_1^\top, \ldots, [\nabla_\theta f]_m^\top)^\top$ is given by

$$[\nabla_\theta f]_r(s, a) = \frac{b_r}{\sqrt{m}} \cdot \mathbb{1}\big\{(s, a)^\top[\theta]_r > 0\big\} \cdot (s, a) = [\phi_\theta]_r(s, a), \quad \forall (s, a) \in \mathcal{S} \times \mathcal{A},$$

where $\phi_\theta(s, a)$ is defined in (3.3). Since $\|\phi_\theta(s, a)\|_2 \leq 1$ for all $\theta \in \mathbb{R}^{md}$ and $(s, a) \in \mathcal{S} \times \mathcal{A}$, we obtain for all $(s, a) \in \mathcal{S} \times \mathcal{A}$ that

$$
\begin{aligned}
|\phi_i(s, a)^\top \theta_i - \phi_0(s, a)^\top W_{\text{init}}| &= \big|f\big((s, a); \theta_i\big) - f\big((s, a); W_{\text{init}}\big)\big| \\
&\leq \sup_{\theta \in \mathbb{R}^{md}} \big\|\nabla_\theta f\big((s, a); \theta\big)\big\|_2 \cdot \|\theta_i - W_{\text{init}}\|_2 \\
&= \sup_{\theta \in \mathbb{R}^{md}} \|\phi_\theta(s, a)\|_2 \cdot \|\theta_i - W_{\text{init}}\|_2 \leq R, \qquad \text{(I.13)}
\end{aligned}
$$

where the last inequality holds since $\theta_i \in \mathcal{B}$.

By plugging (I.12) and (I.13) into (I.9), we have

$$
|\tau_{i+1} \cdot \phi_i(s, a)^\top \theta_{i+1} - \tau_i \cdot \phi_i(s, a)^\top \theta_i| \leq \eta \cdot (M_0 + 3R), \quad \forall (s, a) \in \mathcal{S} \times \mathcal{A},
$$

where $M_0$ is defined in (I.10). Therefore, it holds for all $s \in \mathcal{S}$ that

$$
\begin{aligned}
\|\tau_{i+1} \cdot \phi_i(s, \cdot)^\top \theta_{i+1} - \tau_i \cdot \phi_i(s, \cdot)^\top \theta_i\|_{\infty, \mathcal{A}} &= \sup_{a \in \mathcal{A}} |\tau_{i+1} \cdot \phi_i(s, a)^\top \theta_{i+1} - \tau_i \cdot \phi_i(s, a)^\top \theta_i| \\
&\leq \eta \cdot (M_0 + 3R). \qquad \text{(I.14)}
\end{aligned}
$$

Finally, by the Pinsker's inequality, it follows from (I.14) that

$$
\begin{aligned}
&\|\phi_i(s, \cdot)^\top (\tau_{i+1} \cdot \theta_{i+1} - \tau_i \cdot \theta_i)\|_{\infty, \mathcal{A}} \cdot \|\Delta_i(\cdot \,|\, s)\|_{1, \mathcal{A}} - D_{\text{KL}}\big(\pi_{i+1}(\cdot \,|\, s)\big\|\pi_i(\cdot \,|\, s)\big) \\
&\leq \eta \cdot (M_0 + 3R) \cdot \|\pi_{i+1}(\cdot \,|\, s) - \pi_i(\cdot \,|\, s)\|_{1, \mathcal{A}} - 1/2 \cdot \|\pi_{i+1}(\cdot \,|\, s) - \pi_i(\cdot \,|\, s)\|_{1, \mathcal{A}}^2. \qquad \text{(I.15)}
\end{aligned}
$$

By completing the squares, we further upper bound the right-hand side of (I.15) by

$$
\begin{aligned}
&\|\phi_i(s, \cdot)^\top (\tau_{i+1} \cdot \theta_{i+1} - \tau_i \cdot \theta_i)\|_{\infty, \mathcal{A}} \cdot \|\Delta_i(\cdot \,|\, s)\|_{1, \mathcal{A}} - D_{\text{KL}}\big(\pi_{i+1}(\cdot \,|\, s)\big\|\pi_i(\cdot \,|\, s)\big) \\
&= -1/2 \cdot \big(\|\pi_{i+1}(\cdot \,|\, s) - \pi_i(\cdot \,|\, s)\|_{1, \mathcal{A}} - \eta \cdot (M_0 + 3R)\big)^2 + 1/2 \cdot \eta^2 \cdot (M_0 + 3R)^2 \\
&\leq 1/2 \cdot \eta^2 \cdot (M_0 + 3R)^2 \leq \eta^2 \cdot (M_0^2 + 9R^2), \qquad \text{(I.16)}
\end{aligned}
$$

which holds for all $s \in \mathcal{S}$. Here the last inequality follows from the fact that $(x + y)^2 \leq 2x^2 + 2y^2$.

**Upper Bounding (ii) in (I.8).** It holds for all $s \in \mathcal{S}$ that

$$
\begin{aligned}
&|\langle \phi_{i+1}(s, \cdot)^\top \theta_{i+1} - \phi_i(s, \cdot)^\top \theta_{i+1}, \Delta_i(\cdot \,|\, s)\rangle| \\
&\leq |\langle \phi_{i+1}(s, \cdot)^\top \theta_{i+1} - \phi_i(s, \cdot)^\top \theta_{i+1}, \pi_i(\cdot \,|\, s)\rangle| + |\langle \phi_{i+1}(s, \cdot)^\top \theta_{i+1} - \phi_i(s, \cdot)^\top \theta_{i+1}, \pi_{i+1}(\cdot \,|\, s)\rangle| \\
&\leq \|\phi_{i+1}(s, \cdot)^\top \theta_{i+1} - \phi_i(s, \cdot)^\top \theta_{i+1}\|_{\pi_i, 1} + \|\phi_{i+1}(s, \cdot)^\top \theta_{i+1} - \phi_i(s, \cdot)^\top \theta_{i+1}\|_{\pi_{i+1}, 1}. \qquad \text{(I.17)}
\end{aligned}
$$

Here for any distribution $\pi \in \mathcal{P}(\mathcal{A})$, we denote by $\|\cdot\|_{\pi, p}$ the $L_p(\pi)$-norm, which is defined by $\|v\|_{\pi, p} = [\sum_{a \in \mathcal{A}} \pi(a) \cdot |v(a)|^p]^{1/p}$. Following from Assumption 4.2 and Lemma E.2, it holds that

$$
\begin{aligned}
&\mathbb{E}\Big[\mathbb{E}_{\nu_*}\big[\|\phi_{i+1}(s, \cdot)^\top \theta_{i+1} - \phi_0(s, \cdot)^\top \theta_{i+1}\|_{\pi_i, 1}\big]\Big] \\
&\qquad \leq \mathbb{E}\big[\|\phi_{i+1}(\cdot, \cdot)^\top \theta_{i+1} - \phi_0(\cdot, \cdot)^\top \theta_{i+1}\|_{\pi_i \cdot \nu_*}\big] = O(R^{3/2} \cdot m^{-1/4}), \\
&\mathbb{E}\Big[\mathbb{E}_{\nu_*}\big[\|\phi_i(s, \cdot)^\top \theta_{i+1} - \phi_0(s, \cdot)^\top \theta_{i+1}\|_{\pi_i, 1}\big]\Big] \\
&\qquad \leq \mathbb{E}\big[\|\phi_i(\cdot, \cdot)^\top \theta_{i+1} - \phi_0(\cdot, \cdot)^\top \theta_{i+1}\|_{\pi_i \cdot \nu_*}\big] = O(R^{3/2} \cdot m^{-1/4}), \qquad \text{(I.18)}
\end{aligned}
$$

where the inequalities follow from the Cauchy-Schwartz inequality, and the expectations are taken over all the randomness. Meanwhile, it holds that

$$
\begin{aligned}
&\|\phi_{i+1}(s, \cdot)^\top \theta_{i+1} - \phi_i(s, \cdot)^\top \theta_{i+1}\|_{\pi_i, 1} \\
&\leq \|\phi_{i+1}(s, \cdot)^\top \theta_{i+1} - \phi_0(s, \cdot)^\top \theta_{i+1}\|_{\pi_i, 1} + \|\phi_i(s, \cdot)^\top \theta_{i+1} - \phi_0(s, \cdot)^\top \theta_{i+1}\|_{\pi_i, 1}. \qquad \text{(I.19)}
\end{aligned}
$$

Combining (I.18) and (I.19), we obtain that

$$
\mathbb{E}\Big[\mathbb{E}_{\nu_*}\big[\|\phi_{i+1}(s, \cdot)^\top \theta_{i+1} - \phi_i(s, \cdot)^\top \theta_{i+1}\|_{\pi_i, 1}\big]\Big] = O(R^{3/2} \cdot m^{-1/4}). \qquad \text{(I.20)}
$$

Similarly, it holds that

$$
\mathbb{E}\Big[\mathbb{E}_{\nu_*}\big[\|\phi_{i+1}(s, \cdot)^\top \theta_{i+1} - \phi_i(s, \cdot)^\top \theta_{i+1}\|_{\pi_{i+1}, 1}\big]\Big] = O(R^{3/2} \cdot m^{-1/4}), \qquad \text{(I.21)}
$$

where the expectation is taken over all the randomness. By plugging (I.20) and (I.21) into (I.17), we obtain that

$$\tau_{i+1} \cdot \mathbb{E}\Big[\mathbb{E}_{\nu_*}\big[|\langle \phi_{i+1}(s,\cdot)^\top \theta_{i+1} - \phi_i(s,\cdot)^\top \theta_{i+1}, \Delta_i(\cdot\,|\,s)\rangle|\big]\Big] = O(\tau_{i+1} \cdot R^{3/2} \cdot m^{-1/4}). \quad \text{(I.22)}$$

Finally, by plugging (I.16) and (I.22) into (I.8), it holds under Assumptions 4.2 and A.3 that

$$\mathbb{E}\bigg[\mathbb{E}_{\nu_*}\Big[\big|\big\langle \log\big(\pi_{i+1}(\cdot\,|\,s)/\pi_i(\cdot\,|\,s)\big), \pi_i(\cdot\,|\,s) - \pi_{i+1}(\cdot\,|\,s)\big\rangle\big|\Big]\bigg]$$

$$\leq \mathbb{E}\bigg[\mathbb{E}_{\nu_*}\Big[D_{\mathrm{KL}}\big(\pi_{i+1}(\cdot\,|\,s)\big\|\pi_i(\cdot\,|\,s)\big)\Big]\bigg] + \eta^2 \cdot (9R^2 + M^2) + O(\tau_{i+1} \cdot R^{3/2} \cdot m^{-1/4}),$$

where $M$ is the absolute constant defined in Assumption A.3. Thus, we complete the proof of Lemma H.2. $\qquad\square$

### I.3    PROOF OF LEMMA H.3

*Proof.* Note that $\mathbb{E}_{\pi_{\theta_i}}[\phi_{\theta_i}(s,a')]$ and $\mathbb{E}_{\pi_{\theta_i}}[\phi_{\omega_i}(s,a')]$ depend solely on $s \in \mathcal{S}$, where we write $\mathbb{E}_{\pi_{\theta_i}}[\phi_{\theta_i}(s,a')] = \mathbb{E}_{a' \sim \pi_{\theta_i}(\cdot\,|\,s)}[\phi_{\theta_i}(s,a')]$ for notational simplicity. Thus, we have

$$\big\langle \mathbb{E}_{\pi_{\theta_i}}\big[\phi_{\theta_i}(s,a')^\top \delta_i - \phi_{\omega_i}(s,a')^\top \omega_i\big], \pi^*(\cdot\,|\,s) - \pi_i(\cdot\,|\,s)\big\rangle = 0, \quad \forall s \in \mathcal{S}. \quad \text{(I.23)}$$

Meanwhile, following from the parameterization of $\pi_\theta$ in (3.2) and (I.6) in §I.2, we obtain that

$$\big\langle \log\big(\pi_{i+1}(\cdot\,|\,s)/\pi_i(\cdot\,|\,s)\big) - \eta \cdot Q_{\omega_i}(s,\cdot), \pi^*(\cdot\,|\,s) - \pi_i(\cdot\,|\,s)\big\rangle$$

$$= \big\langle \tau_{i+1} \cdot \phi_{\theta_{i+1}}(s,\cdot)^\top \theta_{i+1} - \tau_i \cdot \phi_{\theta_i}(s,\cdot)^\top \theta_i - \eta \cdot \phi_{\omega_i}(s,\cdot)^\top \omega_i, \pi^*(\cdot\,|\,s) - \pi_i(\cdot\,|\,s)\big\rangle. \quad \text{(I.24)}$$

In what follows, we define $\Delta_i^*(\cdot\,|\,\cdot) = \pi^*(\cdot\,|\,\cdot) - \pi_i(\cdot\,|\,\cdot)$ for notational simplicity. Then, combining (I.23) and (I.24), we obtain for all $s \in \mathcal{S}$ that

$$\big\langle \log\big(\pi_{i+1}(\cdot\,|\,s)/\pi_i(\cdot\,|\,s)\big) - \eta \cdot Q_{\omega_i}(s,\cdot), \Delta_i^*(\cdot\,|\,s)\big\rangle$$

$$= \eta \cdot \big\langle \phi_{\theta_i}(s,\cdot)^\top \delta_i - \phi_{\omega_i}(s,\cdot)^\top \omega_i, \Delta_i^*(\cdot\,|\,s)\big\rangle$$

$$\qquad + \tau_{i+1} \cdot \big\langle \phi_{\theta_{i+1}}(s,\cdot)^\top \theta_{i+1} - \phi_{\theta_i}(s,\cdot)^\top \theta_{i+1}, \Delta_i^*(\cdot\,|\,s)\big\rangle$$

$$= \underbrace{\eta \cdot \big\langle \overline{\phi}_{\theta_i}(s,\cdot)^\top \delta_i - \overline{\phi}_{\omega_i}(s,\cdot)^\top \omega_i, \Delta_i^*(\cdot\,|\,s)\big\rangle}_{\text{(iii)}} \quad \text{(I.25)}$$

$$\qquad + \underbrace{\tau_{i+1} \cdot \big\langle \phi_{\theta_{i+1}}(s,\cdot)^\top \theta_{i+1} - \phi_{\theta_i}(s,\cdot)^\top \theta_{i+1}, \Delta_i^*(\cdot\,|\,s)\big\rangle}_{\text{(iv)}},$$

where $\overline{\phi}_{\theta_i}$ and $\overline{\phi}_{\omega_i}$ are the centered feature mappings defined in (B.2) that correspond to $\theta_i$ and $\omega_i$, respectively, and $\delta_i$ is defined by

$$\delta_i = \eta^{-1} \cdot (\tau_{i+1} \cdot \theta_{i+1} - \tau_i \cdot \theta_i) = \operatorname*{argmin}_{\omega \in \mathcal{B}} \|\widehat{F}(\theta_i)\omega - \tau_i \cdot \widehat{\nabla} J(\pi_{\theta_i})\|_2. \quad \text{(I.26)}$$

In what follows, we upper bound the expectations of (iii) and (iv) over all the randomness separately.

**Upper Bounding (iii) in** (I.25)**.** It holds that

$$\mathbb{E}_{\nu_*}\big[|\langle \overline{\phi}_{\theta_i}(s,\cdot)^\top \delta_i - \overline{\phi}_{\omega_i}(s,\cdot)^\top \omega_i, \pi^*(\cdot\,|\,s)\rangle|\big]$$

$$\leq \int_{\mathcal{S}\times\mathcal{A}} |\overline{\phi}_{\theta_i}(s,a)^\top \delta_i - \overline{\phi}_{\omega_i}(s,a)^\top \omega_i| \mathrm{d}\sigma_*(s,a)$$

$$= \int_{\mathcal{S}\times\mathcal{A}} |\overline{\phi}_{\theta_i}(s,a)^\top \delta_i - \overline{\phi}_{\omega_i}(s,a)^\top \omega_i| \cdot \frac{\mathrm{d}\sigma_*}{\mathrm{d}\sigma_i}(s,a) \mathrm{d}\sigma_i(s,a)$$

$$\leq \varphi_i \cdot \|\overline{\phi}_{\theta_i}(\cdot,\cdot)^\top \delta_i - \overline{\phi}_{\omega_i}(\cdot,\cdot)^\top \omega_i\|_{\sigma_i}, \quad \text{(I.27)}$$

where $\mathrm{d}\sigma_*/\mathrm{d}\sigma_i$ is the Radon-Nikodym derivative, $\varphi_i$ is defined in (A.1) of Assumption A.2, and the last inequality follows from the Cauchy-Schwartz inequality. Similarly, it holds that

$$
\begin{aligned}
\mathbb{E}_{\nu_*}\big[|\langle\overline{\phi}_{\theta_i}&(s,a)^\top\delta_i - \overline{\phi}_{\omega_i}(s,a)^\top\omega_i, \pi_i(a\,|\,s)\rangle|\big] \\
&\leq \int_{\mathcal{S}\times\mathcal{A}}|\overline{\phi}_{\theta_i}(s,a)^\top\delta_i - \overline{\phi}_{\omega_i}(s,a)^\top\omega_i|\mathrm{d}\pi_i(a\,|\,s)\cdot\nu_*(s) \\
&= \int_{\mathcal{S}\times\mathcal{A}}|\overline{\phi}_{\theta_i}(s,a)^\top\delta_i - \overline{\phi}_{\omega_i}(s,a)^\top\omega_i|\cdot\frac{\mathrm{d}\nu_*}{\mathrm{d}\nu_i}(s)\mathrm{d}\sigma_i(s,a) \\
&\leq \psi_i\cdot\|\overline{\phi}_{\theta_i}(\cdot,\cdot)^\top\delta_i - \overline{\phi}_{\omega_i}(\cdot,\cdot)^\top\omega_i\|_{\sigma_i},
\end{aligned}
\tag{I.28}
$$

where $\mathrm{d}\nu_*/\mathrm{d}\nu_i$ is the Radon-Nikodym derivative, $\psi_i$ is defined in (A.1) of Assumption A.2, and the last inequality follows from the Cauchy-Schwartz inequality. Combining (I.27) and (I.28), we obtain that

$$
\begin{aligned}
\mathbb{E}_{\nu_*}\big[|\langle\overline{\phi}_{\theta_i}(s,\cdot)^\top&\delta_i - \overline{\phi}_{\omega_i}(s,\cdot)^\top\omega_i, \Delta_i^*(\cdot\,|\,s)\rangle|\big] \\
&\leq (\varphi_i + \psi_i)\cdot\|\overline{\phi}_{\theta_i}(\cdot,\cdot)^\top\delta_i - \overline{\phi}_{\omega_i}(\cdot,\cdot)^\top\omega_i\|_{\sigma_i}.
\end{aligned}
\tag{I.29}
$$

It now suffices to upper bound $\|\overline{\phi}_{\theta_i}(\cdot,\cdot)^\top\delta_i - \overline{\phi}_{\omega_i}(\cdot,\cdot)^\top\omega_i\|_{\sigma_i}$. With a slight abuse of notation, we write $\overline{\phi}_{\theta_i} = \overline{\phi}_{\theta_i}(\cdot,\cdot)$ and $\overline{\phi}_{\omega_i} = \overline{\phi}_{\omega_i}(\cdot,\cdot)$ hereafter for notational simplicity. Note that

$$
\begin{aligned}
\|\delta_i^\top\overline{\phi}_{\theta_i} - \omega_i^\top\overline{\phi}_{\omega_i}\|_{\sigma_i} &= \sqrt{\mathbb{E}_{\sigma_i}\big[(\delta_i^\top\overline{\phi}_{\theta_i} - \omega_i^\top\overline{\phi}_{\omega_i})\cdot(\delta_i^\top\overline{\phi}_{\theta_i} - \omega_i^\top\overline{\phi}_{\omega_i})\big]} \\
&\leq \underbrace{\sqrt{\big|(\delta_i - \omega_i)^\top\mathbb{E}_{\sigma_i}\big[\overline{\phi}_{\theta_i}\cdot(\delta_i^\top\overline{\phi}_{\theta_i} - \omega_i^\top\overline{\phi}_{\omega_i})\big]\big|}}_{\text{(iii.a)}} \\
&\quad + \underbrace{\sqrt{\mathbb{E}_{\sigma_i}\big[(\omega_i^\top\overline{\phi}_{\theta_i} - \omega_i^\top\overline{\phi}_{\omega_i})\cdot(\delta_i^\top\overline{\phi}_{\theta_i} - \omega_i^\top\overline{\phi}_{\omega_i})\big]}}_{\text{(iii.b)}}.
\end{aligned}
\tag{I.30}
$$

We now upper bound the expectations of the right-hand side of (I.30) over all the randomness.

**Upper Bounding (iii.a) in** (I.30). Note that $\omega_i, \delta_i \in \mathcal{B}$, where $\delta_i$ is defined in (I.26) and $\mathcal{B} = \{\alpha \in \mathbb{R}^{md} : \|\alpha - W_{\text{init}}\|_2 \leq R\}$. Therefore, we obtain that

$$
\|\omega_i - \delta_i\|_2 \leq 2R.
\tag{I.31}
$$

Meanwhile, following from Proposition 3.1 and (3.7), it holds that

$$
\begin{aligned}
\mathbb{E}_{\sigma_i}\big[\widehat{F}(\theta_i)\big] &= F(\theta_i) = \tau_i^2\cdot\mathbb{E}_{\sigma_i}\big[\overline{\phi}_{\theta_i}(\overline{\phi}_{\theta_i})^\top\big], \\
\mathbb{E}_{\sigma_i}\big[\widehat{\nabla}_\theta J(\pi_{\theta_i})\big] &= \tau_i\cdot\mathbb{E}_{\sigma_i}\big[\overline{\phi}_{\theta_i}\cdot(\overline{\phi}_{\omega_i})^\top\omega_i\big],
\end{aligned}
\tag{I.32}
$$

where the expectations are taken over $\sigma_i$ given $\theta_i$ and $\omega_i$. In what follows, we write $g_i = \mathbb{E}_{\sigma_i}[\widehat{\nabla}J(\pi_{\theta_i})]$ for notational simplicity, where the expectation is taken over $\sigma_i$ given $\theta_i$ and $\omega_i$. By plugging (I.31) and (I.32) into (iii.a) in (I.30), we obtain that

$$
\begin{aligned}
\Big|(\delta_i - \omega_i)^\top\mathbb{E}_{\sigma_i}\Big[\big(\overline{\phi}_{\theta_i}\cdot(\delta_i^\top\overline{\phi}_{\theta_i} - \omega_i^\top\overline{\phi}_{\omega_i})\big)\Big]\Big| &= \tau_i^{-2}\cdot\big|(\delta_i - \omega_i)^\top\big(F(\theta_i)\cdot\delta_i - \tau_i\cdot g_i\big)\big| \\
&\leq 2R\cdot\tau_i^{-2}\cdot\|F(\theta_i)\cdot\delta_i - \tau_i\cdot g_i\|_2,
\end{aligned}
\tag{I.33}
$$

where the last inequality follows from the Cauchy-Schwartz inequality and (I.31). By (I.33), we have

$$
\begin{aligned}
\mathbb{E}\Big[\big|(\delta_i - \omega_i)^\top\mathbb{E}_{\sigma_i}\big[\overline{\phi}_{\theta_i}(\delta_i^\top\overline{\phi}_{\theta_i} - \omega_i^\top\overline{\phi}_{\omega_i})\big]\big|^{1/2}\Big] &\leq C_i\cdot\mathbb{E}\Big[\big(\big\|F(\theta_i)\cdot\delta_i - \tau_i\cdot g_i]\big\|_2\big)^{1/2}\Big] \\
&\leq C_i\cdot\mathbb{E}\Big[\big(\|\widehat{F}(\theta_i)\cdot\delta_i - \tau_i\cdot\widehat{\nabla}_\theta J(\pi_{\theta_i})\|_2 + \|\xi_i(\delta_i)\|_2\big)^{1/2}\Big] \\
&\leq C_i\cdot\Big\{\mathbb{E}\big[\|\widehat{F}(\theta_i)\cdot\delta_i - \tau_i\cdot\widehat{\nabla}_\theta J(\pi_{\theta_i})\|_2\big] + \mathbb{E}\big[\|\xi_i(\delta_i)\|_2\big]\Big\}^{1/2},
\end{aligned}
\tag{I.34}
$$

where the expectations are taken over all the randomness. Here the last inequality follows from the Jensen's inequality, $C_i = \sqrt{2R} \cdot \tau_i^{-1}$, and $\xi_i(\delta_i)$ is defined by

$$\xi_i(\delta_i) = \widehat{F}(\theta_i) \cdot \delta_i - \tau_i \cdot \widehat{\nabla}_\theta J(\pi_{\theta_i}) - \big(F(\theta_i) \cdot \delta_i - \tau_i \cdot g_i\big). \tag{I.35}$$

In what follows, we upper bound $\|\widehat{F}(\theta_i) \cdot \delta_i - \tau_i \cdot \widehat{\nabla}_\theta J(\pi_{\theta_i})\|_2$ on the right-hand side of (I.34). Recall that we define $\delta_i$ by

$$\delta_i = \eta^{-1} \cdot (\tau_{i+1} \cdot \theta_{i+1} - \tau_i \cdot \theta_i) = \underset{\omega \in \mathcal{B}}{\arg\min} \|\widehat{F}(\theta_i) \cdot \omega_i - \tau_i \cdot \widehat{\nabla}_\theta J(\pi_{\theta_i})\|_2. \tag{I.36}$$

Therefore, since $\omega_i \in \mathcal{B}$, we obtain from (I.36) that

$$\|\widehat{F}(\theta_i) \cdot \delta_i - \tau_i \cdot \widehat{\nabla}_\theta J(\pi_{\theta_i})\|_2 \leq \|\widehat{F}(\theta_i) \cdot \omega_i - \tau_i \cdot \widehat{\nabla}_\theta J(\pi_{\theta_i})\|_2$$
$$\leq \|F(\theta_i) \cdot \omega_i - \tau_i \cdot g_i\|_2 + \|\xi_i(\omega_i)\|_2, \tag{I.37}$$

where recall that, similar to (I.35), we define $\xi_i(\omega_i)$ by

$$\xi_i(\omega_i) = \widehat{F}(\theta_i) \cdot \omega_i - \tau_i \cdot \widehat{\nabla}_\theta J(\pi_{\theta_i}) - \big(F(\theta_i) \cdot \omega_i - \tau_i \cdot g_i\big). \tag{I.38}$$

By plugging (I.37) into (I.34), we obtain that

$$\mathbb{E}\Big[\big|(\delta_i - \omega_i)^\top \mathbb{E}_{\sigma_i}\big[\overline{\phi}_{\theta_i} \cdot (\delta_i^\top \overline{\phi}_{\theta_i} - \omega_i^\top \overline{\phi}_{\omega_i})\big]\big|^{1/2}\Big]$$
$$\leq C_i \cdot \Big\{\mathbb{E}\big[\|F(\theta_i) \cdot \omega_i - \tau_i \cdot g_i\|_2\big] + \mathbb{E}\big[\|\xi_i(\delta_i)\|_2\big] + \mathbb{E}\big[\|\xi_i(\omega_i)\|_2\big]\Big\}^{1/2}, \tag{I.39}$$

where $C_i = \sqrt{2R} \cdot \tau_i^{-1}$ and $\xi_i(\delta_i), \xi_i(\omega_i)$ are defined in (I.35) and (I.38), respectively. To upper bound the right-hand side of (I.39), it now suffices to upper bound the expectation $\mathbb{E}[\|F(\theta_i) \cdot \omega_i - \tau_i \cdot g_i\|_2]$. By (I.32), we obtain that

$$\|F(\theta_i) \cdot \omega_i - \tau_i \cdot g_i\|_2 = \tau_i^2 \cdot \big\|\mathbb{E}_{\sigma_i}\big[\overline{\phi}_{\theta_i} \cdot (\overline{\phi}_{\theta_i} - \overline{\phi}_{\omega_i})^\top \omega_i\big]\big\|_2$$
$$\leq \tau_i^2 \cdot \mathbb{E}_{\sigma_i}\big[\|\overline{\phi}_{\theta_i} \cdot (\overline{\phi}_{\theta_i} - \overline{\phi}_{\omega_i})^\top \omega_i\|_2\big] = \tau_i^2 \cdot \mathbb{E}_{\sigma_i}\big[\|\overline{\phi}_{\theta_i}\|_2 \cdot |(\overline{\phi}_{\theta_i} - \overline{\phi}_{\omega_i})^\top \omega_i|\big], \tag{I.40}$$

where the inequality follows from the Jensen's inequality. In what follows, we upper bound the right-hand side of (I.40). Note that $\|\overline{\phi}_{\theta_i}(s,a)\|_2 \leq 2$ for all $(s,a) \in \mathcal{S} \times \mathcal{A}$. By further plugging into (I.40), we obtain that

$$\|F(\theta_i) \cdot \omega_i - \tau_i \cdot g_i\|_2 \leq 2\tau_i^2 \cdot \mathbb{E}_{\sigma_i}\big[|(\overline{\phi}_{\theta_i} - \overline{\phi}_{\omega_i})^\top \omega_i|\big] \leq 2\tau_i^2 \cdot \|(\overline{\phi}_{\theta_i} - \overline{\phi}_{\omega_i})^\top \omega_i\|_{\sigma_i}, \tag{I.41}$$

where the last inequality follows from the Jensen's inequality. Recall that $\omega_i, \theta_i \in \mathcal{B}$. Therefore, by Assumption 4.2 and Corollary E.3, we have

$$\mathbb{E}\big[\|(\overline{\phi}_{\theta_i} - \overline{\phi}_{\omega_i})^\top \omega_i\|_{\sigma_i}\big]$$
$$\leq \mathbb{E}\big[\|(\overline{\phi}_{\theta_i} - \overline{\phi}_0)^\top \omega_i\|_{\sigma_i}\big] + \mathbb{E}\big[\|(\overline{\phi}_0 - \overline{\phi}_{\omega_i})^\top \omega_i\|_{\sigma_i}\big] = \mathcal{O}(R^{3/2} \cdot m^{-1/4}), \tag{I.42}$$

where the expectations are taken over all the randomness. Combining (I.41) and (I.42), we obtain that

$$\mathbb{E}\big[\|F(\theta_i) \cdot \omega_i - \tau_i \cdot g_i\|_2\big] = \mathcal{O}(2\tau_i^2 \cdot R^{3/2} \cdot m^{-1/4}), \tag{I.43}$$

where the expectation is taken over all the randomness. Finally, by plugging (I.43) into (I.39), we conclude that

$$\mathbb{E}\Big[\big|(\delta_i - \omega_i)^\top \mathbb{E}_{\sigma_i}\big[\overline{\phi}_{\theta_i} \cdot (\delta_i^\top \overline{\phi}_{\theta_i} - \omega_i^\top \overline{\phi}_{\omega_i})\big]\big|^{1/2}\Big]$$
$$\leq C_i \cdot \Big\{\mathbb{E}\big[\|F(\theta_i) \cdot \omega_i - \tau_i \cdot g_i\|_2\big] + \mathbb{E}\big[\|\xi_i(\omega_i)\|_2\big] + \mathbb{E}\big[\|\xi_i(\delta_i)\|_2\big]\Big\}^{1/2}$$
$$= \mathcal{O}(R^{5/4} \cdot m^{-1/8}) + \sqrt{2R} \cdot \tau_i^{-1} \cdot \Big\{\mathbb{E}\big[\|\xi_i(\delta_i)\|_2 + \|\xi_i(\omega_i)\|_2\big]\Big\}^{1/2}, \tag{I.44}$$

where $C_i = \sqrt{2R} \cdot \tau_i^{-1}$ and $\xi_i(\delta_i), \xi_i(\omega_i)$ are defined in Assumption A.1.

**Upper Bounding (iii.b) in** (I.30)**.** Following from the Cauchy-Schwartz inequality, it holds that

$$
\sqrt{\mathbb{E}_{\sigma_i}\big[(\omega_i^\top\overline{\phi}_{\theta_i} - \omega_i^\top\overline{\phi}_{\omega_i})\cdot(\delta_i^\top\overline{\phi}_{\theta_i} - \omega_i^\top\overline{\phi}_{\omega_i})\big]}
$$
$$
\leq \big(\|\omega_i^\top\overline{\phi}_{\theta_i} - \omega_i^\top\overline{\phi}_{\omega_i}\|_{\sigma_i} \cdot \|\delta_i^\top\overline{\phi}_{\theta_i} - \omega_i^\top\overline{\phi}_{\omega_i}\|_{\sigma_i}\big)^{1/2}. \tag{I.45}
$$

To upper bound the right-hand side of (I.45), we first upper bound $\|\omega_i^\top\overline{\phi}_{\theta_i} - \omega_i^\top\overline{\phi}_{\omega_i}\|_{\sigma_i}$. Recall that $\omega_i, \theta_i \in \mathcal{B}$. Following from Assumption 4.2 and Corollary E.3, it holds that

$$
\mathbb{E}\big[\|\omega_i^\top\overline{\phi}_{\theta_i} - \omega_i^\top\overline{\phi}_0\|_{\sigma_i}^2\big] = \mathcal{O}(R^3\cdot m^{-1/2}),
$$
$$
\mathbb{E}\big[\|\omega_i^\top\overline{\phi}_{\omega_i} - \omega_i^\top\overline{\phi}_0\|_{\sigma_i}^2\big] = \mathcal{O}(R^3\cdot m^{-1/2}), \tag{I.46}
$$

where $\overline{\phi}_0$ is defined in (B.1) and the expectations are taken over all the randomness. Therefore, following from (I.46), we obtain that

$$
\mathbb{E}\big[\|\omega_i^\top\overline{\phi}_{\theta_i} - \omega_i^\top\overline{\phi}_{\omega_i}\|_{\sigma_i}^2\big]
$$
$$
\leq 2\mathbb{E}\big[\|\omega_i^\top\overline{\phi}_{\theta_i} - \omega_i^\top\overline{\phi}_0\|_{\sigma_i}^2\big] + 2\mathbb{E}\big[\|\omega_i^\top\overline{\phi}_{\omega_i} - \omega_i^\top\overline{\phi}_0\|_{\sigma_i}^2\big] = \mathcal{O}(R^3\cdot m^{-1/2}). \tag{I.47}
$$

It remains to upper bound $\|\delta_i^\top\overline{\phi}_{\theta_i} - \omega_i^\top\overline{\phi}_{\omega_i}\|_{\sigma_i}$ on the right-hand side of (I.45). Since $\delta_i \in \mathcal{B}$, by Assumption 4.2 and Corollary E.3, we obtain that

$$
\mathbb{E}\big[\|\delta_i^\top\overline{\phi}_{\theta_i} - \delta_i^\top\overline{\phi}_0\|_{\sigma_i}^2\big] = \mathcal{O}(R^3\cdot m^{-1/2}), \tag{I.48}
$$

where the expectation is taken over all the randomness. Meanwhile, following from the fact that $\|\overline{\phi}_0(s,a)\|_2 \leq 2$ for all $(s,a) \in \mathcal{S}\times\mathcal{A}$, we obtain that

$$
|\delta_i^\top\overline{\phi}_0(s,a) - \omega_i^\top\overline{\phi}_0(s,a)|
$$
$$
\leq \|\overline{\phi}_0(s,a)\|_2 \cdot \|\delta_i - \omega_i\|_2 \leq 4R, \quad \forall(s,a) \in \mathcal{S}\times\mathcal{A}, \tag{I.49}
$$

where the first inequality follows from the Cauchy-Schwartz inequality and the second inequality follows from the fact that $\delta_i, \omega_i \in \mathcal{B}$. Combining (I.46), (I.48), and (I.49), we obtain that

$$
\mathbb{E}\big[\|\delta_i^\top\overline{\phi}_{\theta_i} - \omega_i^\top\overline{\phi}_{\omega_i}\|_{\sigma_i}^2\big] \leq 3\mathbb{E}\big[\|\delta_i^\top\overline{\phi}_{\theta_i} - \delta_i^\top\overline{\phi}_0\|_{\sigma_i}^2\big] + 3\mathbb{E}\big[\|\delta_i^\top\overline{\phi}_0 - \omega_i^\top\overline{\phi}_0\|_{\sigma_i}^2\big]
$$
$$
+ 3\mathbb{E}\big[\|\omega_i^\top\overline{\phi}_{\omega_i} - \omega_i^\top\overline{\phi}_0\|_{\sigma_i}^2\big] = \mathcal{O}(R^2 + R^3\cdot m^{-1/2}), \tag{I.50}
$$

where the expectations are taken over all the randomness. Finally, plugging (I.47) and (I.50) into (I.45), we obtain that

$$
\mathbb{E}\bigg[\Big\{\mathbb{E}_{\sigma_i}\big[(\omega_i^\top\overline{\phi}_{\theta_i} - \omega_i^\top\overline{\phi}_{\omega_i})(\delta_i^\top\overline{\phi}_{\theta_i} - \omega_i^\top\overline{\phi}_{\omega_i})\big]\Big\}^{1/2}\bigg]
$$
$$
\leq \Big\{\mathbb{E}\big[\|\omega_i^\top\overline{\phi}_{\theta_i} - \omega_i^\top\overline{\phi}_{\omega_i}\|_{\sigma_i} \cdot \|\delta_i^\top\overline{\phi}_{\theta_i} - \omega_i^\top\overline{\phi}_{\omega_i}\|_{\sigma_i}\big]\Big\}^{1/2}
$$
$$
\leq \Big\{\mathbb{E}\big[\|\omega_i^\top\overline{\phi}_{\theta_i} - \omega_i^\top\overline{\phi}_{\omega_i}\|_{\sigma_i}^2\big] \cdot \mathbb{E}\big[\|\delta_i^\top\overline{\phi}_{\theta_i} - \omega_i^\top\overline{\phi}_{\omega_i}\|_{\sigma_i}^2\big]\Big\}^{1/4}
$$
$$
= \mathcal{O}(R^{3/2}\cdot m^{-1/4} + R^{5/4}\cdot m^{-1/8}), \tag{I.51}
$$

where the inequalities follow from the Cauchy-Schwartz inequality and the expectations are taken over all the randomness.

Finally, by plugging (I.44), (I.51), and (I.30) into (I.29), we obtain that

$$
\mathbb{E}\Big[\mathbb{E}_{\nu_*}\big[|\langle\overline{\phi}_{\theta_i}(s,\cdot)^\top\delta_i - \overline{\phi}_{\omega_i}(s,\cdot)^\top\omega_i, \Delta_i^*(\cdot\,|\,s)\rangle|\big]\Big]
$$
$$
= \eta\cdot(\varphi_i + \psi_i)\cdot\Big(\mathcal{O}(R^{5/4}\cdot m^{-1/8} + R^{3/2}\cdot m^{-1/4}) \tag{I.52}
$$
$$
+ \sqrt{2R}\cdot\tau_i^{-1}\cdot\Big\{\mathbb{E}\big[\|\xi_i(\delta_i)\|_2 + \|\xi_i(\omega_i)\|_2\big]\Big\}^{1/2}\Big),
$$

where $\xi_i(\delta_i)$ and $\xi_i(\omega_i)$ are defined in Assumption A.1. Here the expectations are taken over all the randomness.

**Upper Bounding (iv) in** (I.25)**.** The analysis of (iv) is similar to that of (ii) in §H.8. It holds that

$$
\begin{aligned}
&|\langle \phi_{\theta_{i+1}}(s,\cdot)^\top \theta_{i+1} - \phi_{\theta_i}(s,\cdot)^\top \theta_{i+1}, \Delta_i^*(\cdot \,|\, s)\rangle| \\
&\quad \le |\langle \phi_{\theta_{i+1}}(s,\cdot)^\top \theta_{i+1} - \phi_{\theta_i}(s,\cdot)^\top \theta_{i+1}, \pi^*(\cdot \,|\, s)\rangle| + |\langle \phi_{\theta_{i+1}}(s,\cdot)^\top \theta_{i+1} - \phi_{\theta_i}(s,\cdot)^\top \theta_{i+1}, \pi_i(\cdot \,|\, s)\rangle| \\
&\quad \le \|\phi_{\theta_{i+1}}(s,\cdot)^\top \theta_{i+1} - \phi_{\theta_i}(s,\cdot)^\top \theta_{i+1}\|_{\pi^*,1} + \|\phi_{\theta_{i+1}}(s,\cdot)^\top \theta_{i+1} - \phi_{\theta_i}(s,\cdot)^\top \theta_{i+1}\|_{\pi_i,1}. \quad \text{(I.53)}
\end{aligned}
$$

Note that $\theta_i, \theta_{i+1} \in \mathcal{B}$. Following from Assumption 4.2 and Lemma E.2, it holds that

$$
\begin{aligned}
&\mathbb{E}\Big[\mathbb{E}_{\nu_*}\big[\|\phi_{\theta_{i+1}}(s,\cdot)^\top \theta_{i+1} - \phi_0(s,\cdot)^\top \theta_{i+1}\|_{\pi^*,1}\big]\Big] \\
&\qquad \le \mathbb{E}\big[\|\phi_{\theta_{i+1}}(\cdot,\cdot)^\top \theta_{i+1} - \phi_0(\cdot,\cdot)^\top \theta_{i+1}\|_{\sigma_*}\big] = \mathcal{O}(R^{3/2} \cdot m^{-1/4}), \\
&\mathbb{E}\Big[\mathbb{E}_{\nu_*}\big[\|\phi_{\theta_i}(s,\cdot)^\top \theta_{i+1} - \phi_0(s,\cdot)^\top \theta_{i+1}\|_{\pi^*,1}\big]\Big] \\
&\qquad \le \mathbb{E}\big[\|\phi_{\theta_i}(\cdot,\cdot)^\top \theta_{i+1} - \phi_0(\cdot,\cdot)^\top \theta_{i+1}\|_{\sigma_*}\big] = \mathcal{O}(R^{3/2} \cdot m^{-1/4}), \quad \text{(I.54)}
\end{aligned}
$$

where the inequalities follow from the Jensen's inequality, $\phi_0$ is the feature mapping defined in (3.3) with $\theta = W_{\text{init}}$, and the expectations are taken over all the randomness. Following from (I.54), we obtain that

$$
\begin{aligned}
&\mathbb{E}\Big[\mathbb{E}_{\nu_*}\big[\|\phi_{\theta_{i+1}}(s,\cdot)^\top \theta_{i+1} - \phi_{\theta_i}(s,\cdot)^\top \theta_{i+1}\|_{\pi^*,1}\big]\Big] \\
&\qquad \le \mathbb{E}\Big[\mathbb{E}_{\nu_*}\big[\|\phi_{\theta_{i+1}}(s,\cdot)^\top \theta_{i+1} - \phi_0(s,\cdot)^\top \theta_{i+1}\|_{\pi^*,1}\big]\Big] \\
&\qquad\quad + \mathbb{E}\Big[\mathbb{E}_{\nu_*}\big[\|\phi_{\theta_i}(s,\cdot)^\top \theta_{i+1} - \phi_0(s,\cdot)^\top \theta_{i+1}\|_{\pi^*,1}\big]\Big] \\
&\qquad = \mathcal{O}(R^{3/2} \cdot m^{-1/4}), \quad \text{(I.55)}
\end{aligned}
$$

where the expectations are taken over all the randomness. Similarly, it holds that

$$
\mathbb{E}\Big[\mathbb{E}_{\nu_*}\big[\|\phi_{\theta_{i+1}}(s,\cdot)^\top \theta_{i+1} - \phi_{\theta_i}(s,\cdot)^\top \theta_{i+1}\|_{\pi_i,1}\big]\Big] = \mathcal{O}(R^{3/2} \cdot m^{-1/4}). \quad \text{(I.56)}
$$

By plugging (I.55) and (I.56) into (I.53), we obtain that

$$
\mathbb{E}\Big[\mathbb{E}_{\nu_*}\big[|\langle \phi_{\theta_{i+1}}(s,\cdot)^\top \theta_{i+1} - \phi_{\theta_i}(s,\cdot)^\top \theta_{i+1}, \Delta_i^*(\cdot \,|\, s)\rangle|\big]\Big] = \mathcal{O}(R^{3/2} \cdot m^{-1/4}). \quad \text{(I.57)}
$$

Finally, by plugging (I.52) and (I.57) into (I.25), we obtain that

$$
\begin{aligned}
&\mathbb{E}\Big[\mathbb{E}_{\nu_*}\big[|\langle \log(\pi_{i+1}(\cdot \,|\, s)/\pi_i(\cdot \,|\, s)) - \eta \cdot Q_{\omega_i}(s,\cdot), \pi^*(\cdot \,|\, s) - \pi_i(\cdot \,|\, s)\rangle|\big]\Big] \\
&\quad \le \sqrt{2}(\varphi_i + \psi_i) \cdot \eta \cdot R^{1/2} \cdot \tau_i^{-1} \cdot \Big\{\mathbb{E}\big[\|\xi_i(\delta_i)\|_2\big] + \mathbb{E}\big[\|\xi_i(\omega_i)\|_2\big]\Big\}^{1/2} \\
&\qquad + \mathcal{O}\big((\tau_{i+1} + 1) \cdot R^{3/2} \cdot m^{-1/4} + \eta \cdot R^{5/4} \cdot m^{-1/8}\big),
\end{aligned}
$$

where $\varphi_i, \psi_i$ are defined in Assumption A.2 and $\xi_i(\delta_i), \xi_i(\omega_i)$ are defined in Assumption A.1. Thus, we complete the proof of Lemma H.3. $\qquad\square$

## J   AUXILLIARY LEMMA

**Lemma J.1** (Performance Difference (Kakade and Langford, 2002))**.** It holds for any $\pi$ and $\widetilde{\pi}$ that

$$
J(\widetilde{\pi}) - J(\pi) = (1-\gamma)^{-1} \cdot \mathbb{E}_{\widetilde{\pi}\cdot\nu_{\widetilde{\pi}}}\big[A^\pi(s,a)\big].
$$

Here $\nu_{\widetilde{\pi}}$ is the state visitation measure corresponding to $\widetilde{\pi}$, which is defined in (2.3).

*Proof.* See Kakade and Langford (2002) for a detailed proof. $\qquad\square$

