# OpenReview forum: "Neural Policy Gradient Methods: Global Optimality and Rates of Convergence"
_ICLR.cc/2020/Conference — Accept (Poster)_

### Official Review · AnonReviewer1 · 2019-10-23
**Official Blind Review #1**

**Rating:** 8

**Review:**

This paper provides theoretical studies for neural policy gradient descents for reinforcement learning problems. The authors prove global optimality and rates of convergence of neural natural/vanilla policy gradient. Their results rely on the key factor for "compatibility" between the actor and critic. This is ensured by sharing neural architectures and random initializations across the actor and critic.

The paper is well written with clear derivations. I suggest the publication of this paper.

**Experience Assessment:**

I have read many papers in this area.

**Review Assessment: Checking Correctness Of Derivations And Theory:**

I assessed the sensibility of the derivations and theory.

**Review Assessment: Checking Correctness Of Experiments:**

I assessed the sensibility of the experiments.

**Review Assessment: Thoroughness In Paper Reading:**

I read the paper at least twice and used my best judgement in assessing the paper.

---

> ### Author Response · Authors · 2019-11-12
> **Reply to Reviewer 1.**
>
> We appreciate your review of our work. We have addressed the issues raised by the other reviewers and revised our work accordingly.

---

### Official Review · AnonReviewer3 · 2019-10-23
**Official Blind Review #3**

**Rating:** 6

**Review:**


[Summary]
This paper studies the convergence of actor-critic algorithms with two-layer neural networks under iid assumption. Theoretical results show that, in the aforementioned setting, policy gradient and natural policy gradient converge to a stationary point at a sublinear rate and natural policy gradient's solution is globally optimal.

[Decision]
I recommend accepting this paper. While these results may not have immediate practical interest, the analysis is an important step in understanding the behavior of actor-critic algorithms with neural networks. The final revision needs to be more clear on the limiting assumptions and include a conclusion section that assembles the results.

[Comments]
The first important assumption is the architecture of the neural network. The results in the paper consider two-layer neural networks but the abstract implies that the analysis applies to general neural networks.

The second assumption is that the state-action pairs are sampled iid from the policy's stationary distribution. In reality, these samples are either gathered online (and are therefore temporally correlated), or from a buffer that is also affected by previous policies. The description of results in the abstract and introduction should clarify this setting.

Section G in the appendix shows the analysis for the projection-free method. The projection radius (R) does not seem to play a role in the new algorithm, but the convergence rate still depends on R. Does R have a different definition in this context?

Subsection 3.1 says "without loss of generality, we keep b_r fixed at the initial parameter throughout training and only update W." Whether this modification affects the optimization of the neural network and its convergence rate is not obvious to me.

The paragraph above Theorem 4.7 defines the stationary point. Is this point guaranteed to, or assumed to, exist?

**Experience Assessment:**

I have published one or two papers in this area.

**Review Assessment: Checking Correctness Of Derivations And Theory:**

I assessed the sensibility of the derivations and theory.

**Review Assessment: Checking Correctness Of Experiments:**

N/A

**Review Assessment: Thoroughness In Paper Reading:**

I read the paper at least twice and used my best judgement in assessing the paper.

---

> ### Author Response · Authors · 2019-11-12
> **Reply to Reviewer 3.**
>
> We appreciate the valuable review and suggestions. We have revised our work accordingly. In what follows, we address your concerns in detail.
>
> 1. On two-layer neural network parameterization. Thank you for the suggestion. We revise our abstract and introduction and highlight the two-layer neural network setting. See P.1 and P.2 of the revised work. Besides, our work can be readily extended to the multi-layer settings based on the recent progress in the generalization of overparameterized deep neural networks (See, e.g., [Cao et al., 2019], [Zou et al., 2019a], [Zou et al., 2019b], [Frei et al., 2019]).
>
> 2. On independent sampling. We revise our abstract and introduction and highlight the independent sampling setting. See P.1 and P.2 of the revised work. Such an assumption is imposed for the ease of analysis, which can be relaxed to a weakly dependent setting with $\beta$-mixing Markov chains.  Specifically, drawing from the Markov Chain until it mixes leads to weakly dependent data that converges to the stationary distribution, which can be handled by standard techniques ([Bhandari et al., 2018]).
>
> 3. On radius $R$ in the projection-free setting. The projection-free version of vanilla gradient descent requires the neural TD algorithm for the critic update. The radius $R$ comes from the neural TD algorithm, which is the Algorithm 2 (P. 27) in our work.
>
> 4. Keeping $b_r$ fixed in training. Such an approach is the standard procedure in the recent analysis of overparameterized neural networks (see, e.g., [Allen-Zhu et al., 2018], [Arora et al., 2019]). According to [Allen-Zhu et al., 2018], the joint training may cause confusion in some scenario, where the joint training is equivalent to the optimization solely over the last layer ([Daniely, 2017]). Besides, based on the recent progress in the generalization of deep neural networks ([Cao et al., 2019]), our work can be readily extended to the joint training regime.
>
> 5. Existence of stationary point. The existence of a stationary point is guaranteed in our setting since the parameter set is bounded and closed, and the objective is assumed to be smooth. A smooth function has a minimum on the closed and bounded domain, which is a stationary point.
>
> [Cao et al., 2019]: Yuan Cao, Quanquan Gu. (2019) Generalization Bounds of Stochastic Gradient Descent for Wide and Deep Neural Networks.
>
> [Zou et al., 2019a]: Difan Zou, Yuan Cao, Dongruo Zhou,  Quanquan Gu. (2019) Stochastic Gradient Descent Optimizes Over-parameterized Deep ReLU Networks.
>
> [Frei et al., 2019]: Spencer Frei, Yuan Cao, Quanquan Gu. (2019) Algorithm-Dependent Generalization Bounds for Overparameterized Deep Residual Networks.
>
> [Zou et al., 2019b]: Difan Zou, Quanquan Gu. (2019) An Improved Analysis of Training Over-parameterized Deep Neural Networks.
>
> [Bhandari et al., 2018]: Jalaj Bhandari, Daniel Russo, Raghav Singal. (2018) A Finite Time Analysis of Temporal Difference Learning With Linear Function Approximation.
>
> [Allen-Zhu et al., 2018]: Zeyuan Allen-Zhu, Yuanzhi Li, Zhao Song. (2018) A Convergence Theory for Deep Learning via Over-Parameterization.
>
> [Arora et al., 2019]: Sanjeev Arora, Simon S. Du, Wei Hu, Zhiyuan Li, Ruosong Wang. (2019) Fine-Grained Analysis of Optimization and Generalization for Overparameterized Two-Layer Neural Networks.

---

### Official Review · AnonReviewer2 · 2019-10-23
**Official Blind Review #2**

**Rating:** 3

**Review:**

This paper studies policy gradient where the policy is parameterized by an extremely wide neural network. The authors assume that the number of nodes ($m$) in the network is extremely large ($T^{8}R^{18}$, $T$ is the total runtime and $R$ is the radius of the function class that the network falls into), and they restrict their convergence analysis of the policy gradient algorithm in a particular function class and claims that the approximation error between the true network and the function class goes to zero when $m$ is large. The paper is written in a rigorous way and the presentation is mostly clear. I have some concerns about many of the assumptions made across the paper that are not explained or verified. This may potentially further decrease the usefulness of the analysis in this paper though the theoretical result with $m=T^8$ is already very impractical. Another major issue with this paper is that the theoretical analysis is not novel in terms of bringing new insights and results to the field given many other papers including global convergence of policy gradient (Agarwal et al., 2019;), convergence of neural TD learning (Cai et el., 2019), theory for overparameterized neural networks (Jacot et al., 2018; Allen-Zhu et al., 2018a;b; Du et al., 2018a;b; Zou et al., 2018; Chizat and Bach, 2018; Jacot et al., 2018, etc.) and other very similar papers.

There is a prior work by Agarwal et al. (2019) that proves the global convergence of both vanilla policy gradient and natural policy gradient methods. In the related work, the authors distinguish their paper from that of Agarwal et al. (2019) by claiming that they are studying the non-tabular setting and they use the actor-critic scheme. However, the first claim is incorrect because Agarwal et al. (2019) also studied the non-tabular setting (see Section 6 in Agarwal et al. (2019)) and proved an O(1/\sqrt{T}) convergence rate. Moreover, the actor-critic scheme in this paper is just a trivial modification of the nonlinear policy gradient method by calling existing result for TD learning in Cai et al. (2019). Therefore, the contribution of this paper is not so clear given existing papers.

In Algorithm 1, the policy gradient estimator $\widehat{\nabla} J(\pi_{\theta_i})$ also depends on the critic parameter $\omega_i$. It is better to show this dependency in the notation as well.

In Algorithm 1, the temperature parameter $\tau_i$ is updated in natural policy gradient but not in vanilla policy gradient. It seems that $\tau_i$ increases linearly with $i$, which makes the policy defined in eq (3.1) close to a uniform distribution when the time horizon goes to infinity. This seems to offset the update of parameter $\theta_i$.

In the update of natural policy gradient, solving eq (3.8) is really expensive in computation, especially in the setting of this paper where $m$ is chosen as $T^{8}$. It seems impossible to obtain a reasonable solution within the claimed $O(1/T^{1/4})$ runtime.

What is the function $\iota(w)$ in Assumption 4.1?

It would be better for the authors to discuss more about Assumption 4.1. It is unknown why the action-value function $Q^{\pi}$ for all policy can fall into this class.

The equation in Assumption 4.2 is exceeding the paper margin. Please make sure the paper follows the format guidelines.

Assumption 4.2 seems to be very strong. The remark after the assumption says that this condition is made on the Markov transition kernel. However, this may not be true since the assumption needs to hold for any two arbitrary policies. It is not known what kind of transition kernel $\mathcal{P}$ will satisfy this.

In each step of the neural policy gradient (Algorithm 1), the authors need to call a TD learning (Algorithm 2) to approximate the unknown action-value function $Q_{\omega_i}$ associated with the policy $\pi_{\theta_i}$ at the $i$-th step. It seems that in the learning process of ALgorithm 2, at each iteration, it samples independent data from the stationary state-action distribution which is unknown.

In the proof of Theorem 4.8, it seems that eq (D.14) and (D.15) are the same. Why it needs to be proved twice? In addition, why the equation after (D.14) holds?

The authors should provide more details about the function $u_{\hat \theta}$ defined in eq (4.4), which seems to approximate the critic function. Specifically, why are there the derivative terms instead of just the inner product term in eq (4.4).

Other comments:
In the last sentence of Section 3.1, “... approximate aompatible function approximation ...”


**Experience Assessment:**

I have published one or two papers in this area.

**Review Assessment: Checking Correctness Of Derivations And Theory:**

I carefully checked the derivations and theory.

**Review Assessment: Checking Correctness Of Experiments:**

N/A

**Review Assessment: Thoroughness In Paper Reading:**

I read the paper at least twice and used my best judgement in assessing the paper.

---

> ### Author Response · Authors · 2019-11-12
> **Reply to Reviewer 2 and clarification for a few misclaims in the review.**
>
> We appreciate the valuable review and suggestions. We have revised our work accordingly.
>
> First, we would like to point out that the reviewer seems to have made a mistake by stating that:
>
> "In the related work, the authors distinguish their paper from that of Agarwal et al. (2019) by claiming that they are studying the non-tabular setting and they use the actor-critic scheme. However, the first claim is incorrect because Agarwal et al. (2019) also studied the non-tabular setting (see Section 6 in Agarwal et al. (2019)) and proved an $O(1/\sqrt{T})$ convergence rate."
>
> Such a statement is unsubstantiated. In fact, on page 3 of our submission file, we have acknowledged the results in [Agarwal et al., 2019] on the non-tabular setting. Specifically, we wrote:
>
> " In independent work, Agarwal et al. (2019) prove that vanilla policy gradient and natural policy gradient converge to globally optimal policies at $1/\sqrt{T} $-rates in the tabular and linear setting."
>
> Here, by "linear setting", we mean that they parametrize the value function using linear functions of the score function $\nabla_{\theta} \log \pi_{\theta}$, which is known as the compatible features [Sutton et al., 2000]. Thus, their policy evaluation problem is only for linear value functions, and results such as [Bhandari et al, 2018] are readily applicable. In contrast, we parametrize the value function using neural networks, which would incur an unremovable bias in the policy gradient due to incompatibility.
>
> In what follows, we address your concerns in detail.
>
> 1. A comparison between our work and [Agarwal et al., 2019]: *simultaneous works*. According to our personal communication record with one of the authors of [Agarwal et al., 2019] (available upon request), our work and [Agarwal et al, 2019] are simultaneous works. Indeed, the first version of [Agarwal et al., 2019] is released on  August 1st, 2019, the current version is released on  August 29th, 2019, whereas our work is first released on August 29th, 2019 (arXiv link omitted here for double-blind review), and the deadline of ICLR 2020 submission is on September 27th, 2019.
>
> Besides, our work is different from [Agarwal et al., 2019] in many aspects. First, they require policy smoothness (Assumption 6.2), which does not cover our case, since the score function ($\nabla_\theta \log\pi_\theta$) has indicators when parameterized using the ReLU activation function.
>
> Second, when nonlinear functions are used to represent the value functions, we are faced with the compatibility issue, where the parametrization of value functions leads to an unremovable bias of the policy gradient. We directly tackle this challenge by proving that shared neural network architecture and random initialization of the weights lead to approximately compatible function approximations under the overparameterized regime.
>
> Moreover, we also explicitly quantify the effect of such approximate compatibility on the convergence and global optimality of policy gradient algorithms. In contrast with our work, [Agarwal et al., 2019] studies the restricted function class, where the value functions are parameterized by linear compatible functions (which is also suggested by [Sutton et al., 2000]).
>
> 2. A comparison between our work, [Cai et al., 2019], and other literature on overparameterized neural networks: different objectives. Our work has different objectives from [Cai et al., 2019]. The main objective of our work is to understand the policy gradient and natural policy gradient under the overparameterized neural-network regime. In contrast, [Cai et al., 2019] analyze the neural TD algorithm, which serves as the policy evaluation step of the algorithms we analyze. Both the analysis in [Cai et al., 2019] and the analysis in overparameterized neural networks do not carry over directly to our work, as the analysis of neural TD and stochastic gradient descent is different from that of policy gradient algorithms.
>
> Besides, combining the analysis in [Agarwal et al., 2019] and [Cai et al., 2019] does not directly lead to our results, as [Agarwal et al., 2019] does not cover the neural policy class with the ReLU activation function and they require the value functions to be linear in the score function to achieve compatibility.
>
> 3. Extremely large requirement on the network width $m$. Even in the supervised learning setting, most of the analysis ( [Jacot et al., 2018], [Allen-Zhu et al., 2018], [Du et al., 2018], [Zou et al., 2019a,b], [Chizat et al., 2018], [Arora et al., 2019]) need large network width for the corresponding generalization error guarantees. Our analysis hinges on such a regime. Therefore, our convergence and optimality guarantees also need a large network width $m$. Meanwhile, the radius $R$ here can be treated as a constant, which corresponds to the capacity of the neural network class.

---

> > ### Author Response · Authors · 2019-11-12
> > **Reply to Reviewer 2. (continued)**
> >
> >  4. Inadequate notation of $\hat\nabla_\theta J(\pi_{\theta_i})$. Thank you for the suggestion on notation. We keep the notation for simplicity, and add remarks to clarify that $\hat\nabla_\theta J(\pi_{\theta_i})$ depends on $\omega_i$. See P.6 of the revised version.
> >
> > 5. On the temperature parameter. Our work abuses the term "temperature parameter." Recall that our parameterization takes the form of $\pi\propto \exp(\tau\cdot f)$. Under standard terminology, the temperature parameter under such a parameterization should be $1/\tau$. When $\tau$ grows, the policy will become a deterministic policy eventually, which is the optimal policy. We clarify the term abuse in the definition of policy parameterization (see P.5 of the revised work) and we thank the reviewer for point it out.
> >
> > 6. Solving (3.8) with a limited budget. Though it is not our primary concern, up to minor modifications, our analysis allows for approximately solving (3.8), which is the common practice of approximate second-order optimization ([Martens, J., 2015]). Meanwhile, the count $T$ in our work is the number of policy updates (which require solving (3.8) in each update), which is not the runtime of the algorithm.
> >
> > 7. What is the function $\iota(\omega)$ ? The function $\iota(\omega)$ is arbitrary, which parameterizes the element of function class $\mathcal F_{R, \infty}$ together with $f_0$. Such a function characterizes the size of the reproducing kernel Hilbert space (RKHS) ball.  See the revised version.
> >
> > 8. On  Assumption 4.1. In Assumption 4.1, we assume that the function class falls in an RKHS, which is known to be rich. Many previous works have similar assumptions. See, for e.g., Assumption A.6 of [Farahmand et al., 2016]. Such an assumption is standard in the literature of nonparametric analysis. %{\color{red}Remove the following? which is realistic and is satisfied for rich function spaces such as RKHS.}
> >
> > 9. On Assumption 4.2. Such an assumption holds for certain behavior policy $\mu$ that is sufficiently explorative. If it holds in addition that for any policy $\pi$, the density ratios of the corresponding stationary distributions and visitation measures over that of $\mu$ has an upper bounded $L_2$-norm, then Assumption 4.2 holds by following the Cauchy-Schwartz inequality. Hence, Assumption 4.2 is in the same flavor as the concentrability assumptions, which is standard and required ([Munos et al., 2008], [Chen et al., 2019]), and is indeed an assumption over the transition kernel.
> >
> > 10. Sampling the stationary distribution of Markov Chain. The analysis of TD algorithms allows for weakly dependent data. Specifically, drawing from the Markov Chain until it mixes leads to weakly dependent data that converges to the stationary distribution, which can be handled by standard techniques ([Bhandari et al., 2018]).
> >
> > 11. Equation after (D.14). The equations after (D.14) shows that (D.14) leads to (D.15), which follows from the direct calculation. See P.22 for the revised presentation.
> >
> > 12. Understanding $u_{\hat \theta}$. The optimality of policy gradient hinges on two facts, which are the representation power of the function parameterization and the mismatch between the given policy and the optimal policy. The mismatch is characterized by the density ratios and is the major effect among the components of $u_{\hat \theta}$. In contrast, the extra term $\phi_{\hat\theta}^\top \hat\theta$ in $u_{\hat \theta}$ is a remainder that appears due to our analysis technique.

---

> > > ### Author Response · Authors · 2019-11-12
> > > **Reply to Reviewer 2. (continued)**
> > >
> > > [Agarwal et al., 2019]: Alekh Agarwal, Sham M. Kakade, Jason D. Lee, Gaurav Mahajan. (2019) Optimality and Approximation with Policy Gradient Methods in Markov Decision Processes.
> > >
> > > [Sutton et al., 2000]: Richard S. Sutton, David McAllester, Satinder Singh, Yishay Mansour. (2000) Policy Gradient Methods for Reinforcement Learning with Function Approximation.
> > >
> > > [Cai et al., 2019]: Qi Cai, Zhuoran Yang, Jason D. Lee, Zhaoran Wang. (2019) Neural Temporal-Difference Learning Converges to Global Optima.
> > >
> > > [Martens, J. et al., 2015]: Martens, J. and Grosse, R. (2015) Optimizing neural networks with kronecker-factored approximate curvature.
> > >
> > > [Farahmand et al., 2016]: Amir-massoud Farahmand, Mohammad Ghavamzadeh, Csaba Szepesv\'ari, Shie Mannor. (2016) Regularized Policy Iteration with Nonparametric Function Spaces.
> > >
> > > [Bhandari et al., 2018]: Jalaj Bhandari, Daniel Russo, Raghav Singal. (2018) A Finite Time Analysis of Temporal Difference Learning With Linear Function Approximation.
> > >
> > > [Allen-Zhu et al., 2018]: Zeyuan Allen-Zhu, Yuanzhi Li, Zhao Song. (2018) A Convergence Theory for Deep Learning via Over-Parameterization.
> > >
> > > [Cao et al., 2019]: Yuan Cao, Quanquan Gu. (2019) Generalization Bounds of Stochastic Gradient Descent for Wide and Deep Neural Networks.
> > >
> > > [Zou et al., 2019a]: Difan Zou, Yuan Cao, Dongruo Zhou,  Quanquan Gu. (2019) Stochastic Gradient Descent Optimizes Over-parameterized Deep ReLU Networks.
> > >
> > > [Frei et al., 2019]: Spencer Frei, Yuan Cao, Quanquan Gu. (2019) Algorithm-Dependent Generalization Bounds for Overparameterized Deep Residual Networks.
> > >
> > > [Zou et al., 2019b]: Difan Zou, Quanquan Gu. (2019) An Improved Analysis of Training Over-parameterized Deep Neural Networks.
> > >
> > > [Chizat et al., 2018]: Lenaic Chizat, Francis Bach. (2018) On the Global Convergence of Gradient Descent for Over-parameterized Models using Optimal Transport.
> > >
> > >  [Jacot et al., 2018]: Arthur Jacot, Franck Gabriel, Cl\'ement Hongler. (2018) Neural Tangent Kernel: Convergence and Generalization in Neural Networks.
> > >
> > > [Munos et al., 2008]: R\'emi Munos, Csaba Szepesv\'ari. (2008) Finite-Time Bounds for Fitted Value Iteration.
> > >
> > > [Chen et al., 2019]: Jinglin Chen, Nan Jiang. (2019) Information-Theoretic Considerations in Batch Reinforcement Learning.
> > >
> > > [Arora et al., 2019]: Sanjeev Arora, Simon S. Du, Wei Hu, Zhiyuan Li, Ruosong Wang. (2019) Fine-Grained Analysis of Optimization and Generalization for Overparameterized Two-Layer Neural Networks.

---

### Decision · Program_Chairs · 2019-12-19

**Decision:**

Accept (Poster)

**Comment:**

The paper makes a solid contribution to understanding the convergence properties of policy gradient methods with over-parameterized neural network function approximators.  This work is concurrent with and not subsumed by other strong work by Agarwal et al. on the same topic.  There is sufficient novelty in this contribution to merit acceptance.  The authors should nevertheless clarify the relationship between their work and the related work noted by AnonReviewer2, in addition to addressing the other comments of the reviewers.